# Harnessing macrophage-drug conjugates for allogeneic cell-based therapy of solid tumors via the TRAIN mechanism

Bartlomiej Taciak[1,2,14], Maciej Bialasek[1,2,14], Malgorzata Kubiak [1,2,14], Ilona Marszalek[1], Malgorzata Gorczak[1,2], Olha Osadchuk[1], Daria Kurpiel[1], Damian Strzemecki[1], Karolina Barwik[1], Marcin Skorzynski[3], Julia Nowakowska [1,2], Waldemar Lipiński[1], Łukasz Kiraga[1,2], Jan Brancewicz [1], Robert Klopfleisch [4], Łukasz Krzemiński[5], Emilia Gorka[1,2], Anna Smolarska[2], Irena Padzinska-Pruszynska[2], Małgorzata Siemińska[1], Jakub Guzek[2], Jan Kutner[6], Marlena Kisiala [7], Krzysztof Wozniak [7], Giacomo Parisi[8,9], Roberta Piacentini[8,9], Luca Cassetta [10], Lesley M. Forrester [11], Lubomir Bodnar [1,12], Tobias Weiss [13], Alberto Boffi[1,8,9], Paulina Kucharzewska [1,2] ✉, Tomasz P. Rygiel [1,3] ✉ & Magdalena Krol [1,2] ✉

Treatment of solid tumors remains challenging and therapeutic strategies require continuous development. Tumor-infiltrating macrophages play a pivotal role in tumor dynamics. Here, we present a Macrophage-Drug Conjugate (MDC) platform technology that enables loading macrophages with ferritin-drug complexes. We first show that macrophages actively take up human heavy chain ferritin (HFt) in vitro via macrophage scavenger receptor 1 (MSR1). We further manifest that drug-loaded macrophages transfer ferritin to adjacent cancer cells through a process termed 'TRAnsfer of Iron-binding protein' (TRAIN). The TRAIN process requires direct cell-to-cell contact and an immune synapse-like structure. At last, MDCs with various anti-cancer drugs are formulated with their safety and anti-tumor efficacy validated in multiple syngeneic mice and orthotopic human tumor models via different routes of administration. Importantly, MDCs can be prepared in advance and used as thawed products, supporting their clinical applicability. This MDC approach thus represents a promising advancement in the therapeutic landscape for solid tumors.

Despite the remarkable advances in cancer therapy, solid tumors remain a significant medical challenge due to their heterogeneity and limited blood supply resulting in a poor delivery of anticancer agents. Systemically administered drugs typically reach only a small fraction of the tumor, with their distribution limited to well-vascularized areas, leaving hypoxic regions largely untreated. These regions harbor surviving tumor cells that tend to be more aggressive and metastatic[1–3]. Additional challenges include issues with drug stability, bioavailability,

and solubility. Although cell-based therapies such as CAR-T and CAR-NK cells have shown efficacy against hematological malignancies, they encounter significant obstacles in treating solid tumors, including poor targeting, limited efficacy, and severe side effects[4].

Macrophages, which are abundant in solid tumor tissues, play a critical role in maintaining tissue homeostasis, immunity, and promotion and resolution of inflammation[5]. They are recruited to sites of tissue injury and migrate into tumors, reaching both vascular and

hypoxic regions[6,7]. This migration is primarily driven by cytokines and chemokines, such as CCL2 and CSF-1[8]. Macrophages are the only cell type that actively infiltrates these hypoxic regions, making them an attractive vehicle for delivering therapeutic agents to tumors[9–11].

Macrophage-based therapies are emerging as a promising avenue in cancer treatment, sparking increasing interest and research. Leveraging the innate ability of macrophages to infiltrate tumors, particularly into hypoxic regions, has led to the development of diverse therapeutic approaches[11]. For instance, engineered tumor-migrating macrophages have been developed as sensors for early cancer detection[8], and CAR-macrophages have demonstrated promising preclinical results in treating ovarian cancer in xenograft models[12]. These advances led to the initiation of early clinical trials for patients with various cancers, including abdominal cavity solid tumors (NCT04660929)[13]. However, CAR-macrophage technology is still in its infancy and requires further investigation to optimize its safety, efficacy, production, and administration methods. Another approach uses monocytes engineered to express IFNα under the Tie2 promoter, effectively migrating to tumors where they activate immune cells and inhibit tumor growth and angiogenesis[14]. An innovative "backpacks" approach involves attaching IFNα-containing soft particles to macrophages, thereby inducing an M1 phenotype with anti-tumor activity. When injected directly into tumors, these macrophages retain their M1 phenotype despite the immunosuppressive tumor microenvironment, which significantly reduces tumor growth and metastatic burden in mouse models[15]. Additional methods for loading macrophages with anticancer agents are being explored, including the use of macrophages to deliver oncolytic viruses (OVs) to difficult-to-access tumor regions. By employing magnetic resonance targeting with MRI scanners, OV-loaded, magnetically labeled macrophages can be directed to both primary and metastatic tumors in mice, enhancing the anti-tumor effects of macrophage-based virotherapy[16]. Another approach involves lipid nanoparticles phagocytosed by macrophages, capitalizing on M1 macrophages' tumor-infiltrating properties[17,18]. Although macrophages can engulf various nanoparticles, the release of these particles is uncontrolled, lacking a specific mechanism to target cancer cells directly. Consequently, ongoing research is essential to advance this promising field.

Our approach involves using macrophages as carriers of anticancer drugs complexed with ferritin, an iron-binding protein, in a strategy known as macrophage–drug conjugate (MDC) therapy. The ferritin nanocage architecture allows effective drug encapsulation or conjugation, overcoming challenges with drug stability, bioavailability, solubility, and toxicity[19–22]. We discovered a novel mechanism, referred to as TRAnsfer of Iron-binding proteiN (TRAIN), whereby macrophages that have efficiently taken up ferritin can transfer the cargo directly to cancer cells. We elucidated the intricate molecular mechanism underlying macrophage ferritin uptake and the TRAIN process, which involves forming an immune synapse-like structure between macrophages and cancer cells to facilitate ferritin transfer. Additionally, we generated stable ferritin–drug complexes with a broad range of anticancer drugs from different pharmacological categories, demonstrating the versatility of this platform.

In in vivo studies, we tested various MDC constructs to evaluate their therapeutic efficacy across syngeneic mouse models of lung metastasis in breast cancer, bladder cancer, head and neck cancer, and orthotopic xenograft models of pancreatic and ovarian cancers. These experiments provided substantial evidence of the platform's preclinical benefits, including reduced tumor burden and prolonged survival in mice. Importantly, our MDC platform was effective in both autologous and allogeneic settings, and stability studies showed that frozen MDC products could be safely stored for months, paving the way for an "off-the-shelf" product. This addresses significant challenges in cell-based therapies, such as the limitations of autologous approaches and manufacturing difficulties.

Our study shows that MDCs can effectively deliver anticancer drugs to tumors through the TRAIN mechanism. We demonstrate the broad therapeutic efficacy of MDCs in preclinical models of multiple tumor types, highlighting their potential as a versatile platform for solid tumor treatment. The ability to store MDCs as "off-the-shelf" products underscores their suitability for clinical translation. Based on these promising preclinical results, a Phase I clinical trial will be initiated to assess the safety and surrogates of efficacy of MDC in patients with solid tumors (Supplementary Fig. 1), with the hope to expand the landscape of cancer treatment.

## Results

### Macrophages efficiently internalize human H-ferritin (HFt) via endocytic pathway

The uptake of human heavy chain ferritin (HFt) by different leukocytes was examined in CD4+ or CD8+ T lymphocytes, monocytes, and human monocyte-derived macrophages (hMDM) (Fig. 1a). While all tested cells internalized fluorescently labeled HFt (HFt-AF488), hMDM showed the highest uptake efficiency. Our results consistently demonstrated that both primary macrophages and macrophages derived from THP-1 monocytes (Fig. 1a, b; Supplementary Fig. 2a) exhibited significantly greater HFt-AF488 uptake than their corresponding monocytes, regardless of incubation time and HFt concentration. Therefore, macrophages were selected for further studies.

Incubation with HFt at 4 °C completely blocked its internalization, indicating an energy-dependent endocytosis mechanism rather than passive membrane passage (Fig. 1c, d, Supplementary Fig. 2b). Confocal microscopy revealed that at 4 °C HFt-AF488 bound exclusively to surface receptors, as evidenced by the green fluorescence on the cell membrane. In contrast, at 37 °C, the green signal was detected in the cytoplasm, confirming internalization and delivery to early endosomes, as shown by colocalization with the early endosome antigen 1 (EEA1) marker, resulting in a yellow signal on merged images (Fig. 1d, Supplementary Fig. 2c). The uptake of HFt by macrophages obtained from different sources (hMDM, macrophages derived from human-induced pluripotent stem cells HiPSC-DMs, THP-1 monocytes, and mouse bone marrow-derived macrophages BMDM) was both time- and concentration-dependent (Fig. 1c, e, Supplementary Fig. 2a–e, g–n). This suggests that metabolic energy is crucial for this process, thereby confirming the role of endocytosis in HFt uptake. Furthermore, HFt was transported from early endosomes to lysosomes, as shown by colocalization with the lysosomal marker—LAMP1 (Fig. 1f, Supplementary Fig. 3a). The stability of the fluorescently labeled HFt within macrophages was confirmed by colocalization with anti-ferritin antibodies (Supplementary Fig. 2f).

Since the concentration-dependent uptake curve began to plateau at higher concentrations, we investigated whether HFt uptake is saturable, suggesting a receptor-mediated endocytosis process. To investigate this further, hMDM and THP-1 macrophages were incubated with HFt-AF488 in the presence of increasing concentrations of unlabeled HFt as a competitor. As expected, the presence of unlabeled HFt significantly inhibited HFt-AF488 uptake in both hMDM (Fig. 1g, h) and THP-1 macrophages (Supplementary Fig. 3b, c) in a concentration-dependent manner. However, even at the highest concentrations of unlabeled HFt, macrophages retained a relatively high level of fluorescently conjugated HFt, indicating additional uptake mechanisms may be involved.

To assess alternative pathways of HFt uptake, we utilized macropinocytosis inhibitors (EIPA, Rottlerin, and Cytochalasin D), to block membrane ruffling. The inhibition of HFt internalization by these inhibitors suggests that micropinocytosis also contributes to ferritin uptake in macrophages (Fig. 1i–l, Supplementary Fig. 3d–p)[23]. Thus, our findings indicate that, while receptor-mediated endocytosis serves as the primary route for HFt entry into macrophages, micropinocytosis plays a complementary role.

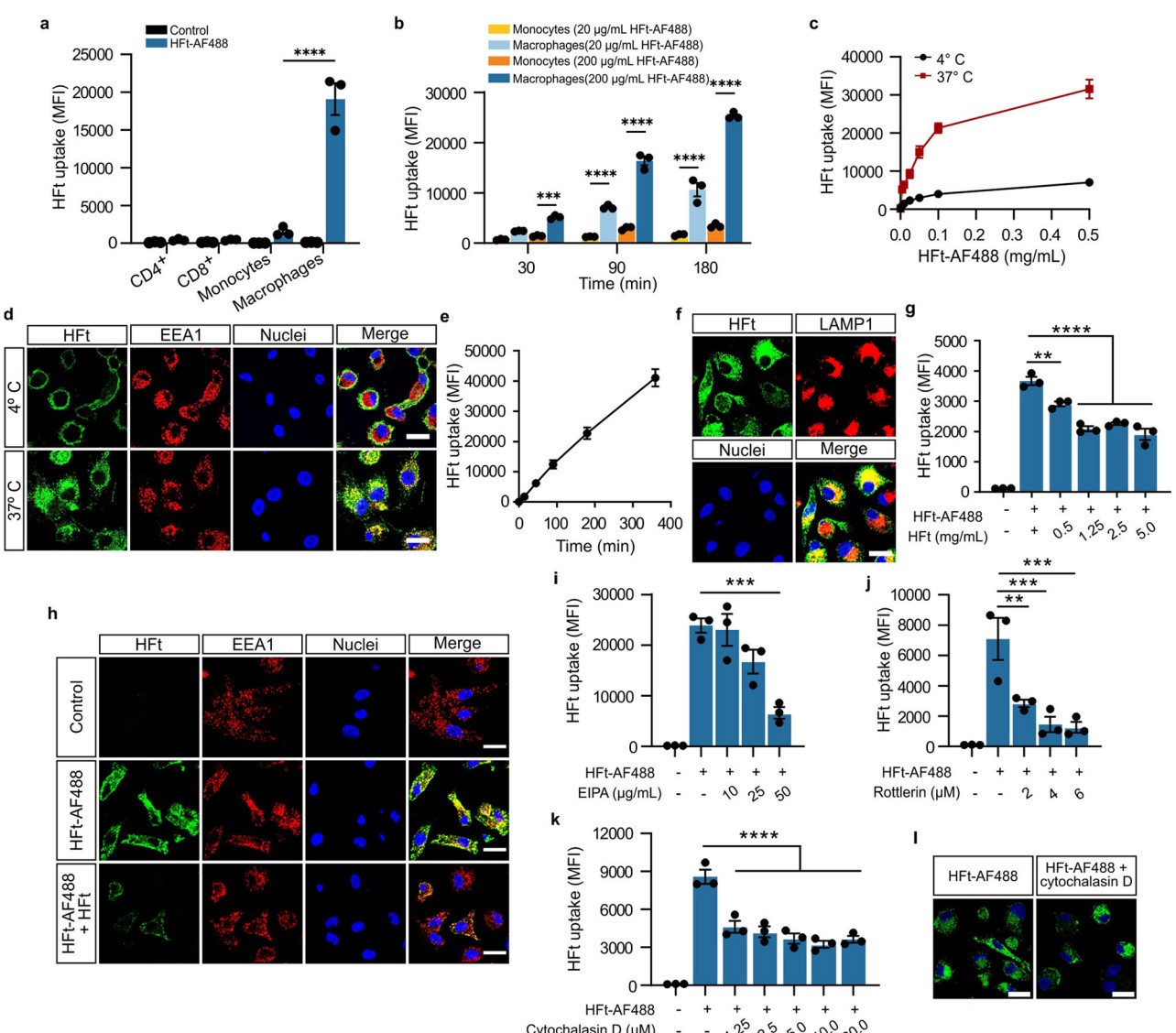

**Fig. 1 | H-ferritin uptake by human macrophages is an endocytosis-dependent process. a** Flow cytometry analysis of internalized HFt-AF488 by primary CD4[+] and CD8[+] lymphocytes, monocytes and macrophages within 30 min at 37 °C; $n = 3$ biologically independent replicates. **b** Flow cytometry analysis of internalized HFt-AF488 by primary monocytes and hMDM at indicated concentrations and time points at 37 °C; $n = 3$ biologically independent replicates. **c** Flow cytometry analysis of internalized HFt-AF488 by hMDM at various concentrations (5, 10, 25, 50, 100, 500 µg/ml) within 30 min at 4 or 37 °C; $n = 3$ biologically independent replicates. **d** Representative confocal microscopy images of internalized HFt-AF488 (green) by hMDM at a concentration of 50 µg/ml within 30 min at 4 or 37 °C. After HFt-AF488 uptake, cells were fixed and stained with anti-EEA1 antibody (red) and Hoechst 33342 (blue). Merged fluorescence images show colocalization of EEA1 marker and HFt-AF488 (yellow foci) in macrophages incubated at 37 °C. Scale bar = 20 µm. **e** Flow cytometry analysis of internalized HFt-AF488 (100 µg/ml) by hMDM at indicated time points at 37 °C; $n = 3$ biologically independent replicates. **f** Representative confocal microscopy images of internalized HFt-AF488 (green) by hMDM at a concentration of 50 µg/ml within 60 min at 37 °C. After HFt-AF488 uptake, cells were fixed and stained with anti-LAMP1 antibody (red) and Hoechst 33342 (blue). Merged fluorescence images show colocalization of LAMP1 and HFt-AF488 (yellow foci) in macrophages. Scale bar = 20 µm. **g** Flow cytometry analysis

of internalized HFt-AF488 (10 µg/ml) by hMDM in the absence or presence of indicated concentrations of unlabeled HFt within 30 min at 37 °C; $n = 3$ biologically independent replicates. **h** Representative confocal microscopy images of internalized HFt-AF488 (10 µg/ml) (green) by hMDM in the absence or presence of unlabeled HFt (2.5 mg/ml) within 30 min at 37 °C. After HFt-AF488 uptake, cells were fixed and stained with anti-EEA1 antibody (red) and Hoechst 33342 (blue). Scale bar = 20 µm. **i–k** Flow cytometry analysis of internalized HFt-AF488 (50 µg/ml) by hMDM in the absence or presence of various inhibitors of macropinocytosis: EIPA (**i**), rottlerin (**j**) and cytochalasin D (**k**) at indicated concentrations within 30 min at 37 °C; $n = 3$ biologically independent replicates. **l** Representative confocal microscopy images of internalized HFt-AF488 (50 µg/ml) (green) by hMDM in the absence or presence of cytochalasin D (10 µM) within 30 min at 37 °C. After HFt-AF488 uptake, cells were fixed and stained with Hoechst 33342 (blue). Scale bar = 20 µm. Flow cytometry data are presented as the mean fluorescence intensity (MFI) of HFt-AF488. Data are presented as mean ± SEM. The two-way ANOVA and Tukey's post-hoc tests were used for statistical analysis in panels **a** and **b**. The one-way ANOVA and Dunnett's post-hoc tests were used for statistical analysis in panels **g** and **i-k**. For all panels, **$P ≤ 0.01, ***P ≤ 0.001, ****P ≤ 0.0001. Source data are provided as a Source Data file.

## Clathrin-dependent endocytosis is involved in HFt uptake by macrophages

Given the established role of endocytosis in HFt uptake by macrophages, we aimed to determine the specific contribution of the

clathrin-dependent pathway. We utilized two well-established inhibitors, chlorpromazine (CPZ) and dynasore, to assess this process. Macrophages were treated with various concentrations of CPZ in a serum-free medium, followed by incubation with HFt-AF488 or control

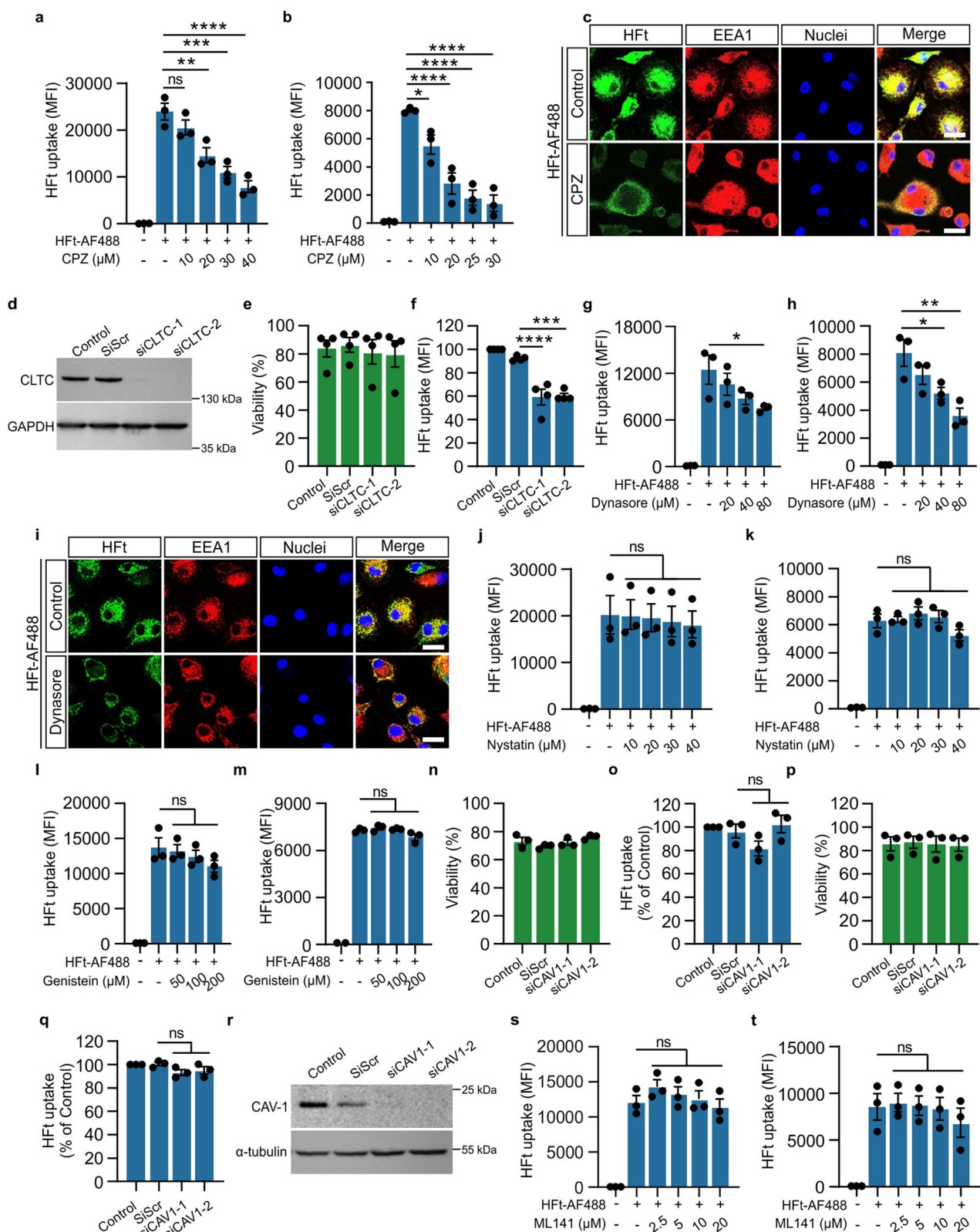

ligands: transferrin (Tfn-AF488), a known clathrin-mediated endocytosis marker[24] and cholera toxin B subunit (CTxB-AF647), which serves as a negative control (Fig. 2a–c; Supplementary Fig. 4a–g). CPZ treatment led to a significant, concentration-dependent reduction in HFt-AF488 uptake in both hMDM and THP-1 cells (Fig. 2a–c). Importantly, CPZ did not affect cell viability in the concentration range of 10–30 μM, confirming that reduced uptake was not due to cytotoxicity

(Supplementary Fig. 4c, g). Confocal microscopy confirmed the flow cytometry findings, showing reduced HFt uptake in CPZ-treated cells (Fig. 2c, Supplementary Fig. 4d).

Further validation came from silencing clathrin heavy chain (CLTC) using siRNA in THP-1 macrophages, which resulted in approximately 40% reduction in HFt uptake (Fig. 2d–f). This suggests a critical role for clathrin-mediated endocytosis in HFt internalization.

**Fig. 2 | Human macrophages internalize H-ferrtin via clathrin-dependent endocytosis. a** and (**b**) Flow cytometry analysis of internalized HFt-AF488 (100 µg/ml) by hMDM (**a**) and THP-1 macrophages (**b**) in the absence or presence of chlorpromazine (CPZ) at indicated concentrations within 30 min at 37 °C. Data are mean ± SEM from $n = 3$ separate donors (**a**, hMDM) or $n = 3$ independent replicates (**b**, THP-1). **c** Representative confocal microscopy images of internalized HFt-AF488 (100 µg/ml) (green) by hMDM in the absence or presence of CPZ (30 µM) within 30 min at 37 °C. After HFt-AF488 uptake cells were fixed and stained with anti-EEA1 antibody (red) and Hoechst 33342 (blue). Scale bar = 20 µm. **d** Western blot analysis showing clathrin heavy chain (CLTC) expression in either untransfected THP-1 macrophages (Control) or cells transfected with one of the following siRNA sequences: scramble siRNA (siScr), no. 1 siRNA targeting CLTC (siCLTC-1), no. 2 siRNA targeting CLTC (siCLTC-2), 72 h after transfection. **e** Flow cytometry analysis of THP-1 macrophages' viability at 72 h upon transfection with scramble siRNA (siScr) or siRNA targeting CLTC (siCLTC-1 and siCLTC-2). For comparison, untreated, control cells were used. Data are presented as % of live cells. **f** Flow cytometry analysis of internalized HFt-AF488 (100 µg/ml) by THP-1 after CLTC gene-knockdown within 30 min at 37 °C. For comparison, untreated cells, control cells, or cells treated with a negative scramble control siRNA (siScr) were used. Flow cytometry data are presented as % of HFt-AF488 uptake in control cells. **g** and (**h**) Flow cytometry analysis of internalized HFt-AF488 (100 µg/ml) by hMDM (**g**) and THP-1 macrophages (**h**) in the absence or presence of dynasore at indicated concentrations within 30 min at 37 °C. **e–h** Data are mean ± SEM from $n = 3$ separate donors (hMDM) or $n = 4$ (**e** and **f**) independent replicates (THP-1). **i** Representative confocal microscopy images of internalized HFt-AF488 (100 µg/ml) (green) by hMDM in the absence or presence of dynasore (80 µM) within 30 min at 37 °C. After HFt-AF488 uptake, cells were fixed and stained with anti-EEA1 antibody (red) and Hoechst 33342 (blue). Scale bar = 20 µm. **j** and (**k**) Flow cytometry analysis of internalized HFt-AF488 (50 µg/ml) by hMDM (**j**) and THP-1 macrophages (**k**) in the absence or presence of nystatin at indicated concentrations within 30 min at 37 °C. **l** and (**m**) Flow cytometry analysis of internalized HFt-AF488 (50 µg/ml) by hMDM (**l**) and THP-1 macrophages (**m**) in the absence or presence of genistein at indicated concentrations within 30 min at 37 °C. **j–m** Data presented as mean ± SEM from $n = 3$ separate donors (hMDM) or $n = 3$ independent replicates (THP-1). **n** and (**p**) Flow cytometry analysis of hMDM (**n**) and THP-1 macrophages' (**p**) viability at 72 h upon transfection with scramble siRNA (siScr) or siRNA targeting CAV1 (siCAV1-1 and siCAV1-2). For comparison, untreated, control cells were used. Data are presented as mean ± SEM % of live cells from $n = 3$ separate donors (hMDM) or $n = 3$ independent replicates (THP-1). **o** and (**q**) Flow cytometry analysis of internalized HFt-AF488 (50 µg/ml) by hMDM (**o**) and THP-1 macrophages (**q**) after CAV1 gene-knockdown within 30 min at 37 °C. For comparison, untreated cells, control cells, or cells treated with a negative scramble control siRNA (siScr) were used. Data are presented as mean ± SEM % of live cells from $n = 3$ separate donors (hMDM) or $n = 3$ independent replicates (THP-1). **r** Western blot analysis showing CAV1 expression in either untransfected THP-1 macrophages (Control) or cells transfected with one of the following siRNA sequences: scramble siRNA (siScr), no. 1 siRNA targeting CAV1 (siCAV1-1), no. 2 siRNA targeting CAV1 (siCAV1-2), 72 h after transfection. **s** and (**t**) Flow cytometry analysis of internalized HFt-AF488 (50 µg/ml) by hMDM (**s**) and THP-1 macrophages (**t**) in the absence or presence of ML-141 at indicated concentrations within 30 min at 37 °C. Flow cytometry data are presented as the mean fluorescence intensity (MFI) of HFt-AF488. Data presented as mean ± SEM from $n = 3$ separate donors (**s**, hMDM) or $n = 3$ independent replicates (**t**, THP-1). The one-way ANOVA and Dunnett's post-hoc tests were used for statistical analysis. For all panels, *$P \leq 0.05$, **$P \leq 0.01$, ***$P \leq 0.001$, ****$P \leq 0.0001$. Source data are provided as a Source Data file.

To extend these findings, we tested the effect of the dynamin GTPase inhibitor dynasore, which also significantly decreased HFt-AF488 and Tfn-AF488 uptake by both hMDM and THP-1 macrophages in a concentration-dependent manner without affecting cell viability (Fig. 2g–i; Supplementary Fig. 4h–l) Confocal images confirmed reduced cytoplasmic fluorescence signal in dynasore-treated cells, supporting its inhibitory effect on HFt internalization (Fig. 2i, Supplementary Fig. 4j).

To evaluate the involvement of other endocytic pathways, we used inhibitors for lipid raft-mediated endocytosis, including caveolin-mediated endocytosis (Nystatin, Genistein and siRNA targeting caveolin), and CDC42-dependent endocytosis (ML141)[25,26] (Fig. 2j–t). None of these inhibitors significantly blocked HFt uptake, suggesting a minimal contribution from these pathways. Importantly, cell viability remained unaffected across all treatments (Fig. 2n, p; Supplementary Fig. 4m–r). These results collectively indicate that clathrin-mediated endocytosis is the primary route for HFt uptake in human macrophages, with a potential minor role in macropinocytosis.

Transferrin receptor 1 (TfR1) is a significant receptor for HFt in human cells[27], with its expression notably higher in macrophages compared to monocytes (Supplementary Fig. 5a). Our competition assays and surface staining demonstrated that TfR1 plays a crucial role in HFt uptake by human macrophages, as indicated by its internalization and subsequent degradation following HFt binding (Supplementary Fig. 5b). Specifically, HFt stimulation led to increased degradation of TfR1 compared to control cells treated with cycloheximide (CHX) alone, suggesting that the TfR1–HFt complex is translocated to lysosomes in both hMDM and THP-1 macrophages (Supplementary Fig. 5c–f).

Further support for TfR1's involvement came from colocalization studies showing HFt accumulation in lysosomes, as indicated by LAMP1 marker colocalization (Fig. 1f, Supplementary Fig. 3a). However, knockdown of TfR1 resulted in only a modest decrease in HFt uptake, while significantly reducing Tfn internalization, highlighting a functional impact on Tfn but a limited effect on HFt (Supplementary Fig. 5g–k). Importantly, TfR1 knockdown did not impair cell viability, ensuring that the observed changes in uptake were not due to cytotoxicity (Supplementary Fig. 5i).

These findings suggest that while TfR1 contributes to HFt internalization, it is not the primary receptor responsible for this process in human macrophages. This indicates the presence of additional receptors that play a more dominant role in mediating HFt uptake, which aligns with our previous observations of alternative pathways involved in macrophage endocytosis.

## Class A scavenger receptor MSR1 identified as the receptor responsible for HFt uptake by macrophages

To identify additional receptors involved in HFt uptake, we focused on class A scavenger receptors (SR-A), given their known interaction with ferritin[28] and high expression in macrophages[29]. Co-incubation of macrophages (both hMDM and THP-1) with acetylated low-density lipoprotein (AcLDL)[30] and HFt showed colocalization, suggesting that both ligands use a shared pathway (Fig. 3a, Supplementary Fig. 6a). Further competition assays with unlabeled AcLDL significantly reduced HFt uptake in a dose-dependent manner, confirming that SR-A receptors are involved (Fig. 3b–e, Supplementary Fig. 6b–e).

Additional experiments using SR-A ligands poly(I) and poly(G), which are known to induce receptor internalization[31], demonstrated a significant decrease in HFt uptake without affecting cell viability (Fig. 3f, I; Supplementary Fig. 7a, d, g, j, m). Conversely, structurally related negative controls, such as poly (C), did not impact HFt or AcLDL internalization (Fig. 3f, i; Supplementary Fig. 7a, d, g, j, m). These findings were further corroborated using sulfated polysaccharides like fucoidan and dextran sulfate, which inhibited HFt uptake, while their negative controls (mannan and chondroitin sulfate, respectively)[31] did not affect internalization (Fig. 3g, j, h, k; Supplementary Fig. 7b, c, e, f, h, i, k, l, n, o). These results support the involvement of SR-A members in this process.

We next examined the expression of three SR-A family members (MSR1, SCARA5, MARCO) in macrophages. MSR1 was highly expressed in both hMDM and THP-1 macrophages (in contrast to monocytes), whereas SCARA5 was absent, and MARCO expression was minimal (Fig. 3l, m; Supplementary Fig. 6f). Flow cytometry showed significant surface internalization of MSR1 in response to HFt stimulation,

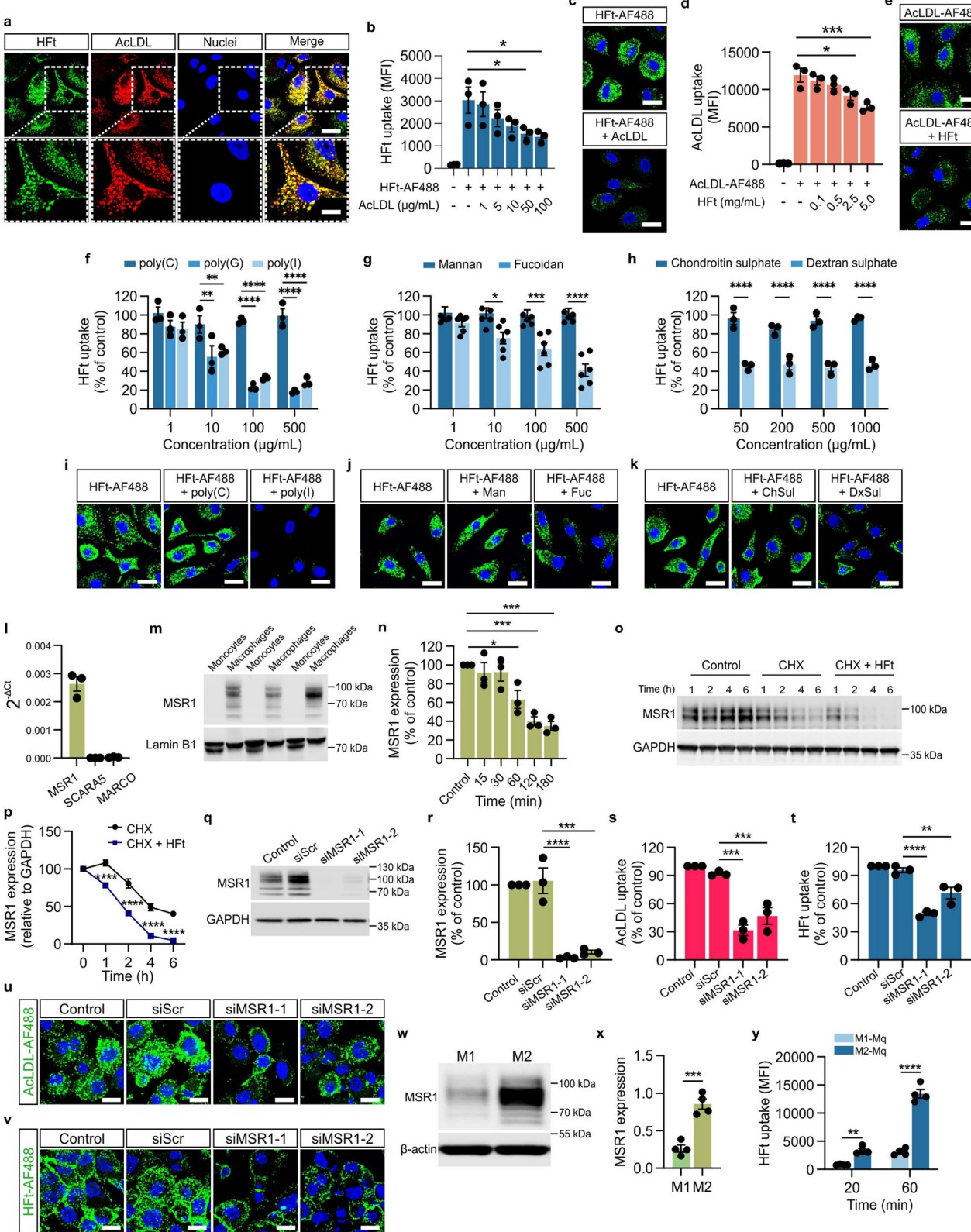

indicating its role as a receptor for HFt in macrophages (Fig. 3n; Supplementary Fig. 6g; Supplementary Fig. 8a). Further analysis confirmed that HFt stimulation increases MSR1 degradation, suggesting lysosomal translocation of the HFt-MSR1 complex (Fig. 3o, p, Supplementary Fig. 6h, i).

Knockdown of MSR1 via RNA interference caused a significant decrease in HFt uptake (by 50–60%) in both hMDM and THP-1 macrophages, along with an expected reduction in internalization of AcLDL (known MSR1 ligand), which confirmed the functional impact of transfection (Fig. 3q–t; Supplementary Fig. 6j, k; Supplementary Fig. 8b–d). In addition, confocal microscopy showed reduced intracellular HFt and AcLDL (control) levels in knockdown cells (Fig. 3u, v). These results firmly establish MSR1 as a key receptor mediating HFt uptake in human macrophages.

**Fig. 3 | MSR-1 scavenger receptor is involved in H-ferritin uptake by human macrophages. a** Representative confocal microscopy images of colocalization of HFt with AcLDL in human macrophages. hMDM were incubated simultaneously with HFt-AF488 (50 μg/ml) (green) and AcLDL-AF594 (5 μg/ml) (red) for 15 min at 37 °C. Nuclei were stained with Hoechst 33342 (blue). Merged fluorescence images show colocalization of HFt and AcLDL (yellow foci) for hMDM. Scale bar = 20 μm (10 μm in zoomed region). **b** Flow cytometry analysis of internalized HFt-AF488 (5 μg/ml) by hMDM in the absence or presence of AcLDL at indicated concentrations within 30 min at 37 °C. Flow cytometry data are presented as the mean fluorescence intensity (MFI) of HFt-AF488. Data are presented as mean ± SEM from $n = 3$ different donors. **c** Representative confocal microscopy images of internalized HFt-AF488 (5 μg/ml) (green) by hMDM in the absence or presence of AcLDL (100 μg/ml) within 30 min at 37 °C. Nuclei were stained with Hoechst 33342 (blue). Scale bar = 20 μm. **d** Flow cytometry analysis of internalized AcLDL-AF488 (1 μg/ml) by hMDM in the absence or presence of HFt at indicated concentrations within 30 min at 37 °C. Data are presented as mean fluorescence intensity (MFI) of AcLDL-AF488. Data are presented as mean ± SEM from $n = 3$ different donors. **e** Representative confocal microscopy images of internalized AcLDL-AF488 (1 μg/ml) (green) by hMDM in the absence or presence of HFt (2,5 mg/ml) within 30 min at 37 °C. Nuclei were stained with Hoechst 33342 (blue). Scale bar = 20 μm. **f–h** Flow cytometry analysis of internalized HFt-AF488 (50 μg/ml) within 30 min at 37 °C by hMDM untreated or pre-treated for 30 min at 37 °C with indicated concentrations of various ligands of class A scavenger receptors or structurally related ligands that do not bind to this group of receptors (negative controls): poly(I), poly(G) and poly(C) (control) (**f**), fucoidan and mannan (control) (**g**) and dextran sulfate or chondroitin sulfate (control) (**h**) Flow cytometry data are presented as % of HFt-AF488 uptake in untreated, control cells. Data are presented as mean ± SEM from $n = 3$ (**f**, **h**) or 6 (**g**) different donors (hMDM). **i–k** Representative confocal microscopy images of internalized HFt-AF488 (50 μg/ml) (green) within 30 min at 37 °C by hMDM untreated or pre-treated for 30 min at 37 °C with indicated concentrations of various ligands of class A scavenger receptors or structurally related ligands that do not bind to this group of receptors (negative controls): poly(I) and poly(C) (control) (**i**), fucoidan and mannan (control) (**j**) and dextran sulfate or chondroitin sulfate (control) (**k**). Nuclei were stained with Hoechst 33342 (blue). Scale bar = 20 μm. **l** Quantitative real-time PCR analysis of MSR1, SCARA5, and MARCO mRNA expression in hMDM. Data are presented as mean ± SEM from $n = 3$ different donors (hMDM). **m** Western blot analysis of MSR1 protein expression in monocytes and hMDM. Representative western blot images are shown. **n** Flow cytometry analysis of MSR1 cell surface staining in hMDM upon stimulation with HFt (200 μg/ml) for indicated time points at 37 °C. For comparison, untreated cells were used (Control). Data are presented as mean ± SEM % of MSR1 expression in control cells, $n = 3$ independent donors (hMDM). **o** Western blot analysis of MSR1 protein expression in hMDM that were untreated (Control) or treated with cycloheximide (CHX) (20 μg/ml) for 1 h prior to HFt stimulation (200 μg/ml) (CHX+HFt) for the indicated time points at 37 °C. For comparison, cells treated only with CHX were used. Representative western blot images are shown. **p** Quantitative analysis of relative MSR1 expression in hMDM shown in (**o**). Data are presented as mean ± SEM from $n = 3$ independent replicates. **q** Western blot analysis showing MSR1 expression in either untransfected hMDM (Control) or cells transfected with one of the following siRNA sequences: scramble siRNA (siScr), no. 1 siRNA targeting MSR1 (siMSR1-1), no. 2 siRNA targeting MSR1 (siMSR1-2) at 72 h after transfection. **r** Quantitative analysis of western blot for MSR1 expression shown in (**q**). Data are presented as mean ± SEM % of MSR1 expression in control cells (Control), $n = 3$ different donors. **s** and (**t**) Flow cytometry analysis of internalized AcLDL-AF488 (5 μg/ml) (**s**) or HFt-AF488 (100 μg/ml) (**t**) by hMDM after MSR1 gene-knockdown within 30 min at 37 °C. For comparison, untreated cells (Control) and cells treated with a negative, scramble control siRNA (siScr) were used. Flow cytometry data are presented as mean ± SEM % of ligand uptake in control cells (Control), $n = 3$ different donors. **u** and (**v**) Representative confocal microscopy images of internalized AcLDL-AF488 (5 μg/ml) (green) (**u**) and HFt-AF488 (50 μg/ml) (green) (**v**) within 30 min at 37 °C by THP-1 macrophages after MSR1 gene-knockdown. Afterwards, cells were fixed and stained with Hoechst 33342 (blue). Scale bar = 20 μm. **w** Western blot analysis of MSR1 protein expression in hMDM macrophages polarized to M1 or M2 phenotypes. Representative western blot images are shown. **x** Relative expression of MSR1 in M1 and M2 hMDM macrophages quantified based on western blot analysis shown in (**w**). Data are presented as mean ± SEM from $n = 4$ different donors. **y** Flow cytometry analysis of internalized HFt-AF488 by M1 and M2 hMDM macrophages given at 25 μg/ml within 20 or 60 min at 37 °C. Flow cytometry data are presented as the mean fluorescence intensity (MFI) of HFt-AF488. Data are presented as mean ± SEM from $n = 4$ different donors. The one-way ANOVA and Dunnett's post-hoc test were used for statistical analysis in panels **b**, **d**, **r–t**. The two-way ANOVA and Tukey's post-hoc tests were used for statistical analysis in panels **f–h**. The two-way ANOVA followed by Sidak's multiple comparisons post hoc test was used for statistical analysis in panels (**p**, **y**). The Student's $t$-test was used for statistical analysis in panel (**x**). For all panels, $*P \leq 0.05$, $**P \leq 0.01$, $***P \leq 0.001$, $****P \leq 0.0001$. Source data are provided as a Source Data file.

MSR1 is recognized as a marker of alternatively polarized (M2) macrophages[32]. Thus, we compared HFt uptake between M1 and M2 polarized macrophages. Western blot analysis confirmed higher MSR1 expression in M2 macrophages, which correlated with significantly higher HFt internalization compared to M1 cells (Fig. 3w–y; Supplementary Fig. 6l; Supplementary Fig. 8e–m).

### MSR1 interacts with HFt and mediates its cellular uptake
To conclusively confirm MSR1's role in HFt uptake, we overexpressed MSR1 in HEK293 and CHOK1 cells, which do not naturally express this receptor (Supplementary Fig. 9a, b; Supplementary Fig. 9e, f, respectively). Overexpression led to a substantial increase in both HFt and AcLDL (used as a control) internalization (Fig. 4a–f; and Supplementary Fig. 9c, d, g, h). Binding assays using AlphaScreen technology[33,34] and isothermal spectral shift analysis demonstrated a specific interaction between MSR1 and HFt, with competitive binding assays yielding an $IC_{50}$ of 14.7 nM (Fig. 4g–i).

Deletion of the SRCR domain in MSR1 significantly reduced HFt uptake in transfected cells, indicating its critical role in HFt binding and internalization (Fig. 4j–n). These results underscore the importance of the SRCR domain in MSR1-mediated HFt uptake, highlighting its potential in macrophage-based drug delivery systems.

### Macrophages transfer ferritin to cancer cells
Our initial experiments revealed that macrophages can transfer HFt to cancer cells in co-culture, a phenomenon observed in both human and mouse cancer cell lines (Fig. 5a–c, j; Supplementary Fig. 10a, b, Supplementary Videos 1–3). Time-lapse videos showed rapid transfer of HFt, with cancer cells acquiring HFt from macrophages within minutes. This transfer was efficient across different macrophage types, including hMDM and HiPSC-derived macrophages, and was independent of macrophage polarization state (Fig. 5c–j).

To compare the transfer properties of another protein, we also examined fluorescently labeled transferrin (Tfn) and bovine serum albumin (BSA). Macrophages transferred Tfn effectively to cancer cells, while BSA transfer was significantly lower, despite similar loading conditions (Fig. 5k, Supplementary Fig. 10c–f). Competition assays demonstrated that HFt transfer was preferred over BSA, indicating selectivity for iron-binding proteins (Supplementary Fig. 8e, f). We termed this selective transfer mechanism "TRAnsfer of Iron-binding proteiN" (TRAIN).

The efficiency of the TRAIN process was influenced by the macrophage-to-cancer cell ratio, with higher ratios resulting in increased transfer efficiency (Supplementary Fig. 10g, h). Time-course analysis showed a decrease in HFt-associated fluorescence in macrophages, concurrent with an increase in cancer cells, indicating a one-way transfer from macrophages to cancer cells (Supplementary Fig. 10i, j). A two-step co-culture experiment confirmed that HFt transfer was restricted to macrophages-to-cancer cells, with no secondary transfer among cancer cells (Supplementary Fig. 11a–c). This suggests limited off-target transfer, reducing potential side effects in non-malignant cells.

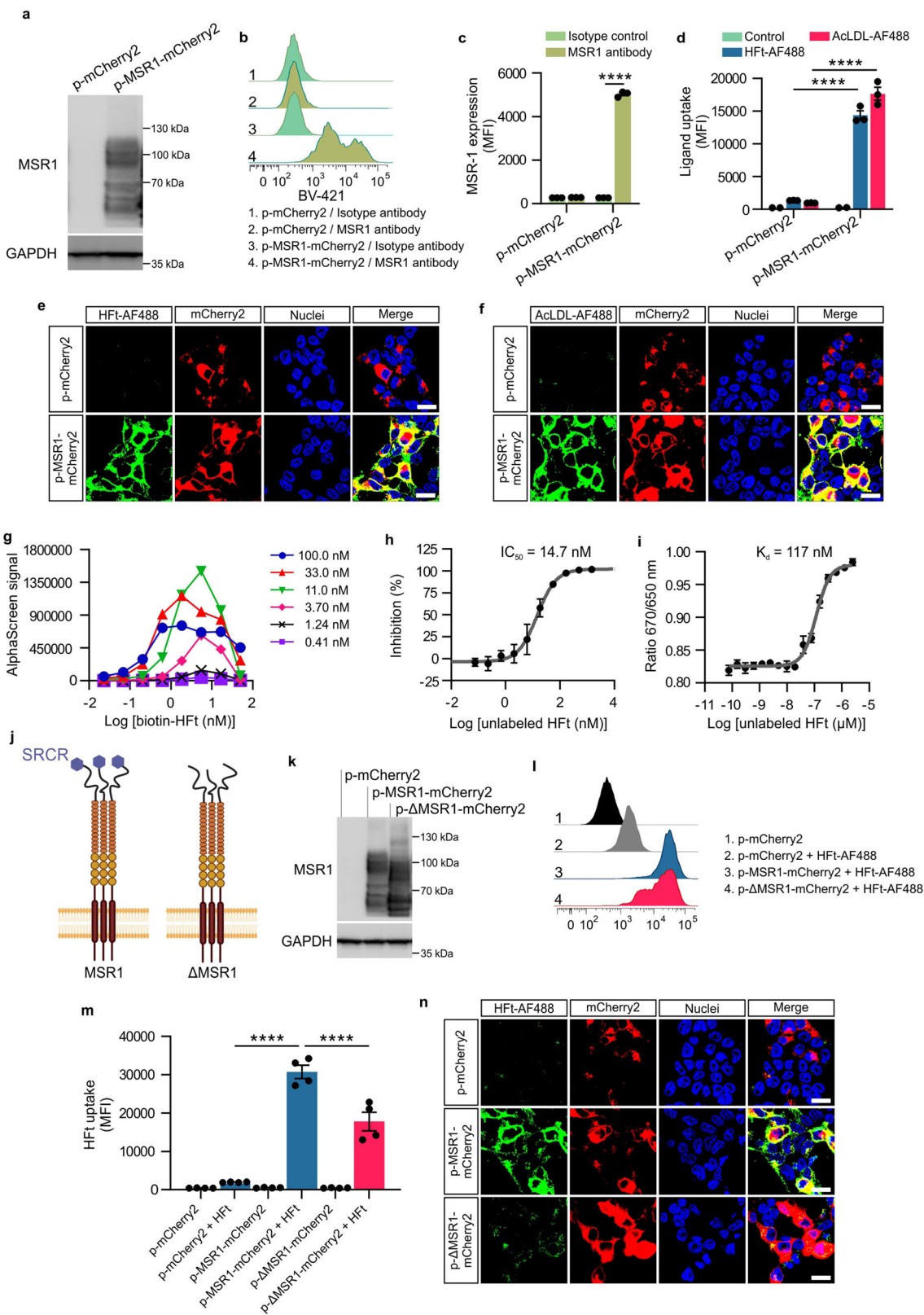

The TRAIN mechanism was further evaluated across various human cancer cell lines, with breast, ovarian, lung, pancreatic, and cervical cancers showing high levels of HFt transfer from THP-1 macrophages (Fig. 5l). Interestingly, transfer efficiency across the studied cancer cells did not depend on their expression of the ferritin receptor, TfR1 (CD71)[35] (Fig. 5m). Knocking down TfR1 in MDA-MB-231 cells impaired HFt uptake from medium but did not affect macrophage-

mediated HFt transfer, suggesting a TfR1-independent mechanism (Fig. 5n–q).

To confirm that TRAIN relies on direct cell-cell contact rather than secreted factors, we used a semipermeable filter that separated macrophages and cancer cells. HFt transfer was minimal across the filter, highlighting the need for direct interaction (Fig. 5r, s). Conditioned media experiments also showed minimal HFt uptake by cancer

**Fig. 4 | HFt binds to SRCR domain of MSR1. a** Western blot analysis of MSR1 protein expression in HEK293 cells at 24 h after transfection with the control plasmid (p-mCherry2) or the plasmid encoding the MSR1 gene (p-MSR1-mCherry2). Representative western blot images are shown. **b** and (**c**) Flow cytometry analysis of cell surface expression of MSR1 in HEK293 cells at 24 h after transfection with the control plasmid (p-mCherry2) or the plasmid encoding the MSR1 gene (p-MSR1-mCherry2). Flow cytometry data are presented as the mean fluorescence intensity (MFI) of BV421. Data are presented as mean ± SEM from $n = 3$ independent replicates. **d** Flow cytometry analysis of internalized AcLDL-AF488 (5 µg/ml) and HFt-AF488 (50 µg/ml) within 30 min at 37 °C by HEK293 cells, transfected with either p-mCherry2 or p-MSR1-mCherry2. Flow cytometry data are presented as mean fluorescence intensity (MFI) of fluorescently labeled ligands. Data are presented as mean ± SEM from $n = 3$ independent replicates. **e, f** Representative confocal microscopy images of internalized HFt-AF488 (50 µg/ml) (**e**) and AcLDL-AF488 (5 µg/ml) (**f**) (green) within 30 min at 37 °C by HEK293 cells, transfected with either p-mCherry2 or p-MSR1-mCherry2 (red). Afterwards, cells were fixed and stained with Hoechst 33342 (blue). Scale bar = 20 µm. **g** Optimization of AlphaScreen assay conditions by cross-titration of His-MSR1 (0.41–100 nM, 3-fold serial dilution) and Biotin-HFt (0.022–50 nM, 3-fold serial dilution) at fixed 10 µg/ml concentration of both nickel chelated acceptor and streptavidin donor beads. Data shown are from a single pilot run conducted for assay optimization. **h** Competitive effect of unlabeled HFt on AlphaScreen signal generated by His-MSR1 and Biotin-HFt interaction. Fixed concentrations of 4 nM His-MSR1 and 2 nM Biotin-HFt were used below the hook point of the bead assay (**g**) in the presence of increasing concentration of unlabeled HFt (0.076–1500 nM, 3-fold serial dilution). IC50 potency of unlabeled HFt in displacing Biotin-HFt from interaction with His-MSR1 was calculated by fitting it to a four-parameter nonlinear regression.

Data presented as mean ± SEM from $n = 3$ independent replicates. **i** Spectral shift dose–response curve between unlabeled HFt and His-MSR1 conjugated with the RED-tris-NTA dye. Data was analyzed using software provided by the manufacturer and is presented as the emission fluorescence ratio 670/650 vs. log(ligand concentration). $K_d$ value was calculated by fitting data with a 1:1 binding model. Data presented as mean ± SEM from $n = 5$ independent replicates. **j** Schematic representation of the full-length MSR1 and SRCR domain deletion mutant of MSR1. Created in BioRender. Taciak, B. (2024) https://BioRender.com/a34q680. **k** Western blot analysis of MSR1 protein expression in HEK293 cells at 24 h after transfection with the control plasmid (p-mCherry2), the plasmid encoding the full-length MSR1 gene (p-MSR1-mCherry2) or the plasmid encoding SRCR domain deletion mutant of MSR1 gene (p-ΔMSR1-mCherry2). Representative western blot images are shown. **l** and (**m**) Flow cytometry analysis of internalized HFt-AF488 (50 µg/ml) within 30 min at 37 °C by HEK293 cells, transfected with the control plasmid (p-mCherry2), the plasmid encoding the full-length MSR1 gene (p-MSR1-mCherry2) or the plasmid encoding SRCR domain deletion mutant of MSR1 gene (p-ΔMSR1-mCherry2). Flow cytometry data are presented as the mean fluorescence intensity (MFI) of HFt-AF488. Data are presented as mean ± SEM from $n = 4$ independent replicates. **n** Representative confocal microscopy images of internalized HFt-AF488 (50 µg/ml) within 30 min at 37 °C by HEK293 cells, transfected with the control plasmid (p-mCherry2), the plasmid encoding the full-length MSR1 gene (p-MSR1-mCherry2) or the plasmid encoding SRCR domain deletion mutant of MSR1 gene (p-ΔMSR1-mCherry2) (red). Nuclei were stained with Hoechst 33342 (blue). Scale bar = 20 µm. The two-way ANOVA followed by Sidak's multiple comparisons post hoc test was used for statistical analysis. For all panels, ****$P \le 0.0001$. Source data are provided as a Source Data file.

cells, further supporting contact-dependent transfer (Supplementary Fig. 10k, l).

To clarify that the observed TRAIN was not a result of macrophages engulfing cancer cells (phagocytosis), distinct fluorescent dyes were used to label macrophages and cancer cells in co-culture experiments. This approach allowed clear differentiation between the two cell types during flow cytometry. The results showed that most cancer cells displayed the AF488 signal, confirming successful HFt transfer. Conversely, very few macrophages exhibited the cancer cell-specific CellTrace label, ruling out significant phagocytosis. Confocal microscopy and flow cytometry further confirmed that macrophages and cancer cells remained distinct. These observations are supported by detailed imaging and flow cytometry data (e.g., Supplementary Figs. 10, 12 and Movies 1–3). Experiments with primary macrophages from different donors also showed no difference in HFt transfer efficiency between M1 and M2 polarized macrophages (Fig. 5g). Importantly, loading macrophages with Ft-drug conjugates did not trigger significant phagocytosis or alter macrophage polarization to promote engulfment (Supplementary Fig. 12c). Together, these findings confirm that TRAIN involves ferritin transfer without macrophage phagocytosis of cancer cells.

### Ferritin is transferred in vesicles from macrophages to cancer cells via immune synapse-like connection

Confocal microscopy and holotomographic imaging suggested that HFt is transferred within vesicles. Small vesicles containing HFt were observed moving from macrophages to cancer cells (Fig. 5b, f, j; Supplementary Fig. 10a, b; Supplementary Fig. 13a; Supplementary Videos 1–3). Further analysis using lipid membrane dyes[36] and anti-CD63 staining confirmed vesicular localization of HFt (Fig. 6a; Supplementary Fig. 13b). However, experiments using engineered RAW 264.7 macrophages expressing GFP in the plasma membrane showed no GFP transfer, indicating that membrane exchange is not involved (Supplementary Fig. 13c–e).

Microvesicle and exosome isolation from HFt-loaded macrophages showed minimal HFt uptake by cancer cells, indicating that extracellular vesicles[37] are not the primary transfer route (Fig. 6b–d). Additionally, exosome release inhibitors like DMA[38] did not impact TRAIN efficiency (Fig. 6e). Piceatannol, known inhibitor of tyrosine

protein kinase (Syk)-dependent signaling involved in vesicle trafficking[39] between interacting cells partially inhibited TRAIN (Fig. 6f). Endocytosis inhibitors such as Wortmannin[40] and siRNA targeting clathrin heavy chain significantly reduced HFt transfer, suggesting a role for endocytosis in cancer cell uptake of vesicle-associated HFt (Fig. 6g–k).

Quantitative imaging flow cytometry revealed F-actin accumulation at the macrophage-cancer cell interface, indicating an immune synapse-like structure (Fig. 6l, m). Inhibition of F-actin polymerization reduced TRAIN significantly, confirming the involvement of actin dynamics (Fig. 6n, o). Analysis of ICAM-1 and CD11b interactions showed that only ICAM-1 accumulation was critical for TRAIN (Fig. 6p–u). Knockdown or blocking of ICAM-1 in macrophages reduced HFt transfer, while CD11b had no effect (Fig. 6v–z). However, TRAIN was not completely inhibited, suggesting additional factors are involved.

### Ferritin drug complexation and development of MDC

Building on the TRAIN mechanism, we developed macrophage-drug conjugates (MDC) by leveraging ferritin's unique structure for drug complexation. HFt forms a 24-subunit spherical nanocage that disassembles at acidic pH and reassembles at neutral pH, a property we exploited for drug loading[41–43]. Using this pH-dependent encapsulation method, we successfully loaded doxorubicin—a widely used chemotherapeutic agent—into HFt cages, achieving efficient encapsulation (Supplementary Fig. 14a–f). Stability studies indicated that the HFt-Dox complex remained stable for several days at 4 °C and room temperature, with minimal drug leakage even after six months at −80 °C, underscoring its suitability for long-term storage (Supplementary Fig. 14c, d). Notably, the HFt-Dox complex achieved a drug-to-protein ratio of 2.2, equating to 53 drug molecules per ferritin nanocage (Supplementary Fig. 14n).

We extended this approach to complex HFt with other therapeutic agents, including kinase inhibitors (coded 250) and tubulin polymerization inhibitors (coded 735), expanding the versatility of our MDC platform. UV–Vis spectroscopy and dynamic light scattering (DLS) analyses confirmed the successful complexation of these drugs within HFt cages, with a high degree of homogeneity (Supplementary Fig. 14g–l). Similarly, the HFt–735 complex demonstrated stable conjugation, with no significant aggregation, preserving the structural

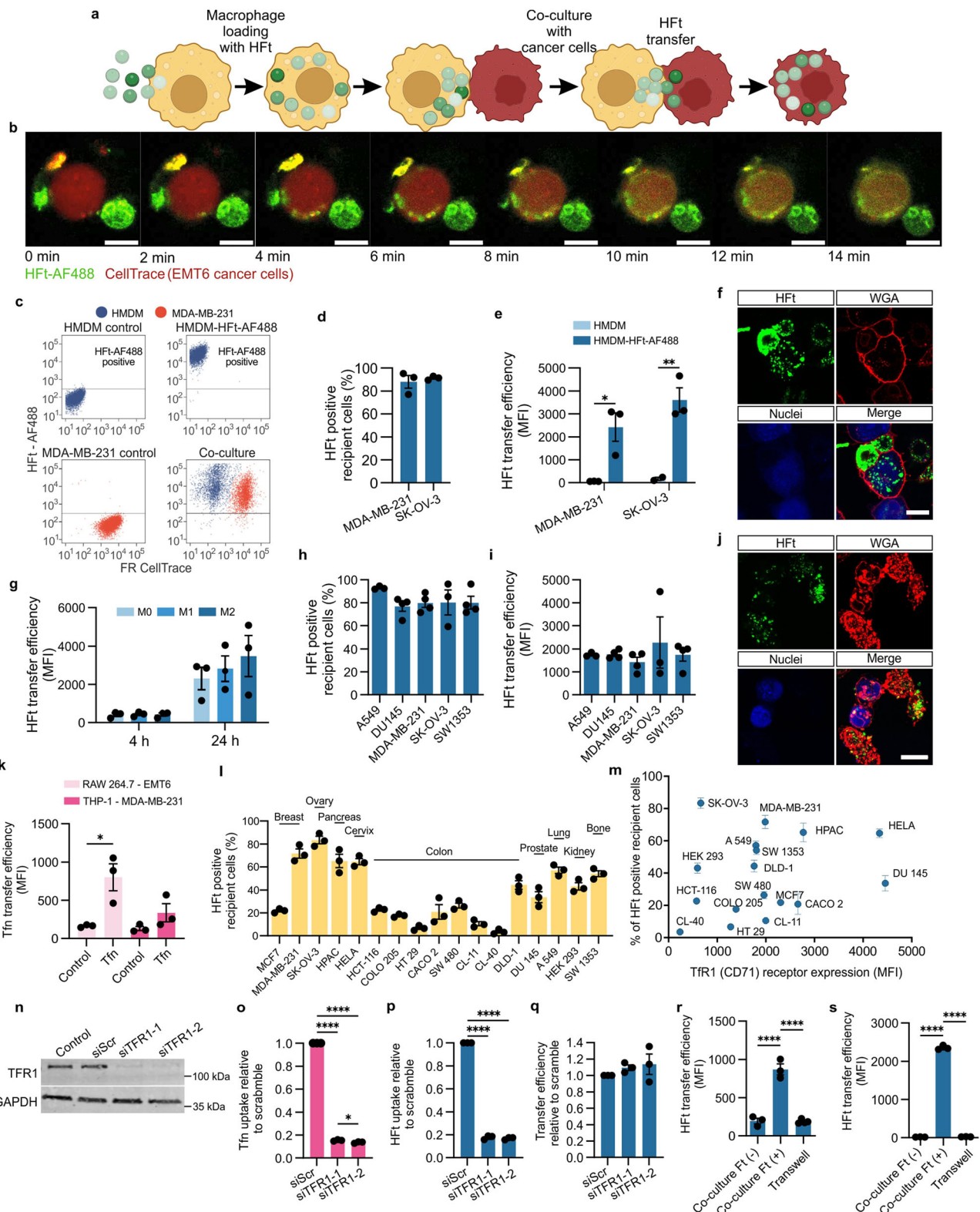

integrity of ferritin as confirmed by DLS and HPLC-MS (Supplementary Fig. 14l, o). To test the most efficient loading conditions, different types of macrophages, including BMDM, hMDM, and HiPSC-DMs, were loaded with the HFt-Dox complex, achieving loading capacities of up to 1.44 µg per $1 \times 10^6$ BMDM under optimal conditions (Supplementary Fig. 14m).

Given the diverse mechanisms of action of different anticancer drugs, we explored the versatility of our encapsulation strategy by preparing HFt complexes with 20 different drugs belonging to various pharmacological families (Supplementary Fig. 15a). The stability of complexes was assessed, showing that HFt-drug formulations retained their structural integrity even after prolonged storage (Supplementary Fig. 14e). The efficacy of other MDCs loaded with different anticancer drugs was demonstrated in vitro using several mouse cancer models (Supplementary Fig. 15b). Comparative studies also revealed that HFt-735-loaded macrophages exhibited superior cancer cell killing ability

**Fig. 5 | Macrophages transfer ferritin to cancer cells. a** Schematic representation of the macrophage loading with HFt and subsequent transfer to cancer cells. Created in BioRender. Taciak, B. (2024) https://BioRender.com/s50b255. **b** Snapshots from the movie recorded under the confocal microscopy (Supplementary Video 1) in 15 min where BMDM macrophages loaded with HFt-FITC transfer it to EMT6 cancer cell labeled with red CellTrace. **c** Representative flow cytometry scatter plot of macrophages with internalized HFt-AF488 and MDA-MB-231 breast cancer cells labeled with CellTrace (CellTrace Far Red−APC). **d** and (**e**) Flow cytometry quantification of HFt-AF488 transfer from hMDM and (loaded with HFt-AF488 at concentration 500 μg/ml for 1 h at 37 °C) to various human cancer cell lines at 24 h following co-culture at 2:1 ratio (macrophages: cancer cells). Data are presented as mean ± SEM from n = 3 different donors (hMDM). **f** Representative confocal microscopy images of HFt-AF488 (green) transfer from hMDM (loaded with HFt-AF488 at concentration 500 μg/ml for 1 h at 37 °C) to MDA-MB-231 breast cancer cells (blue) after 24 h of co-culture. Scale bar = 10 μm. Wheat Germ Agglutinin (WGA), Alexa Fluor-555 Conjugate (red) was used to visualize the cell membrane. **g** Comparative analysis of HFt-AF488 transfer from hMDM of different polarization states−M0, M1 (stimulated with LPS, 100 ng/ml), and M2 (stimulated with IL-4, 20 ng/ml)−to MDA-MB-231 breast cancer cells. hMDM were pre-loaded with HFt-AF488 (500 μg/ml) for 1 h at 37 °C, followed by co-culture with MDA-MB-231 cells for 4 and 24 h. Control conditions included co-cultures of M0, M1, and M2 hMDM with MDA-MB-231 cells without HFt-AF488. Data are presented as mean ± SEM from n = 3 different donors (hMDM). **h** and (**i**) Flow cytometry quantification of HFt-AF488 transfer from iPSC-derived macrophages (loaded with HFt-AF488 at concentration 500 μg/ml for 1 h at 37 °C) to various human cancer cell lines at 24 h following co-culture at 2:1 ratio (macrophages: cancer cells). Data are presented as mean ± SEM from n = 3 (A549, SK-OV-3) or n = 4 (DU145, MDA-MB-231, SW1353) independent replicates. **j** Representative confocal microscopy images of HFt-AF488 (green) transfer from iPSC-derived macrophages (loaded with HFt-AF-647 at

concentration 500 μg/ml for 1 h at 37 °C) to MDA-MB-231 breast cancer cells (blue) after 24 h of co-culture. Scale bar = 10 μm. Wheat Germ Agglutinin (WGA), Alexa Fluor-555 Conjugate (red) was used to visualize the cell membrane. **k** Flow cytometry analysis of Tfn-AF647 transfer from RAW 264.7 and THP-1 macrophages (loaded with Tfn-AF-647 at concentration 0.2 mg/ml for 1 h at 37 °C) to cancer cells (EMT6 and MDA-MB-231, respectively) after 24 h of co-culture. Data are presented as mean ± SEM from n = 3 independent replicates. **l** Flow cytometry analysis of HFt-AF488 transfer from THP1-derived macrophages (loaded with HFt-AF488 at concentration 500 μg/ml for 1 h at 37 °C) to various cancer cell lines after 4 h co-culture. Data are presented as mean ± SEM from n = 3 independent replicates. **m** Correlation of the percentage of HFt-AF488-positive recipient cells from human cell lines shown in (**l**) to their CD71 (TfR1) receptor expression assessed by flow cytometry. **n** Western blot analysis showing TfR1 gene knockdown efficiency in MDA-MB-231 cells using two different siRNA sequences and scramble as a control. **o−q** Flow cytometry analysis of AF488 fluorescence in MDA-MB-231 cancer cells with TfR1 gene knockdown using two different siRNA sequences, cells transfected with negative control (Scramble) siRNA or untreated cells (Control). Effect on (**o**) Tfn-AF488 and (**p**) HFt-AF488 uptake from medium and (**q**) HFt-AF488 transfer from THP-1 macrophages in 4 h co-culture was calculated relative to Scramble. Data are presented as mean ± SEM from n = 3 independent replicates. **r** and (**s**) Flow cytometry quantification of HFt-AF488 transfer from RAW 264.7 or THP-1 macrophages (loaded with HFt-AF488 at concentration 500 μg/ml for 1 h at 37 °C) to (**r**) EMT6 or (**s**) MDA-MB-231 breast cancer cells (respectively) at 24 h following direct, or Transwell membrane-separated co-culture system (macrophages seeded on Transwell insets). Co-culture of macrophages without HFt-AF488 and cancer cells [co-culture HFt(−)] was used as a control. Data are presented as mean ± SEM from n = 3 independent replicates. Statistical analysis was performed using one-way ANOVA with post-hoc Tukey HSD test. For all panels, *P ≤ 0.05, **P ≤ 0.01, ****P ≤ 0.0001. Source data are provided as a Source Data file.

compared to BSA-735-loaded macrophages, underscoring the specificity and potency of the HFt-drug complex in the TRAIN process (Supplementary Fig. 15c).

To investigate the receptor-mediated uptake of these HFt-drug complexes by macrophages, we conjugated HFt with both the fluorescent dye AF488 and 735 drug (HFt-735-AF488). Confocal microscopy confirmed effective internalization of the HFt-drug complexes by macrophages, with colocalization of the HFt and the drug inside the cell, indicating successful intracellular delivery (Fig. 7a). Competitive binding assays using AcLDL demonstrated that HFt-735-AF488 uptake by macrophages was inhibited by AcLDL, suggesting that these complexes are internalized via scavenger receptors, specifically SR-A members (Fig. 7b, c). Further competition studies using poly(G) showed a significant reduction in HFt-735 uptake, whereas poly(C) had no effect, reinforcing the involvement of SR-A receptors (Fig. 7d−g).

MSR1 was identified as a key receptor mediating HFt-drug internalization, as confirmed by siRNA knockdown experiments, which resulted in a substantial decrease in HFt-735 uptake by macrophages (Fig. 7h). Molecular binding assays using AlphaScreen technology demonstrated specific binding between HFt-735 and MSR1, further confirming this interaction (Fig. 7i, j). This targeted uptake mechanism allows for the efficient loading of macrophages with therapeutic agents, positioning MDCs as a potent vehicle for targeted drug delivery.

We evaluated the anti-cancer efficacy of MDCs loaded with various HFt-drug complexes in vitro and in vivo. In co-culture experiments, macrophages loaded with HFt-735 (MDC-735) demonstrated significant cytotoxic effects against multiple cancer cell lines, including MDA-MB-231 (breast cancer), BxPC-3 (pancreatic cancer), SK-OV-3 (ovarian cancer), and A549 (lung cancer) (Fig. 7k, Supplementary Fig. 16). Confocal microscopy analysis showed effective transfer of HFt-735 from macrophages to cancer cells, leading to cancer cell apoptosis (Fig. 7l). Furthermore, MDC loaded with HFt-250 showed robust efficacy in reducing number of pancreatic cancer cells, with dose-dependent killing observed in co-culture assays (Supplementary Fig. 17a).

In vivo studies further demonstrated the therapeutic potential of MDC, where autologous BMDM loaded with HFt-Dox (MDC-Dox) were administered to EMT6 breast cancer models. Mice treated with MDC-Dox showed significant tumor reduction compared to controls, with no observed toxicity, highlighting the safety profile of this approach (Supplementary Fig. 18a−e).

The internalization of HFt−drug complex did not compromise macrophage viability or alter their polarization, as confirmed by flow cytometry and gene expression analysis (Supplementary Fig. 17b−d, Supplementary Figs. 19, 20). Macrophages loaded with HFt-Dox or HFt-250 retained high viability, even up to 96 h post-loading (Supplementary Fig. 17b, c). Additionally, MDCs preserved their viability and cancer-killing functionality even after cryopreservation for long months, confirming their stability as an "off-the-shelf" product[44] (Supplementary Fig. 17d, 21a−d). Gene expression profiling and flow cytometry analysis of the surface markers in fresh and frozen cells showed no significant shift towards M1 or M2 polarization, with only transient changes in inflammatory markers, indicating a favorable safety profile for therapeutic use (Supplementary Fig. 19, Supplementary Fig. 21b, c).

Overall, these findings demonstrate the versatility, stability, and therapeutic potential of the MDC platform, as an allogeneic, scalable, and off-the-shelf cancer therapy. The ability to complex various anticancer agents with ferritin nanocages and deliver them specifically to the tumors via macrophages offers a promising new avenue for targeted cancer treatment.

## MDC confirm in vivo efficacy and safety in multiple mouse and human tumor models

To assess the migration and interaction of hMDMs within the tumor microenvironment, we analyzed the mean fluorescence intensity (MFI) in ovarian cancer and lung-growing cancer models (A549 or MDA-MB231) after intravenous administration of hMDMs. Imaging done 24 h post macrophage administration confirmed that over 20% of the fluorescence signal accumulated in ovarian tumors and over 70% in the lungs with tumors, compared to the 100% signal observed in control

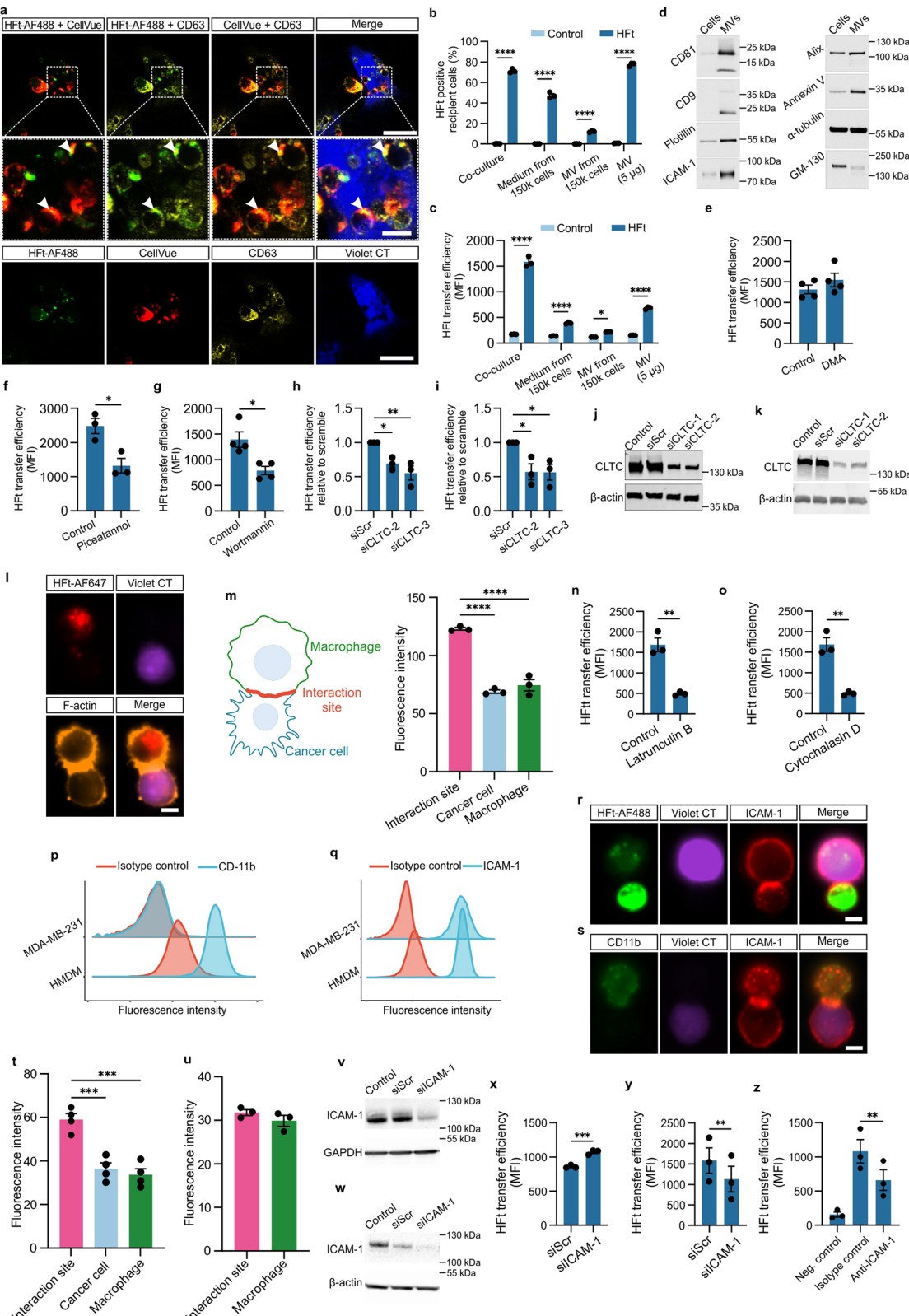

tumors receiving intratumoral macrophage injections. These results demonstrate the strong migratory and tumor-homing capabilities of hMDMs, supporting their potential as a versatile delivery platform for targeting solid tumors (Supplementary Fig. 22a–f). Fluorescence imaging also confirmed efficient localization of macrophages within the tumor stroma, primarily near tumor cell clusters, indicating successful targeting and infiltration (Supplementary Fig. 22d). In a separate study,

local administration of MDC-735 in U87MG glioblastoma tumors led to significant tumor mass reduction, with many tumors showing only small remnants or complete regression (Supplementary Fig. 22k). Immunohistochemical analysis demonstrated significant decrease in CD68+ macrophage density within the tumor stroma, suggesting targeted engagement and modulation of the tumor microenvironment by MDC-735. Additionally, Ki67 staining indicated a reduced proliferation

**Fig. 6 | Direct vesicular transfer of HFt from macrophages to cancer cells via immune synapse-like structure. a** Representative confocal microscopy images captured after 24 h co-culture of donor hMDM labeled with CellVue Claret Far Red dye (red) and loaded with HFt-AF488 (green) and MDA-MB-231 cancer cells labeled with CellTrace Violet dye (blue). Cells fixed after 24 h were additionally stained with anti-CD63 antibody (yellow). Co-localization of AF488, CellVue, and CD63 signals both in macrophages and in cancer cells are pointed out by the arrowheads. Scale bar = 20 μM (5 μm in the zoomed region). **b** and (**c**) Comparison of HFt-AF488 signal in MDA-MB-231 cancer cells when co-cultured with THP-1 macrophages for 24 h or incubated with conditioned-media or microvesicles (MV) isolated from THP-1 carrying HFt-AF488 identical to the cells used in co-culture (equivalent to the number of macrophages used in co-culture (150k) or concentrated (5 μg)). Data are presented as mean ± SEM from n = 3 independent replicates. **d** Typical MV markers found in MVs isolated from THP-1 macrophages by western blot analysis. **e–g** Flow cytometry analysis of AF488 fluorescence in EMT6 cancer cells following 4 h co-culture with HFt-AF488-loaded RAW 264.7 macrophages in the presence of (**e**) DMA, (**f**) Piceatannol, (**g**) Wortmannin or respective vehicle (Control). Data presented as mean ± SEM relative to control from n = 3 (**f**) or 4 (**e, g**) independent replicates. **h** and (**i**) Flow cytometry analysis of AF488 fluorescence in (**h**) MDA-MB-231 and (**i**) EMT6 cancer cells with CLTC gene knockdown using two different siRNA sequences relative to fluorescence in cells transfected with negative control (Scramble) siRNA after 4 h co-culture with THP-1 or RAW 264.7 macrophages, respectively, loaded with HFt-AF488. Data are presented as mean ± SEM from n = 3 independent replicates. **j** and (**k**) Knockdown efficiency in (**j**) MDA-MB-231 and (**k**) EMT6 cancer cells was confirmed with western blot analysis. **l** Representative ImageStream images and (**m**) phalloidin-AF555 fluorescence quantification analysis showing F-actin accumulation at the cell-cell contact site following 2 h co-culture of hMDM and MDA-MB-231 cancer cells (from images recorded by imaging flow cytometry, data are presented as mean ± SEM from n = 3 independent replicates). The one-way ANOVA and Tukey HSD post-hoc tests were used for statistical analysis. Schematic illustration of analyzed image areas created in BioRender. Taciak, B. (2024) https://BioRender.com/r16l469. Scale bar = 7 μm. **n** and (**o**) Flow cytometry quantification of AF488 fluorescence in EMT6 cells following 4 h co-culture with RAW 264.7 macrophages in the presence of actin polymerization inhibitors (**n**) Latrunculin B and (**o**) Cytochalasin D or vehicle (Control). Data are presented as mean ± SEM from n = 3 independent replicates. **p** and (**q**) Representative histograms showing CD11b (**p**) expression in hMDM but not MDA-MB-231 cells and ICAM-1 (**q**) expression in MDA-MB-231 and HMDM cells. Representative ImageStream images showing (**r**) ICAM-1 but not (**s**) CD11b accumulation at the cell-cell contact site following 4 h co-culture of hMDM and MDA-MB-231 cancer cells and fluorescence quantification (**t** and **u**) (from images recorded by imaging flow cytometry, data are presented as mean ± SEM from n = 4 (**t**) and n = 3 (**u**) independent replicates). The one-way ANOVA and Tukey HSD post-hoc test were used for statistical analysis. Scale bar = 7 μm. **v** and (**w**) Western blot analysis showing ICAM-1 expression in hMDM (**v**) or MDA-MB-231 cells (**w**) either untransfected named as control or transfected with one of the following siRNA sequences: scramble siRNA (siScr), no. 1 siRNA targeting ICAM-1 at 72 h after transfection. **x** and (**y**) Flow cytometry quantification of HFt-AF488 transfer from hMDM loaded with HFt-AF488 to MDA-MB-231 cancer cells after ICAM-1 gene knockdown in MDA-MB-231 (**x**) or hMDM (**y**) cells using siRNA (siICAM-1) after 48 h coculture. For comparison, cells treated with a negative scramble control siRNA (Scramble) were used. **z** Flow cytometry quantification of HFt-AF488 transfer from hMDM pre-incubated with ICAM-1 blocking antibodies and co-cultured with MDA-MB-231 cancer cells (anti-ICAM-1 blocking antibodies were used during the co-culture) for 4 h. **x–z** Data are presented as mean ± SEM from n = 3 biologically independent replicates. An unpaired t-test was used for statistical analysis in panels (**n, o, x**), a paired t-test in panel (**y**), and one-way ANOVA with Dunnett's post-hoc test in panel (**z**). For all panels, *P ≤ 0.05, **P ≤ 0.01, ***P ≤ 0.001, ****P ≤ 0.0001. Source data are provided as a Source Data file.

rate in treated tumors, highlighting MDC-735's capability to disrupt tumor growth and stromal interaction, thereby impairing tumor viability and progression. The tumor-to-stroma ratio was significantly lower in MDC-735 treated mice compared to PBS controls, indicating a robust treatment response (Supplementary Fig. 22g–j).

These findings underscore the broad antitumor efficacy of MDC therapy across different models, demonstrating its versatility as a treatment option for various cancers.

For pancreatic cancer treatment, the MDC-250 product (hMDM loaded with the HFt-250 complex) was administered intraperitoneally (i.p.) in a BxPC-3-luc orthotopic xenograft model. Mice with 14-day established tumors received various treatments, including plain macrophages, free drug 250, HFt-250, standard chemotherapy (gemcitabine), and combinations thereof (Fig. 8a). Mice treated with MDC-250 alone or in combination with gemcitabine showed significantly prolonged overall survival and substantial tumor growth inhibition (Fig. 8b–d; Supplementary Fig. 23a). Notably, the free drug equivalent demonstrated no observable therapeutic effects, underscoring the enhanced efficacy of MDC-based delivery. Long-term survival was achieved in 50% of animals receiving combination therapy, demonstrating potential synergistic benefits. Importantly, treatment was well tolerated, with no significant weight loss observed in the mice (Supplementary Fig. 23b).

Additionally, MDC-735 was administered i.p. in the same BxPC-3-luc orthotopic xenograft model as a monotherapy and in combination with gemcitabine (Fig. 8e). MDC-735 showed substantial antitumor activity, with significant reductions in tumor bioluminescence and size compared to controls (Fig. 8f–h). In combination with gemcitabine, it showed superior efficacy over all the groups. The treatment was well tolerated as reflected by the weight of the mice (Supplementary Fig. 23c).

Further evaluation in the SK-OV-3 ovarian cancer model revealed that intravenous administration of MDC-735 significantly reduced tumor burden. This treatment not only decreased tumor weight and bioluminescence but also prevented metastasis to other organs—a unique outcome not observed in other treatment groups, including those treated with paclitaxel (Fig. 8i–l). The high efficacy of MDC-735 in targeting ovarian cancer underscores its potential both as a standalone therapy and in combination with existing chemotherapeutics. The treatment showed no adverse effects on mouse body weight, further supporting its safety profile (Supplementary Fig. 23d).

In models of metastatic breast cancer (EMT6 lung metastasis), MDC-735, generated from BMDM, was administered intravenously, resulting in a marked reduction in lung metastases. Bioluminescence imaging and tumor burden analysis confirmed reduced lung colonization in MDC-735-treated mice, with lung weights comparable to those of healthy controls and a statistically significant reduction in bioluminescence signal compared to the control group (Fig. 9a–d). The treatment was well tolerated, with no significant changes in body weight, indicating minimal toxicity (Supplementary Fig. 23e). These results highlight the superior efficacy of MDC-735 over standard treatments like doxorubicin in managing metastatic disease.

Since MDC therapy is intended as an allogeneic treatment, safety was assessed by administering allogeneic macrophages intravenously in healthy mice. A single dose of autologous or allogeneic macrophages did not cause significant changes in blood counts at 7- and 14-days post-administration (Supplementary Fig. 24a). Histopathological examination showed no evidence of graft-versus-host disease (GvHD), with only mild lymphohistiocytic infiltration observed in the liver of allogeneic recipients (Supplementary Fig. 24b).

In a challenging BALB/c mouse model of breast cancer lung metastasis, allogeneic MDC-735 derived from C57BL/6 mice were tested (Fig. 9e). Dose escalation demonstrated a clear dose-response relationship, with higher concentrations of HFt-735-loaded macrophages yielding great tumor reduction (Fig. 9f, g). The treatment was well tolerated as reflected by the weight of the mice (Supplementary Fig. 23f). Histopathological analysis confirmed normal architecture of critical organs, including the bone marrow, brain, heart, kidneys, liver, and lungs, with no signs of GvHD (Fig. 10a).

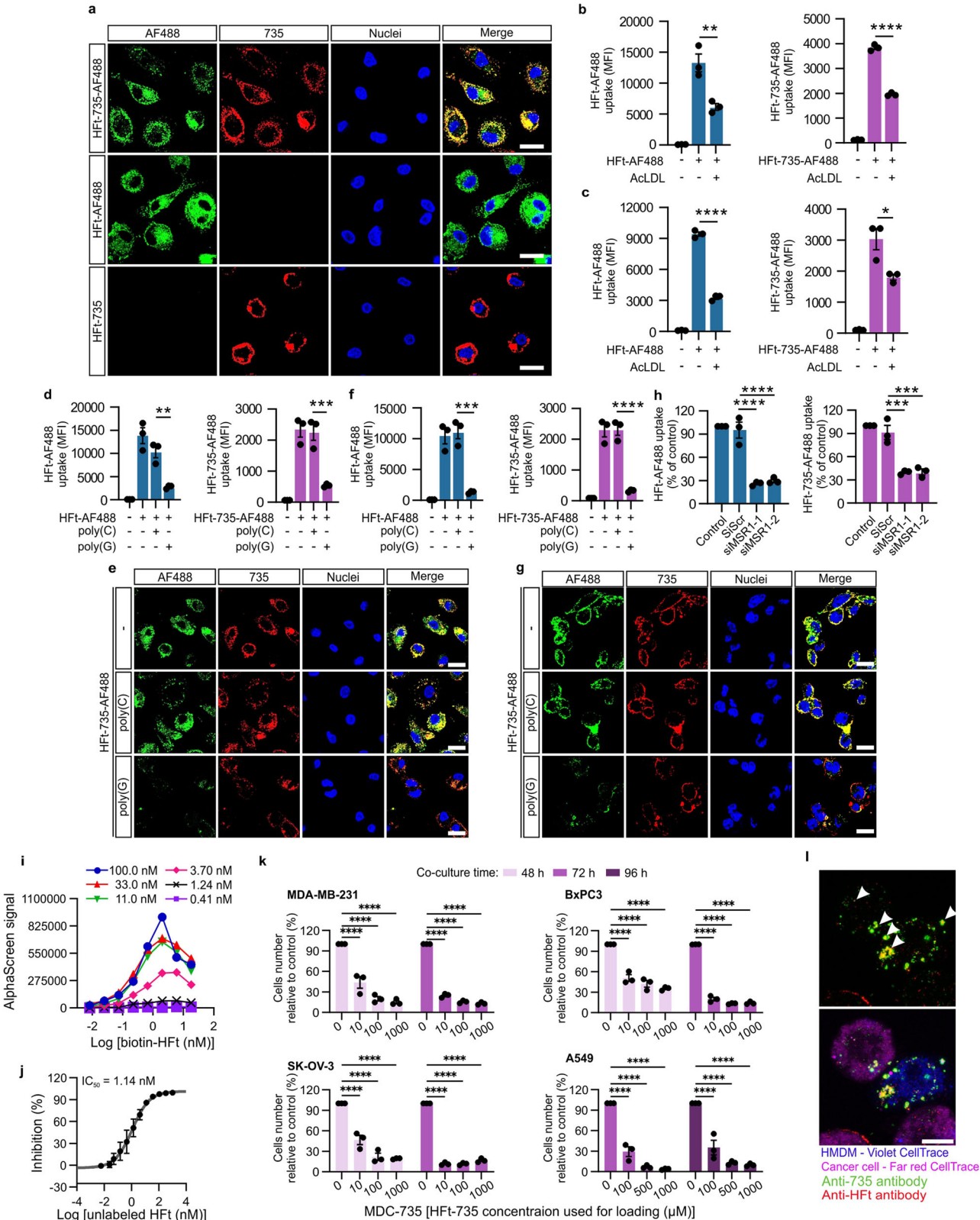

MDC-735 was also evaluated for its potential in combination therapies with anti-PD-1. In the MB49 bladder cancer model intratumoral injection of allogeneic mouse MDC-735, either alone or combined with anti-PD-1 checkpoint inhibitors, significantly improved survival and reduced tumor volume (Fig. 9h–l). The treatment was well tolerated, as reflected by the weight of the mice (Supplementary Fig. 23g). Similar results were seen in the SCC7 squamous cell carcinoma model, where allogeneic mouse MDC-735 enhanced survival and tumor control, especially when used with checkpoint inhibitors (Fig. 9m–q). The treatment was well tolerated, as reflected by the weight of the mice (Supplementary Fig. 23h). Histological analysis of the skin and macrophage-enriched organs (lungs, liver, and spleen) post-treatment showed no adverse effects, supporting the localized safety of intratumoral allogeneic MDC therapy (Fig. 10b).

**Fig. 7 | Human macrophages utilize the same receptor for HFt and HFt-drug complex uptake. a** Representative confocal microscopy images of colocalization of the 735 drug with HFt-735 in human macrophages. The hMDM were incubated with HFt-AF488 (50 µg/ml) or HFt-735-AF488 for 15 min at 37 °C, fixed, and stained with the anti-735 antibody and Hoechst 33342 (blue). Merged fluorescence images show colocalization of HFt and 735 drug (yellow foci). Scale bar = 20 µm. **b** and (**c**) Flow cytometry analysis of internalized HFt-AF488 and HFt-735-AF488 (5 µg/ml) by hMDM (**b**) or THP-1 macrophages (**c**) in the absence or presence of AcLDL within 30 min at 37 °C. Flow cytometry data are presented as mean fluorescence intensity (MFI) of HFt-AF488 or HFt-735-AF488. Data presented as mean ± SEM from $n = 3$ different donors (**b**, hMDM) or $n = 3$ independent replicates (**c**, THP-1). **d** Flow cytometry analysis of internalized HFt-AF488 and HFt-735-AF488 (50 µg/ml) within 30 min at 37 °C by hMDM untreated or pre-treated for 30 min at 37 °C with poly(G) binding to scavenger receptor and poly(C) (control). Flow cytometry data are presented as % of HFt-AF488 and HFt-735-AF488 uptake in untreated, control cells. Data presented as mean ± SEM from $n = 3$ different donors (hMDM). **e** Representative confocal microscopy images of internalized HFt-AF488 and HFt-735-AF488 (50 µg/ml) (green) within 30 min at 37 °C by hMDM untreated or pre-treated for 30 min at 37 °C ligand of class A scavenger receptor or structurally related ligand that does not bind to this group of receptors (negative control): poly(G) and poly(C) (control). Afterwards cells were fixed and stained with Hoechst 33342 (blue) and anti-735 drug antibody. Scale bar = 20 µm. **f** Flow cytometry analysis of internalized HFt-AF488 and HFt-735-AF488 (50 µg/ml) within 30 min at 37 °C by THP-1 cells untreated or pre-treated for 30 min at 37 °C with poly(G) binding to scavenger receptor and poly(C) (control). Flow cytometry data are presented as % of HFt-AF488 and HFt-735-AF488 uptake in untreated, control cells. Data presented as mean ± SEM from $n = 3$ (HFt-AF488) and $n = 3$ (HFt-735-AF488) independent replicates. **g** Representative confocal microscopy images of internalized HFt-AF488 and HFt-735-AF488 (50 µg/ml) (green) within 30 min at 37 °C by THP-1 cells untreated or pre-treated for 30 min at 37 °C ligand of class A scavenger receptor or structurally related ligand that does not bind to this group of receptors (negative control): poly(G) and poly(C) (control). Afterwards cells were fixed and stained with Hoechst 33342 (blue) and anti-735 drug antibody. Scale bar = 20 µm. **h** Flow cytometry analysis of internalized HFt-AF488 (50 µg/ml) and HFt-735-AF488 by hMDM after MSR1 gene-knockdown within 30 min at 37 °C. For comparison, untreated cells (Control) and cells treated with a negative, scramble control siRNA (siScr) were used. Flow cytometry data are presented as % of ligand uptake in control cells (Control). Data presented as mean ± SEM from $n = 3$ different donors (hMDM). **i** Optimization of AlphaScreen assay conditions by cross-titration of His-MSR1 (0.41–100 nM, 3-fold serial dilution) and Biotin-HFt (0.008–18 nM, 3-fold serial dilution) at fixed 10 µg/ml concentration of both nickel chelated acceptor and streptavidin donor beads. Data shown are from a single pilot run conducted for assay optimization. **j** Competitive effect of HFt-735 on AlphaScreen signal generated by His-MSR1 & Biotin-HFt interaction. Fixed concentrations of 8 nM His-MSR1 and 2 nM Biotin-HFt were used below the hook point of the bead assay (**i**) in the presence of increasing concentration of HFt-735 (0.006–1000 nM, 3-fold serial dilution). $IC_{50}$ potency of HFt-735 in displacing Biotin-HFt from interaction with His-MSR1 was calculated by fitting it to a four-parameter nonlinear regression. Data presented as mean ± SEM from $n = 3$ independent replicates. **k** In vitro evaluation of cancer cell killing by MDC-735 against MDA-MB-231, BxPC-3, SK-OV-3 and A549 cancer cells in co-culture. Macrophages were incubated with 10, 100, 1000 µmol of HFt-735 complex or plain medium (0). Data are presented as mean ± SEM of $n = 3$ independent replicates. **l** Representative confocal microscopy images captured after 24 h co-culture of MDC-735 (pre-labeled with CellTrace Violet CT) with SK-OV-3 cancer cells (pre-labeled with CellTrace Far Red). Cells fixed after 24 h additionally stained with anti-735 (green) and anti-HFt (red) antibodies. Co-localization of green and red signals both in macrophages and in cancer cells are pointed out by the arrowheads. Scale bar = 10 µM. The one-way ANOVA with Dunnett's post-hoc test was used for statistical analysis in panels **b–d**, **f**, **h**. The two-way ANOVA and Tukey's post-hoc tests were used for statistical analysis in panel **k**. For all panels, *$P \leq 0.05$, **$P \leq 0.01$, ***$P \leq 0.001$, ****$P \leq 0.0001$. Source data are provided as a Source Data file.

These studies underscore the efficacy and safety of the allogeneic macrophage therapy.

To address concerns about off-target effects, we conducted a series of in vitro studies to evaluate MDC-735's selectivity. Co-culture assays with various normal human cell types (including hepatocytes, dermal microvascular endothelial cells, bladder fibroblasts, renal epithelial cells, brain microvascular cells, and lung fibroblasts) showed that MDC-735 preferentially targets cancer cells over non-malignant cells (Fig. 10c, d). This selective cytotoxicity is consistent with our in vivo findings, suggesting that the TRAIN mechanism favors malignant cells, thereby minimizing the risk of damage to healthy tissues.

Collectively, these comprehensive in vivo and in vitro studies demonstrate the efficacy and safety of MDC-735 in targeting a range of solid tumors, including those with metastatic potential. The therapy leverages the completely novel TRAIN mechanism, enabling precise delivery of therapeutic agents directly to cancer cells while sparing healthy tissue. The ability of MDC-735 to act both as a monotherapy and in combination with standard treatments (e.g., checkpoint inhibitors) highlights its versatility and potential for clinical application in difficult-to-treat cancers. These results provide a strong foundation for advancing MDC therapy into clinical trials, aiming to deliver an allogeneic, off-the-shelf treatment option for patients with advanced solid tumors.

## Discussion

In this manuscript, we describe a novel and highly promising Macrophage-Drug Conjugate (MDC) platform for cancer therapy. Macrophages are gaining significant interest in the field of cell-based therapies, with an increasing number of companies developing autologous macrophages for the treatment of cancer or liver cirrhosis[45]. The strategy involving macrophage therapy is less developed, and to date, there has been limited progress in translating this approach into clinical applications, underscoring the need for further research in this area[46–48]. Our study directly addresses this gap by leveraging the unique physiological properties of macrophages to deliver therapeutic agents efficiently and with fewer off-target effects.

Cell-based therapies frequently involve time-consuming, costly, and complex manufacturing and logistics that make treatments less predictable and harder to scale up. The current reliance on autologous therapies also introduces significant variability, hampering reproducibility and scalability, which are critical for widespread clinical application. In contrast, our MDC platform employs an allogeneic approach, offering significant advantages over autologous therapies. Our technology leverages the ability of macrophages to internalize and transfer heavy-chain ferritin (HFt) proteins to cancer cells. This natural mechanism, termed TRAIN (TRAnsfer of Iron-binding proteiN), allows for targeted delivery of therapeutic agents directly to tumor cells, offering a more precise and effective treatment strategy compared to existing cell therapies. The discovery of iron-binding protein transfer provides new insights into the physiological properties of macrophages and their role in the tumor microenvironment. The goal of this therapy is to advance towards clinical translation as an allogeneic, on-demand treatment, with the potential to become an 'off-the-shelf' therapy in the future.

Although the physiological functions of extracellular ferritin are still unclear, it has previously been shown to be involved in iron delivery to a variety of cells. By binding to cognate receptors, such as TfR1[49], SCARA5[50], and TIM2[51,52], ferritin undergoes endocytosis and delivers iron to the intracellular compartments. Previous studies have shown that macrophages play a critical role in the development of erythroid cells by providing essential iron in the form of HFt, highlighting their relevance in iron metabolism[53–56]. We have now adapted this pathway for therapeutic purposes, utilizing the macrophage's natural mechanism to deliver ferritin-bound drugs selectively to cancer cells. In essence, we leverage the TRAIN mechanism, changing the passenger from iron to anticancer drugs, allowing precise delivery directly to tumor cells. This adaptation not only targets the high iron demand of cancer cells but also enhances drug

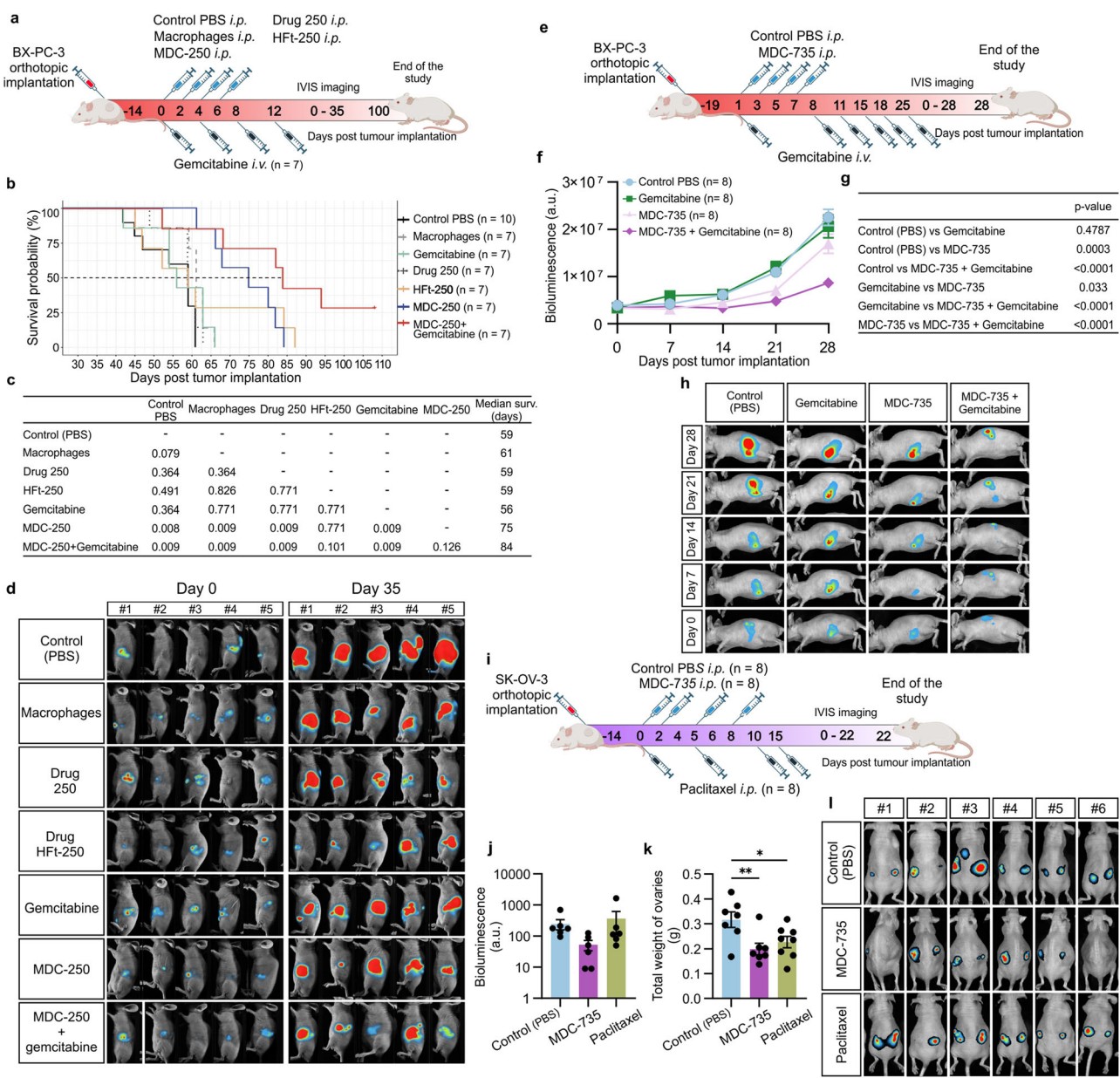

**Fig. 8 | Therapeutic efficacy of MDC against human pancreatic and ovarian cancer. a** Schematic representation of the treatment schedule employed in the in vivo MDC-250 pancreatic study: vehicle control group (PBS control), empty macrophage control group, free HFt-250 drug conjugate control group (at the dose ensuring equivalent 250 drug dose given as MDC-250), free drug control group (250, at the dose ensuring equivalent 250 drug dose given as MDC-250), standard of care group (gemcitabine), treatment group (MDC-250 administered intraperitoneally) and combination group (gemcitabine + MDC-250). Created in BioRender. Taciak, B. (2024) https://BioRender.com/t57p591. **b** Kaplan–Meier survival curve of mice treated with MDC-250 therapy compared to the control groups. **c** The statistical significance of the survival was analyzed using the pairwise log-rank tests. **d** Representative bioluminescence images depicting tumor sizes in mice from the control and MDC-250 treatment groups at various time points. **b**–**d** n = 7 mice per group (n = 10 mice in control group) (**e**) Schematic representation of the treatment schedule employed in the in vivo MDC-735 pancreatic study: vehicle control group (PBS control, intraperitoneal), standard of care group (gemcitabine), treatment group (MDC-735 administered intraperitoneally) and combination group (gemcitabine + MDC-735). Created in BioRender. Taciak, B. (2024) https://BioRender.com/g65f180. **f** Mean bioluminescence curve of mice treated with MDC-735 therapy

compared to the control groups. Data presented as mean ± SEM. **g** The statistical significance of the bioluminescence signal (end of the study at day 28) was analyzed using the two-way ANOVA with Tukey's multiple comparisons test.
**h** Representative bioluminescence images depicting tumor sizes in mice from the control and MDC-735 treatment groups measured during the study. **e**–**h** n = 8 mice per group. **i** Schematic representation of the treatment schedule employed in the in vivo MDC-735 ovarian study: vehicle control group (PBS control), standard of care group (paclitaxel), treatment group (MDC-735, administered intraperitoneally). Created in BioRender. Taciak, B. (2025) https://BioRender.com/j88l839.
**j** Mean total bioluminescence (percent of initial value) of SK-OV-3 tumors treated with MDC-735 therapy compared to the control groups at the end of the study. Data presented as mean ± SEM from n = 6 mice per group. **k** Mean and individual ovary weight (both included) from athymic nude mice orthotopically implanted with SK-OV-3 tumors treated with MDC-735 therapy compared to the control groups. Data presented as mean ± SEM from n = 7 mice in MDC-735 and PBS groups and n = 8 in the Paclitaxel group. **j** and (**k**) The one-way ANOVA with Dunnett's post-hoc test was used for statistical analysis, *P ≤ 0.05, **P ≤ 0.01. **l** Bioluminescence images depicting tumor sizes in mice from the control and MDC-735 treatment groups at the end of the study. Source data are provided as a Source Data file.

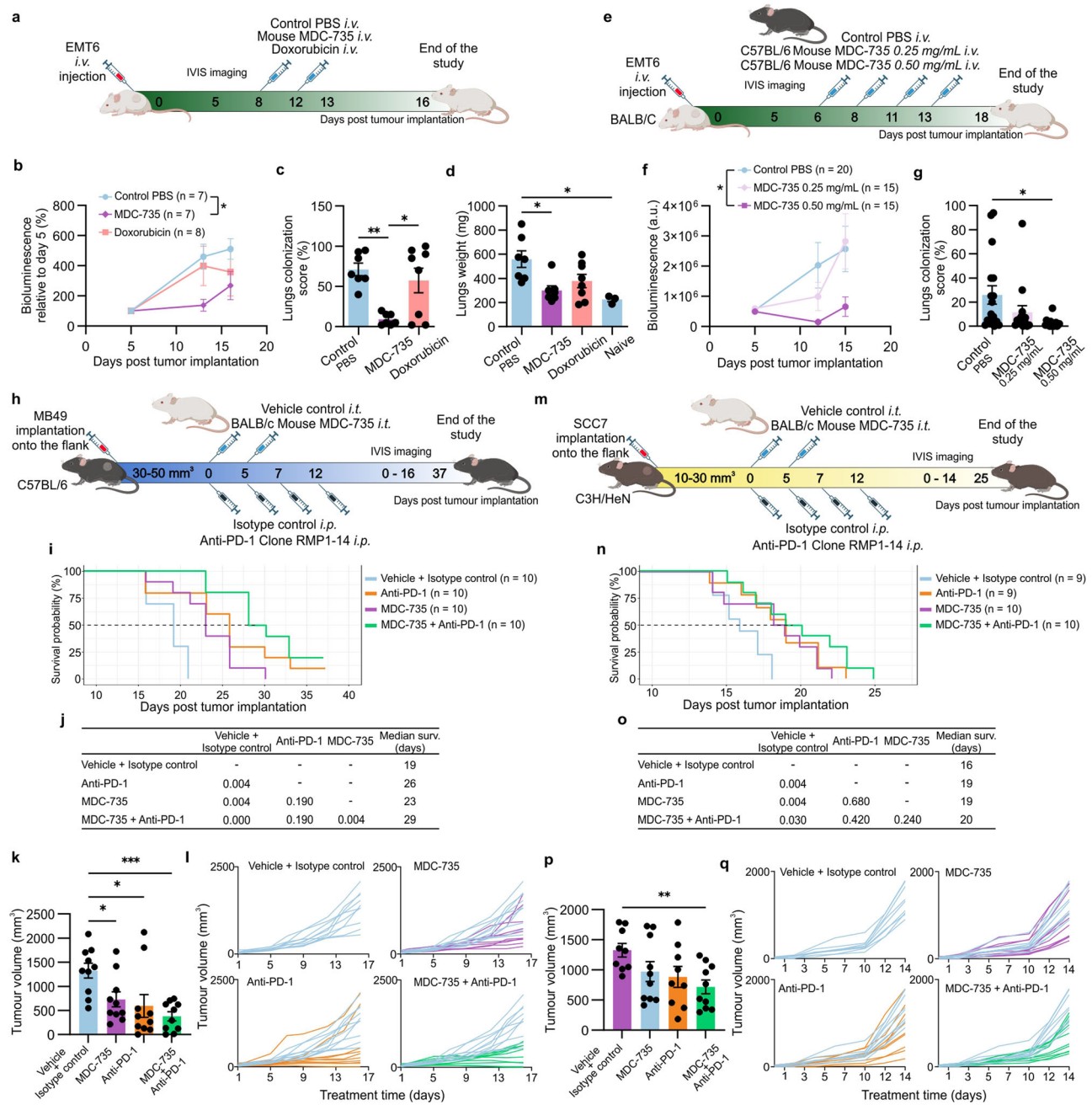

specificity, minimizing off-target effects and improving therapeutic outcomes.

Our study presents several key findings that advance our understanding of HFt uptake by human macrophages. We show that human macrophages internalize significantly more HFt compared to monocytes, primarily through clathrin-dependent endocytosis—a pathway consistent with HFt uptake observed in other cell types, including oligodendrocyte progenitors[57,58], lymphocytes[27,59,60], and hepatocytes[61]. This robust uptake mechanism positions macrophages as ideal delivery vehicles for therapeutic agents in cancer therapy. Once internalized, HFt is trafficked to both endosomes and lysosomes within macrophages, indicating a mechanism for controlled drug release in cancer cells. Lysosomal trafficking, in particular, presents a potential for the regulated release of drug payloads at the tumor site.

In addition to findings indicating partial involvement of micropinocytosis, we initially identified TfR1 as a receptor responsible for facilitating HFt uptake[62,63]. However, despite macrophages exhibiting

higher TfR1 expression than monocytes, knockdown experiments demonstrated that TfR1 contributes only partially to HFt uptake, implying the presence of additional receptors involved in this process. Our investigation of SR-A family members identified MSR1 as a key receptor that is highly expressed in human macrophages and plays a significant role in HFt internalization by these cells. This discovery highlights MSR1 as a crucial receptor in our MDC platform, mediating the uptake of drug-loaded HFt. Knockdown experiments further confirmed MSR1's substantial role in HFt uptake. The interaction between MSR1 and HFt, supported by additional binding affinity studies, underscores its function as a primary receptor for HFt in human macrophages, facilitating endocytosis and lysosomal trafficking. This receptor-ligand interaction not only enhances HFt uptake but also enables precise delivery of therapeutic agents via macrophages.

Surprisingly, we discovered that once macrophages take up HFt, they subsequently transfer it to cancer cells. We investigated this mechanism and found that, unlike macrophage-erythroid precursor

**Fig. 9 | Therapeutic efficacy of MDC against mouse breast metastasis to the lungs, bladder and head and neck cancer. a** Schematic representation of the treatment schedule employed in the in vivo mouse EMT6 breast cancer metastasis to the lungs study: vehicle control group (PBS control), standard of care group (Doxorubicin) and treatment group (autologous MDC-735) administered intravenously. Created in BioRender. Taciak, B. (2024) https://BioRender.com/m50e375. **b** Mean bioluminescence curve of mice treated with MDC-735 therapy compared to the control groups. **c** Effect of mouse MDC-735 and doxorubicin on lung colonization in an EMT6 breast cancer mouse model. **d** Effect of MDC-735 (generated from BMDM) and doxorubicin on lung weight in EMT6 breast cancer mouse model. **b–d** Data are presented as mean ± SEM from $n = 7$ mice in MDC-735 and PBS, $n = 8$ in Doxorubicin group. **d** Additionally $n = 3$ Naïve mice were used as a control. **e** Schematic representation of the treatment schedule employed in the in vivo EMT6 breast metastasis to the lungs study: vehicle control group (PBS control) and treatment groups (allogeneic MDC-735 loaded with HFt-735 at 0.25 mg/ml and 0.50 mg/ml). Created in BioRender. Taciak, B. (2024) https://BioRender.com/p63h677. **f** Mean bioluminescence curve of mice treated with MDC-735 therapy compared to the control group. **g** Effect of allogeneic mouse MDC-735 on lung colonization in an EMT6 breast cancer mouse model. **f** and (**g**) Data are presented as mean ± SEM from $n = 17$ mice in PBS control group and $n = 15$ in each of the MDC-735 group. **h** Schematic representation of the treatment schedule employed in the in vivo mouse MB49 bladder cancer study: vehicle control group, isotype control group, Anti-PD-1 antibody, treatment group (allogeneic MDC-735) and combination

group (MDC-735 + Anti-PD-1 antibody). Created in BioRender. Taciak, B. (2024) https://BioRender.com/z27l686. **i** Kaplan–Meier survival curve of mice treated with MDC-735 therapy compared to the control groups. **j** The statistical significance of the survival was analyzed using the pairwise log-rank tests. **k** Effect of MDC-735 and Anti-PD-1 on tumor volume in an MB49 bladder cancer mouse model. Data are presented as mean ± SEM. **l** Tumor volume progression over time with different treatments of MB49 bladder cancer mouse model. **k** and (**l**) $n = 10$ mice per group. **m** Schematic representation of the treatment schedule employed in the in vivo mouse SCC7 squamous cell carcinoma study: vehicle control group, isotype control group, Anti-PD-1 antibody, treatment group (allogeneic MDC-735), and combination group (MDC-735 + Anti-PD-1 antibody). Created in BioRender. Taciak, B. (2024) https://BioRender.com/m43v541. **n** Kaplan–Meier survival curve of mice treated with MDC-735 therapy compared to the control groups. **o** The statistical significance of the survival was analyzed using the pairwise log-rank tests. **p** Effect of MDC-735 and Anti-PD-1 on tumor volume in a SCC7 squamous cell carcinoma mouse model. Data are presented as mean ± SEM. **q** Tumor volume progression over time with different treatments of SCC7 squamous cell carcinoma mouse model. **p** and (**q**) $n = 9$ mice in Vehicle + Isotype control and Anti-PD-1 groups and $n = 10$ in MDC-735 and MDC-735 + Anti-PD-1 groups. The one-way ANOVA with Tukey's post-hoc test was used for statistical analysis in (**c, d, g**), Welch's $t$-test with Bonferroni multiple comparison correction in (**b, f, p**), one-way ANOVA with Dunnett's post-hoc test in (**k**). For all panels $*P \leq 0.05$, $**P \leq 0.01$, $***P \leq 0.001$. Source data are provided as a Source Data file.

interactions, where macrophages secrete HFt by exocytosis[53–55], only a small fraction of HFt is released into the medium. This crucial finding suggests that direct cell-to-cell contact between macrophages and cancer cells is the primary mechanism of drug transfer. The involvement of vesicular pathways is strongly supported, as inhibitors of vesicular trafficking or clathrin knockdown significantly reduce the efficiency of HFt transfer. These results indicate that an immune synapse-like interface may facilitate this transfer, with F-actin and ICAM-1 accumulation at the interaction site. This mirrors interactions seen in virological synapses, particularly in HIV-1 transmission[64–66]. Our study suggests that the TRAIN mechanism for HFt transfer from macrophages to cancer cells shares similarities with viral synapse formation, potentially revealing new therapeutic targets in the tumor microenvironment.

Our research underscores the extensive versatility and scalability of the MDC platform, validated through rigorous testing across diverse macrophage sources, including human and mouse primary cells, HiPSC-derived macrophages, and macrophage cell lines. These sources consistently demonstrated HFt uptake and transfer capabilities, affirming the platform's broad applicability and robustness, particularly for allogeneic applications.

We encapsulated 20 distinct anticancer drugs within the HFt cavity, achieving significant efficacy in both in vitro and in vivo studies. MDC-735, our leading candidate, demonstrated remarkable tumor reduction and metastasis prevention across multiple cancer models, including ovarian (SK-OV-3) and pancreatic cancer (BxPC-3-luc). For instance, in the SK-OV-3 ovarian cancer model, MDC-735 not only controlled tumor growth but also inhibited metastasis to distant organs—achievements not observed by conventional therapies. Similarly, MDC-250 extended overall survival in pancreatic cancer models while maintaining an excellent safety profile, with no significant weight loss in treated mice. Combining MDC-250 with standard chemotherapy agents, such as gemcitabine, further enhanced therapeutic outcomes, suggesting strong potential for synergistic use with existing treatments.

In metastatic breast cancer models, both autologous and allogeneic MDC-735 significantly reduced lung metastases, demonstrating the platform's utility in treating metastatic disease. Allogeneic MDC-735, used independently or in combination with anti-PD-1 checkpoint inhibitors, achieved substantial tumor volume reduction and survival improvement in the MB49 bladder cancer model, with similar efficacy observed in the SCC7 squamous cell carcinoma model.

The preclinical safety profile of MDC therapy strongly supports its clinical potential, particularly in an allogeneic context. Our studies confirmed that allogeneic administration of MDC is well-tolerated, with no significant adverse effects or signs of graft-versus-host disease (GvHD) observed in treated mice. Comprehensive histopathological analysis of critical organs—including the liver, heart, kidney, brain, femur, and lung—revealed no abnormalities or inflammation. These findings align with earlier clinical evidence of the safe use of allogeneic macrophages in patients with limb ischemia and cerebral palsy[67,68], further validating our approach.

A key advantage of the MDC platform is its ability to deliver therapeutic effects at lower drug doses, minimizing systemic toxicity. Treatment with MDC-250 showed significant therapeutic outcomes even at doses where the free drug alone had no effect. In co-culture experiments, MDCs exhibited selective cytotoxicity against cancer cells while sparing healthy tissues, reducing off-target effects, and enhancing suitability for advanced or refractory cancers. This approach contrasts with other delivery systems that rely on the cell's death for drug release or the uncontrolled diffusion of drugs from exosomes[69].

Additionally, MDCs retain their functional activity and phenotype after long-term cryopreservation, demonstrating the stability required for scalable "off-the-shelf" therapies. This durability, combined with the ability to utilize healthy donor-derived or HiPSC-derived macrophages, addresses critical manufacturing challenges faced by other cell-based treatments[70].

In summary, the MDC platform exemplifies a transformative allogeneic, off-the-shelf therapeutic approach designed for rapid deployment, cost-effectiveness, and compatibility with combination treatments[71]. With Phase I clinical trials planned for 2025, this innovative technology offers a compelling solution for unmet needs in oncology, particularly for patients with advanced or refractory cancers.

## Methods

### Cell lines

Human leukemia monocytic cell line THP-1 (ATCC TIB-202), human breast cancer cell line MDA-MB-231 (ATCC HTB-26), human colon cancer cell line LoVo (ATCC CCL-229), human embryonic kidney cell line HEK-293 (ATCC CRL-1573), Chinese hamster ovary cell line CHO-K1 (ATCC CCL-61), human glioblastoma cell line U-87 MG (ATCC HTB-14), murine macrophage cell line RAW 264.7 (ATCC TIB-71), murine mammary cancer cell lines EMT6 (ATCC CRL-2755), EMT6-Fluc-Puro

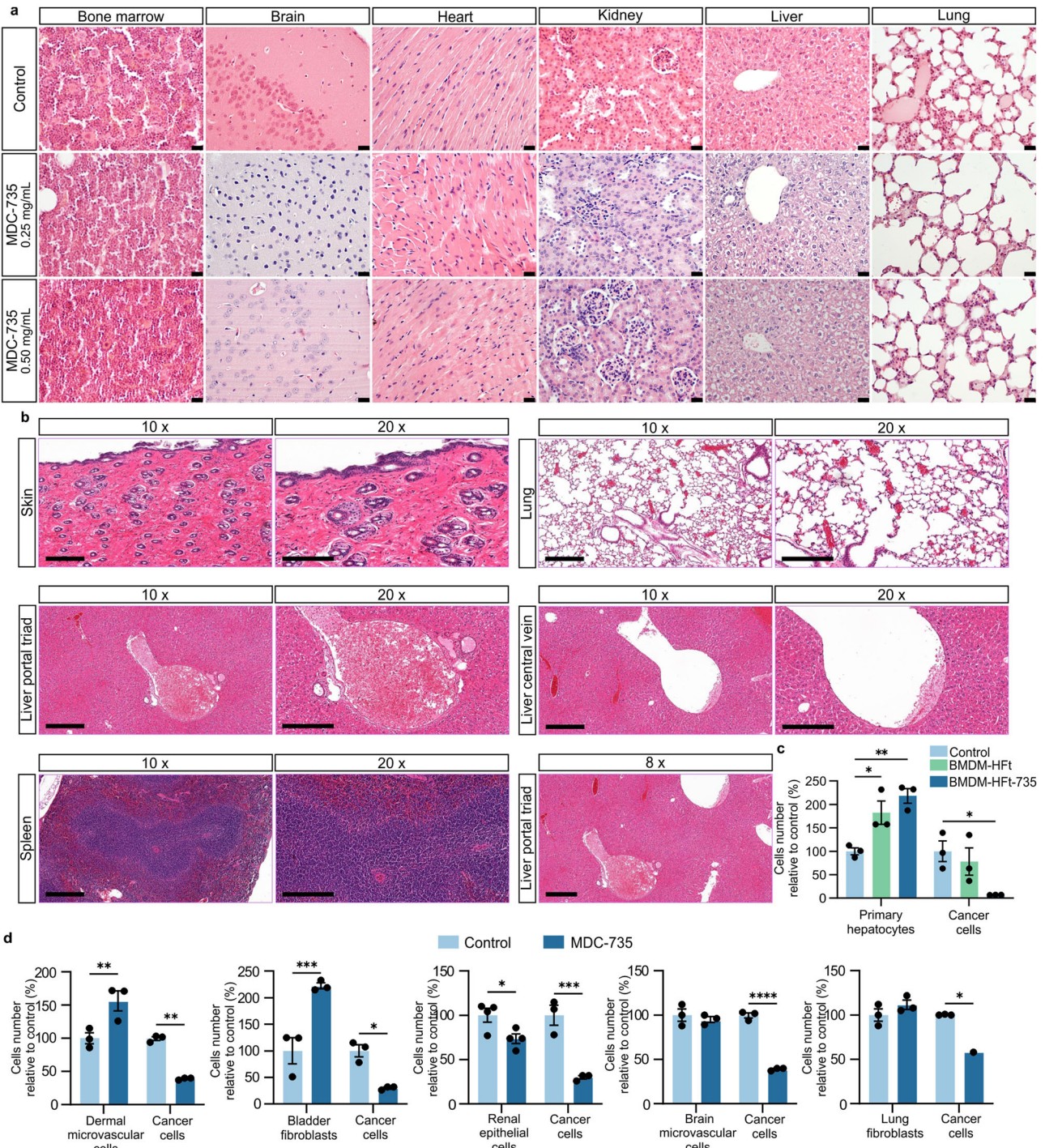

**Fig. 10 | Treatment with MDC-735 therapy does not affect organ anatomy or normal cells. a** Histopathological examination of different types of tissue stained with Hematoxylin and Eosin (H&E) in mice with breast metastasis to the lungs (EMT6) treated with allogeneic mouse MDC-735: vehicle control group (PBS control) and treatment groups (MDC-735 loaded with HFt-735 at 0.25 mg/ml and 0.50 mg/ml). Representative images of *n* = 4 samples in PBS and *n* = 3 samples per MDC-735 group. Scale bar = 20 μm. **b** Histopathological examination of different types of tissue stained with Hematoxylin and Eosin (H&E) in mice dosed subcutaneously with MDC-735 (*n* = 3). 28 days later animals were euthanized, and liver, lung, spleen, and skin tissue resected and processed to FFPE, 4 μM sections were sectioned and H&E stained. Scale bar = 300 μm (8x), 350 μm (10x) or 250 μm (20x). **c** and (**d**) Comparison of normal cells vs. cancer cell line viability in co-culture with MDC-735. Bar plots show the viability of various types of normal cells and the EMT6 cancer cell (**c**) and the LN-229 cancer cell line (**d**) after 48-h co-culture with MDC-735. Viability was measured as the number of live cells relative to the control condition (no macrophages). Data represents the mean ± SEM from *n* = 3 independent replicates. Statistical analysis was performed using one-way ANOVA and post hoc Tukey HSD test; *P ≤ 0.05, **P ≤ 0.01, ***P ≤ 0.001, ****P ≤ 0.001. Source data are provided as a Source Data file.

(Imanis, CL154), and 4T1 (ATTC CRL-2539) were obtained from the American Type Culture Collection (ATCC; www.atcc.org).

Murine colon cancer cell line CT26.WT (ATCC CRL-2638), human cervical cancer cell line HeLa (ATCC CCL-2), human ovarian cancer cell line SK-OV-3 (ATCC HTB-77), human breast cancer cell line MCF7 (ATCC HTB-22), human pancreatic cancer cells line HPAC (ATCC CRL-2119), human colon cancer cell line HCT 116 (ATCC CCL-247), human prostate cancer cell line DU 145 (ATCC HTB-81) and murine connective

mouse tissue L929 cell line (ATCC CCL-1) were kindly provided by Medical University of Warsaw. Murine mouse bladder carcinoma MB49 and mouse head and neck carcinoma SCC7 cell lines were provided by Axis Bio Services Ltd. for in vivo studies. Human colon cancer cell lines Caco2 (ATCC HTB-37), CL-11 (DSMZ ACC 467), CL-40 (DSMZ ACC 535), Colo205 (ATCC CCL-222), DLD-1 (ATCC CCL-221), HT-29 (ATCC HTB-38), SW480 (ATCC CCL-228), human prostate cancer cell line LNCaP (ATCC CRL-1740), human lung cancer cell line A549 (ATCC CCL-185), human chondrosarcoma cell line SW 1353 (ATCC HTB-94), human pancreas cancer cell line BxPC-3 (ATCC CRL-1687) were kindly provided by the Maria Skłodowska Curie Memorial Cancer Centre and Institute of Oncology (MCMCC) in Warsaw. The mouse glioma cell line GL261 was obtained from the Division of Cancer Treatment and Diagnosis (DCTD) Tumor Repository (National Cancer Institute, Warsaw).

Primary human bladder fibroblast cells (ATCC PCS-420-013) were cultured in FibroLife S2 Fibroblast Medium (Lifeline, LL-0011), primary human renal proximal tubule epithelial cells (ATCC, PCS-400-010) in RenaLife Epithelial Medium (Lifeline, LL-0025), human brain microvascular endothelial cells (HBMVEC, iXCells, 10HU-051) in VascuLife VEGF Endothelial Medium Complete Kit (Lifeline, LL-0003), and human lung fibroblast MRC-5 cells (ATCC, CCL-171) in Eagle's Minimum Essential Medium (EMEM, ATCC, 302003).

All cells utilized in this manuscript were cultured at 37 °C in a humidified incubator with a 5% $CO_2$ atmosphere. THP-1 and EMT6 cells were cultured in Roswell Park Memorial Institute 1640 medium (RPMI-1640, Sigma-Aldrich, R8758) supplemented with 10% Fetal Bovine Serum (FBS, HyClone, SV30160.03) and penicillin/streptomycin (100 U/ml, Gibco, 15140122). CHO-K1 cells were cultured in Ham's F-12K medium (Gibco, 21127022) supplemented with 10% Bovine Calf Serum (BCS, HyClone, SH30072.04) and penicillin/streptomycin (100 U/ml, Gibco, 15140122). All remaining cell lines were maintained in high glucose Dulbecco's modified Eagle's medium (DMEM, Gibco, 41965062) supplemented with 10% FBS (HyClone, SV30160.03) and 100 U/mL penicillin/streptomycin (100 U/ml, Gibco, 15140122).

### Human leukocytes and monocyte-derived macrophages
Human monocyte-derived macrophages (hMDM) were obtained from buffy coats commercially available to purchase from Regional Blood and Hemotherapy Centre in Warsaw for research purposes. The buffy coats are provided anonymously in line with Polish legislation section code §13 Dz.U.1997.106.681 and could not be traced back to a specific individual. Polish legislation is in accordance with Directive 2002/98/EC of The European Parliament and of The Council of 27 January 2003, setting standards of quality and safety for the collection, testing, processing, storage, and distribution of human blood and blood components. In line with the code, The Committee on the Ethics of Research Involving Human Involvement at Warsaw University of Life Sciences has been informed about their use for research. Human peripheral blood mononuclear cells (PBMCs) were isolated by density gradient centrifugation (30 min, 21 °C, 400 × g, w/o acceleration and brake) using Histopaque-1077 (Sigma-Aldrich, 10771). CD4 and CD8 lymphocytes were selected using EasySep Human CD4/CD8 T Cell Isolation Kits (STEMCELL, 17952 and 17953) and used for the experiment directly after isolation. Monocytes were collected using Pan Monocyte Isolation Kit (Miltenyi Biotec, 130-096-537), seeded in sterile petri dishes (∅ 90 mm, Promed, FL Medical, 29262, 6 × 10^6 cells/plate), and cultured over 5 days in RPMI-1640 medium (Gibco, 11875085) containing Glutamax (2 mM, Gibco, 35050038), 20% FBS (HyClone, SV30160.03), penicillin/streptomycin (100 U/ml, Gibco, 15140122) and 50 ng/ml M-CSF (BioLegend, 574808). On day 5 medium was replaced and supplemented with M-CSF (50 ng/ml) and various stimulatory agents such as LPS (10 ng/ml, InvivoGen, TLRL-PEKLP), IFN-γ (50 ng/ml, PeproTech, 300-02), IL-4 (20 ng/ml, PeproTech, 200-04) or IL-13 (20 ng/ml, PeproTech, 200-13) for 48 h as described in the

main text. For large-scale production of hMDM used in in vivo studies, PBMCs were collected via leukapheresis from healthy volunteers at the Institute of Hematology and Transfusion Medicine, Warsaw. The leukapheresis procedure was carried out using the Spectra Optia system (Terumo) by MTZ Clinical Research Sp. z o.o., following ethical approval from the Bioethics Committee of the Regional Medical Chamber in Warsaw (study no. KB/1359/21). Monocytes were isolated through positive selection with CliniMACS CD14 microbeads (Miltenyi Biotec, 200-070-118), utilizing the CliniMACS Prodigy system (Miltenyi Biotec), and cultured in MACS GMP Cell Differentiation Bags (Miltenyi Biotec, 170-076-401) using TexMACS Medium (Miltenyi Biotec, 130-097-196) supplemented with penicillin/streptomycin (100 U/ml, Gibco, 15140122) and 100 ng/ml MACS GMP Recombinant Human M-CSF (Miltenyi Biotec, 170-076-172).

### Human iPSC line maintenance
Human induced pluripotent stem-cell (iPSC) line SFCi55, was generated in-house from dermal fibroblasts that were obtained from blood group O Rhesus negative donors under REC 1/AL/0020 ethical approval[72]. The iPSC cell line has been registered, the hPSCreg Name Edi020-A and alternative name(s) SFCi55 (https://hpscreg.eu/cell-line/EDi020-A). iPSCs were maintained in StemPro hESC SFM medium (Gibco, A1000701) containing 20 ng/ml human basic FGF (Gibco, PHG0021). Medium was changed every day and cells were passaged when 70% confluent (every 3–4 days) using the StemPro EZPassage tool (Gibco, 23181010) and cultured on six-well plates (Falcon, 353046), pre-coated for 1 h with CTS CELLstart substrate (Gibco, A1014201).

### Generation of human iPSC-derived macrophages
Previously published differentiation protocols were adapted for macrophage production[73,74]. iPSCs were harvested for embryoid body (EB) formation by incubating the cells for 5 min at 37 °C with 1 ml of TrypLE Express (Gibco, 12604021). Cells were then washed with phosphate buffer saline (PBS, Biowest, L06151000) and resuspended at a final concentration of $5 × 10^4$ cells/ml in EB-medium consisting of StemPro hESC SFM medium (Gibco, A1000701) supplemented with BMP4 (50 ng/ml, R&D Systems, 314-BP-010), VEGF (50 ng/ml, R&D Systems, 293-VE-010), SCF (20 ng/ml, Gibco, PHC2111), additionally supplemented with 10 μM ROCK Inhibitor (Y-27632, Sigma-Aldrich, SCM075). The cell suspension was transferred to a 96-well U-bottom non-adherent plate (100 μl/well, Falcon, 351177), centrifuged at 100×g for 3 min, and incubated for 4 days. On day 2, 50 μl of the medium was aspirated, and 100 μl of fresh EB medium was added to the wells. EBs were transferred to a six-well tissue-culture-grade plate (Falcon, 353046) pre-coated with 0.1% gelatin (Sigma-Aldrich, G7765) at a density of 10–15 EBs/well in 3 ml of X-VIVO 15 medium (Lonza, BE02-060F) supplemented with M-CSF (100 ng/ml, BioLegend, 574808), IL-3 (25 ng/ml, PeproTech, 200-13), penicillin/streptomycin (100 U/ml, Gibco, 15140122), Glutamax (2 mM, Gibco, 35050038) and β-mercaptoethanol (0.055 mM, Sigma-Aldrich, M3148). Medium was changed every 3–4 days. After 3 weeks, non-adherent monocyte-like precursors were harvested from the supernatant every 3–4 days for up to 2 months and seeded on six-well plates in X-VIVO 15 media (Lonza, BE02-060F) supplemented with M-CSF (100 ng/ml, BioLegend, 574808), Glutamax (2 mM, Gibco, 35050038) and penicillin/streptomycin (100 U/ml, Gibco, 15140122) for 9 days.

### THP-1 macrophages
THP-1 monocytes were differentiated into macrophages by 72 h incubation in RPMI-1640 medium (Sigma-Aldrich, R8758) containing 10% FBS (HyClone, SV30160.03), penicillin/streptomycin (100 U/ml, Gibco, 15140122) and 100 nM phorbol-12-myristate-13-acetate (PMA, Sigma-Aldrich, P1585). Then macrophages were either left untreated for 24 h (in PMA-free medium) to rest or were polarized to M1 and M2

activation states by stimulation with LPS (10 ng/ml, InvivoGen, TLRL-PEKLPS) and IFN-γ (50 ng/ml, PeproTech, 300-02) or with IL-4 (20 ng/ml, PeproTech, 200-04) and IL-13 (20 ng/ml, PeproTech, 200-13) for additional 48 h, respectively.

## Mouse bone-marrow-derived macrophages

L929 cell line was cultured in DMEM/F12 (Gibco, 31331093) supplemented with 10% BCS (HyClone, SH30072.04) and penicillin/streptomycin (100 U/ml, Gibco, 15140122) for 7 days. The conditioned medium was then collected, filtered through a 0.45 μm syringe filter (Sigma-Aldrich, CLS431225), and stored at −80 °C for use as a source of M-CSF. The bone marrow cells were harvested from femurs of BALB/c mice (unless specified otherwise) as described by Smith et al.[75] and were seeded into Petri dishes (Ø 90 mm, Promed, FL Medical, 29262) at a density of $2 \times 10^6$ cells per plate in 5 ml of culture medium. The medium used was DMEM/F12 (Gibco, 31331093) supplemented with 10% BCS (HyClone, SH30072.04), penicillin/streptomycin (100 U/ml, Gibco, 15140122), and 20% of L929 conditioned medium. Three days after seeding, an additional 5 ml of culture medium was added. After two days, the medium was replaced with 10 ml of fresh culture medium supplemented with IL-4 (20 ng/ml, BioLegend, 574306) or IFN-γ (50 ng/ml, PeproTech, 300-02) and LPS (10 ng/ml, InvivoGen, TLRL-PEKLPS) or without any additional cytokines and the cells were cultured for another 2 days.

## Primary murine hepatocytes

Primary hepatocytes were isolated from female BALB/c mice using a modified two-step perfusion method. Briefly, mice were euthanized, and their livers perfused via the inferior vena cava at 5 ml/min for 15 min with Liver Perfusion Medium (Gibco, 17701038) following portal vein transection. The medium was then replaced with pre-warmed (37 °C) Liver Digest Medium (Gibco, 17703034), and perfusion continued for another 15 min. Digested livers were disaggregated in Hepatocyte Wash Medium (Gibco, 17704024) and filtered through a 100-μm cell strainer (Falcon, 352360). Cells were further purified by centrifugation on a 1.06 g/ml Percoll gradient (Sigma-Aldrich, P4937) at $750 \times g$ for 20 min (no brake). Non-parenchymal cells (NPCs) were washed twice at $50 \times g$ for 3 min, then centrifuged at $650 \times g$ for 15 min at 4 °C. The pellet was resuspended in RBC Lysis Buffer (BioLegend, 420301), incubated for 3 min, washed, and centrifuged again. NPCs were resuspended in Williams E Medium (Sigma-Aldrich, W1878) with 10% FBS (HyClone, SV30160.03), penicillin/streptomycin (100 U/ml, Gibco, 15140122), 1% Glutamax (Gibco, 35050038), and 10 μg/ml insulin (Sigma-Aldrich, I9278). Cells were seeded onto collagen I-coated (Corning, 354236) plates at 30,000 cells/cm². After 3 h, the medium was replaced with Williams E Medium (Sigma-Aldrich, W1878) containing 4% FBS (HyClone, SV30160.03), penicillin/streptomycin (100 U/ml, Gibco, 15140122), and 1% Glutamax (Gibco, 35050038).

## Ferritin technology

Cloning, expression, and purification of the human heavy chain ferritin (HFt; NCBI Reference Sequence: NP_002023.2) were provided by Genscript (NJ, USA). The synthetic gene encoding of HFt was optimized for *Escherichia coli* codon usage, and cloned in a pET30a vector and overexpressed in *E. coli* BL21 cells. The cultures were induced with 0.5 mM IPTG (Isopropyl-β-D-1-thiogalactopyranoside, Sigma-Aldrich, I6758) and grown for 16 h at 25 °C before centrifugation. The cells were resuspended in 20 mM HEPES pH 7.5 (Sigma-Aldrich, H0887) with 300 mM NaCl (Sigma-Aldrich, S9888) and disrupted in a homogenizer (PandaPLUS 2000, GEA Niro Soavi) 3 rounds on ice. After centrifugation at $10,000 \times g$ at 4 °C for 30 min, the supernatant was subjected to gradual purification by 2 h incubation with 30% ammonium sulfate (Sigma-Aldrich, A4915) dialysis for 48 h against deionized water. As a final step, the protein was loaded onto a 16/600 Superdex 200 column (Cytiva, GE28-9893-35) using an NGC chromatography system (Bio-

Rad), the main fraction was collected, and the purity of the product was assessed via SDS−PAGE. The concentration of purified protein was calculated by measuring the UV−Vis spectroscopy using an extinction coefficient of $18,600 \, M^{-1} \, cm^{-1}$.

For the purpose of fluorescent labeling, HFt was conjugated to Alexa Fluor 488 NHS Ester (Succinimidyl Ester, Invitrogen, A20000) following the manufacturer's instructions. In the first step, the volume of protein solution corresponding to 10 mg HFt was transferred to a light-protecting tube and buffered with 1 M sodium bicarbonate (pH 8.3, Sigma-Aldrich, S5761). Then, 1 mg of fluorescent dye dissolved in DMSO (Sigma-Aldrich, D2650) was added to the buffered HFt solution and the reaction tube was incubated for 1 h at 25 °C under continuous shaking (500 rpm). In the next step, the solution was centrifuged at $10,000 \times g$ to remove any precipitate. Unconjugated fluorescent dye was removed by multiple washes of the supernatant on Amicon Ultra-4 Centrifugal Filter Unit (10 kDa cutoff, Merck, UFC801024) with PBS (Biowest, L06151000). The concentration of the HFt-AF488 and the level of fluorochrome labeling were determined by UV−Vis spectrophotometry.

HFt complexes were obtained as follows. The encapsulations of the drugs inside the HFt cages were performed according to previously described protocols[18,23]. Briefly, a solution of 0.1 mM HFt and 0.9 mM drug (doxorubicin, MedChem Express, HY-15142; drug 250, MedChem Express) was made in PBS (Biowest, L0615) with 10% DMSO (Sigma-Aldrich, D2650), pH 7.4. Gradually, with magnetic stirring, the pH was decreased to a value of 3.0 by the gradual addition of 1 M HCl (Supelco, 1003171011). After 20 min at RT incubation, the pH of the solution was increased again to 7.4 by the gradual addition of 1 M NaOH (Supelco, 1064981000). The sample was incubated for 45 min at RT with mixing and finally dialyzed against PBS (Biowest, L0615) at 4 °C overnight and centrifuged for 10 min at 4 °C at $10,000 \times g$, the supernatant was retained. The protein concentration in the complex was calculated based on the measured absorbance at 280 nm using a spectrophotometer (DeNovix DS-11 FX+) with an extinction coefficient of $18,600 \, M^{-1} \, cm^{-1}$. HFt cage formation was verified by native electrophoresis performed on 4−15% gradient polyacrylamide gels (Bio-Rad, 4561083). Samples were loaded on the gel at 20 μg. Electrophoresis was performed at a constant 100 V for 90 min. Dynamic light scattering (DLS) instrument measures the size of particles contained within a sample using protein model analysis with scanning time 10 s per one run duration (automatic number of runs: 10−20). Volume of the prepared sample was 1 ml and the concentration was 0.6 mg/ml. Prepared sample was filtered with 0.2 μm sterile syringe filters (Nalgene, Thermo Scientific, 723-2520). RP-HPLC was performed on Eurospher 100-5 C18 column (250 mm × 4 mm, Knauer, 25DE181ESJ) at 25 °C with solvent A (water with 1% orthophosphoric acid, Supelco, 49685) and solvent B (acetonitrile, VWR, 20060.420 with 1% orthophosphoric acid). Flow rate 1 ml/min. Injection was performed at 90% solvent A, then linear gradient B from 10% to 90% in 15 min (15−30 min), then 90% solvent A in 15 min. Injection on column 10 μl. The separated sample was analyzed with a UV detector at 215 nm.

The conjugation was performed with a five-fold excess of drug over protein. Drug 735, obtained from a commercial supplier (MedChem Express), was dissolved in a polar solvent before being conjugated with HFt in an aqueous buffer at neutral pH. The successful conjugation of 735 to HFt was confirmed using ultra-pressure liquid chromatography-mass spectrometry (UPLC-MS), which identified fractions of single and double-labeled conjugates and free protein. The observed mass increase, corresponding to the attachment of drug 735 to HFt, confirmed successful conjugation. The efficiency of the conjugation was calculated by UV-Vis spectrophotometry, using extinction coefficients of 19425 for HFt at 280 nm and 13642 for 735 drug at 252 nm. DLS analysis demonstrated the hydrodynamic diameter of the HFt-735 conjugate (15.51 ± 4.82 nm), indicating no significant aggregation and preservation of the ferritin structure. Native PAGE

confirmed the stable formation of HFt 24-mer cages following the conjugation process. Furthermore, the stability of the conjugate was evaluated, showing that conjugation efficiency remained consistent over six months of storage at −80 °C.

### Ferritin uptake assays

Monocytes and macrophages were incubated with fluorochrome-labeled HFt in serum-free RPMI 1640 (Sigma-Aldrich, R8758) at the indicated concentrations, time, and temperature. Subsequently, cells were harvested for flow cytometry analysis or fixed with 4% paraformaldehyde (PFA, Thermo Scientific, 28908) for immunostaining and confocal microscopy analysis. Competitive inhibition assays were performed by simultaneous incubation of macrophages with HFt-AF488 and an excess of one of the unlabeled ligands: HFt or AcLDL (Invitrogen, L35354) in serum-free RPMI-1640 (Sigma-Aldrich, R8758) for 30 min at 37 °C. HFt-AF488 uptake was then measured using flow cytometry (FACSCanto II, BD Biosciences) and confocal microscopy (Leica TCS SP5II, Leica Microsystems). In order to characterize the role of clathrin-mediated endocytosis in HFt uptake, macrophages were pretreated with different concentrations of chlorpromazine (Sigma-Aldrich, C8138), dynasore (Sigma-Aldrich, D7693), EIPA (MedChem Express, HY-101840), Rottlerin (Abcam, AB120377), Nystatin (Med-Chem Express, HY-17409), Genistein (Sigma-Aldrich, G6649), ML141 (MedChem Express, HY-12755) and a vehicle solution (DMSO, Sigma-Aldrich, D2650) in serum-free RPMI-1640 (Sigma-Aldrich, R8758) for 30 or 15 min, respectively, at 37 °C prior to addition of the ligand. Subsequently, HFt-AF488, transferrin conjugated with Alexa Fluor 488 (Tfn-AF488, Invitrogen, T13342), Dextran conjugated with Fluorescein (Invitrogen, D1822) or Alexa Fluor 647 (Invitrogen, D22914) or cholera toxin subunit B conjugated with Alexa Fluor 647 (CTxB-AF488, Invitrogen, C34778) was added and incubation was continued for another 30 min. HFt-AF488, Tfn-AF488, and CTxB-AF488 uptake were then measured as described above. For cycloheximide chase experiment followed by TfR1 and MSR1 expression analysis with western blot, macrophages were treated with 20 μg/ml cycloheximide (Sigma-Aldrich, C4859) or vehicle solution (DMSO, Sigma-Aldrich, D2650) in serum-free RPMI-1640 (Sigma-Aldrich, R8758) for 1 h at 37 °C prior to addition of 200 μg/ml HFt.

### Blocking of scavenger receptors

On the day of the experiment, THP-1 macrophages and hMDM were pretreated for 30 min at 37 °C with various concentrations of ligands, known to bind to the scavenger receptors class A, diluted in serum-free RPMI 1640 (Sigma-Aldrich, R8758) such as polyinosinic (poly(I), Sigma-Aldrich, P4154) and polyguanylic (poly(G), Sigma-Aldrich, P4404) acid, fucoidan (Sigma-Aldrich, F5631) and dextran sulfate (Sigma-Aldrich, D6001) or with their negative controls: polycytidylic acid (poly(C), Sigma-Aldrich, P4903), mannan (Sigma-Aldrich, M3640) and chondroitin sulfate (Sigma-Aldrich, C9819), respectively. Then, HFt-AF488 or positive control, AcLDL conjugated with Alexa Fluor 488 (AcLDL-AF488, Invitrogen, L23380) were added to cells to yield a final concentration of 100 or 5 μg/ml, respectively. Macrophages were incubated for another 30 min at 37 °C and then they were detached from wells using Accutase (Sigma-Aldrich, A6964), washed twice with PBS (Biowest, L0615) and finally resuspended in FACS buffer consisting of PBS (Biowest, L0615), 2% BCS (Hyclone, SH30072.03), 2 mM EDTA (Invitrogen, AM9260G) in the presence of DRAQ7 Dead Cell stain (BioLegend, 424001) for 10 min before flow cytometry analysis.

### Real-time PCR

Total RNA was isolated using the Total RNA Mini Plus kit (A&A Biotechnology, 036-100) according to the manufacturer's protocol. Briefly, cells were lysed in Fenozol (A&A Biotechnology, 203-100P) and

RNA was purified using silica columns. cDNA synthesis was performed using the High-Capacity RNA-to-cDNA Kit (Applied Biosystems, 4387406), and quantitative RT-PCR was conducted with Universal SYBR Green Supermix (Bio-Rad, 1725124) on a Dx Real-Time PCR System (Thermo Fisher Scientific) or TaqMan Universal PCR Master Mix (Applied Biosystems, K1082) for scavenger receptor gene expression analysis, with data acquisition on AriaMx Real-time PCR System (Agilent) and analysis using Agilent Aria 1.71 software. The list of primer sequences is provided in Supplementary Table 1. Additionally, the TaqMan Gene Expression Assays (Applied Biosystems, 4331182) included *MSR1* (Hs00234007_m1), *SCARA5* (Hs01073151_m1), *MARCO* (Hs00198935_m1), and *ACTB (β-actin)* (Hs01060665_g1). Results were normalized to the housekeeping gene *ACTB*, and data were analyzed using the comparative Ct method.

### Cytokine measurements by bead-based immunoassay

Cytokines present in a conditioned medium collected from polarized macrophages were analyzed using the bead-based immunoassay LEGENDplex Human Macrophage/Microglia Panel (BioLegend, 740503) according to the manufacturer's instructions. Analytes were measured by flow cytometry using the FACSCanto II (BD Biosciences) flow cytometer and the LEGENDplex Data Analysis Software Version 8 (BioLegend).

### Biotinylation of ferritin

HFt was biotinylated using EZ-Link Maleimide-PEG2-Biotin kit (Thermo Scientific, A39261) according to the manufacturer's instruction. Briefly, in the first step disulfide bonds in HFt were reduced by incubation with 5 mM Bond-Breaker TCEP Solution (Thermo Scientific, 77720) for 30 min at RT, followed by TCEP removal using a Zeba Spin Desalting Column (7K MWCO, Thermo Scientific, 89891). Immediately after, an appropriate volume of freshly prepared 20 mM EZ-Link Maleimide-PEG2-Biotin solution (in PBS, Biowest, L0615) was added to reduced HFt and incubated overnight. The next day, free biotin was removed using Zeba Spin Desalting Column, and the resulting biotinylated protein (Biotin-HFt) was concentrated with Amicon Ultra-4 Centrifugal Filter Unit (10 kDa cutoff, Merck, UFC801024) in PBS (Biowest, L0615).

### Kinetic analysis of HFt binding with scavenger receptors

Kinetic analysis of HFt binding to MSR1 was performed using two methods—AlphaScreen Technology and spectral shift technology combined with microscale thermophoresis.

**AlphaScreen.** All AlphaScreen (PerkinElmer) assays were performed in duplicates using white 96-well half area microplates (Greiner, 675075) in a total volume of 40 μl, additions of reagents were carried out in subdued lighting due to photosensitivity of the beads and sealed plates were incubated in the dark while shaking at 300 rpm. Biotin-HFt, His-tagged MSR1 (His-MSR1, R&D Systems, 2708-MS-050), nickel chelated acceptor and streptavidin-coated donor beads (PerkinElmer, 6760619C) were diluted in assay buffer consisting of PBS (Biowest, L06151000), 1 mg/ml BSA (Sigma-Aldrich, A3294) and 0.05% Tween 20 (Sigma-Aldrich, P1379). A Biotin-HFt/His-MSR1 concentration matrix was set up as follows: 10 μl of Biotin-HFt (final concentration: 0.022–50 nM, 3-fold serial dilution) was mixed with 10 μl of His-MSR1 (final concentration: 0.41–100 nM, 3-fold serial dilution) and incubated at 37 °C for 1 h. Then, 10 μl of nickel-chelated acceptor beads (final concentration: 10 μg/ml) were added to each well followed by incubation at room temperature while shaking in the dark at RT for 1 h. In the last step, 10 μl of streptavidin donor beads were added, and after another hour of incubation at RT, the AlphaScreen signal was measured at a wavelength of 520–620 nM after laser excitation at 680 nm using the Spark multimode fluorescence microplate reader (Tecan).

Unlabeled HFt of varying concentration (0.076–1500 nM, 3-fold serial dilution) was used to compete with Biotin-HFt in an assay

containing 4 nM His-MSR1 and 2 nM Biotin-HFt. Unlabeled HFt-735 of varying concentration (0.006–1000 nM, 3-fold serial dilution) was used to compete with Biotin-HFt in an assay containing 8 nM His-MSR1 and 2 nM Biotin-HFt. The AlphaScreen assay setup was similar to the one described above with two modifications: (1) in the first step the addition sequence was as follows: (i) unlabeled HFt, (ii) Biotin-HFt, (iii) His-MSR1, (2) the volume of all reagents per well was changed to 8 µl to keep the final volume at 40 µl. The $IC_{50}$ value was determined by nonlinear regression with GraphPad Prism version 9.3.1 (Graphpad Software, USA).

**Spectral shift technology.** All measurements were performed using Monolith X (NanoTemper Technologies). In the first step, a kinetic buffer (used for sample dilution and as a binding buffer), was prepared by addition of Tween 20 (Sigma-Aldrich, P1379) to PBS (Biowest, L06151000) to a final concentration of 0.05%. For isothermal spectral shift analysis, the His-MSR1 (His-MSR1, R&D Systems, 2708-MS-050) served as a target, and it was conjugated with a red dye using His-Tag Labeling Kit RED-tris-NTA (NanoTemper Technologies, MO-L018) according to the manufacturer's protocol. MSR1-NTA-RED was then diluted to yield a concentration of 50 nM and mixed with a set of dilutions of unlabeled HFt (0.076–2500 nM). Well-mixed solutions were then transferred to capillaries (provided in the kit), incubated for 20 min at 37 °C and loaded into the instrument. The isothermal spectral shift caused by the interaction between the fluorescently labeled target and unlabeled ligand was measured using the Monolith X device. 670 nm/650 nm ratios and $K_d$ were automatically calculated in the software provided by the manufacturer (MO.Control 2 software). $K_d$ was determined by fitting the data with a 1:1 binding model.

## MTT assay

Cells were plated on 96-well plates (Falcon, 353072) at the concentration of $4 \times 10^4$ cells/well and incubated (37 °C, 5% $CO_2$) for 24 h. The next day, cells were treated with increasing concentrations of HFt, HFt-doxorubicin, doxorubicin, drug 250, or HFt-250. At respective time points, 100 µl of MTT stock solution (0.5 mg/ml, Sigma-Aldrich, M2128) was added into each well and cells were incubated for 2 h at 37 °C. Solution was then removed and the formazan crystals were dissolved in DMSO (Sigma-Aldrich, D4540). The absorbance was measured with a microplate reader at 570 nm absorption wavelength (Infinite 200 PRO, Tecan).

## Cell transfection

GFP-C1-PLCdelta-PH plasmid was a gift from Tobias Meyer (Addgene plasmid 21179; RRID:Addgene_21179)[76]. RAW 264.7 cells were transfected with GFP-C1-PLCdelta-PH plasmid by electroporation. Cells were resuspended in 250 µl Ingenio solution ($5 \times 10^6$ cells/ml, MirusBio, MIR 50117) with 5 µg of plasmid DNA, transferred to a 4 mm electroporation cuvette (Sigma-Aldrich, Z706094-50EA) and electroporated with AmaxaNucleofector (Lonza). 48 h after electroporation, GFP-C1-PLCdelta-PH-positive cells were sorted using BD FACSAria II (BD Sciences) and cultured in DMEM medium (Gibco, 41965062) with 0.8 mg/ml G418 (Gibco, 10131027) for two weeks. pT3-Neo-EF1a-GFP was a gift from Martin Bonamino (Addgene plasmid 69134; RRID:Addgene_69134) and modified by GenScript Biotech (USA) to exchange GFP for zsGreen sequence (GenBank ID: AF168422.1). SK-OV-3 cells were transfected with pT3-Neo-EF1a-zsGreen by electroporation with Sleeping Beauty transposase mRNA (GenScript Biotech, SC2346) 1:1 using the Neon Transfection System (Invitrogen) and selected in DMEM medium (Gibco, 41965062) with 0.6 mg/ml G418 (Gibco, 10131027).

DNA plasmid transfection of CHO-K1 and HEK293 cell lines was performed using the negative control vector (pCMV3-SP-N-His, Sino Biological, CV023) and the plasmid encoding human MSR1 (CD204/MSR1 cDNA ORF Clone, N-His tag expression plasmid (Sino Biological, HG10427-NH). Both plasmids were multiplied in Subcloning Efficiency

DH5α Competent Cells (Invitrogen, 18265017) and purified in-house using EndoFree Plasmid Maxi Kit (Qiagen, 12362). Plasmids pcDNA3.4 coding for mCherry2 and either wild-type MSR1 (p-MSR1-mCherry2) or MSR1 with a deletion of the SRCR domain (p-ΔMSR1-mCherry2) were custom-made and ordered from GenScript. Day before transfection cells were seeded in 24-well plates (Falcon, 353047) for flow cytometry analysis or 12-well plates (Falcon, 353043) for protein isolation and western blot analysis and grown in their respective medium at 37 °C in a humidified 5% $CO_2$ atmosphere. Cells were transfected with plasmids using the lipofection method with a commercially available kit Lipofectamine LTX Reagent with PLUS Reagent (Invitrogen, 15338100) according to the transfection protocol optimized for CHO-K1 and HEK293 provided by the manufacturer. Cells were incubated for 24 h at 37 °C in a humidified 5% $CO_2$ atmosphere before further experiments.

Luciferase-expressing cell lines were generated by lentiviral transfection with EF1a-(Red-Luciferase) Lentivirus (Amsbio, LVP475) and selected using puromycin (Gibco, A1113803) according to the manufacturer's protocol.

Knockdown of selected genes (TfR1, CLTC, ICAM-1, and MSR1) was achieved using siRNA transfection with Lipofectamine RNAiMAX (Invitrogen, 13778075) or electroporation using the Neon Transfection System (Invitrogen). Predesigned and validated Silencer Select siRNAs (Thermo Scientific) were used, including two different siRNAs for each gene. For TfR1: siTfR1-1 (Assay ID: s725) and siTfR1-2 (Assay ID: s727); for CLTC: siCLTC-1 (Assay ID: s475) and siCLTC-2 (Assay ID: s477); for ICAM-1: siICAM-1-v2 (Assay ID: s7087); for MSR1: siMSR1-1 (Assay ID: s8987) and siMSR1-2 (Assay ID: s8988); and for CAV1: siCAV1-1 (Assay ID: s2446) and siCAV1-2 (Assay ID: s2448). A non-targeting control siRNA (Scramble siRNA, 4390843) was used in all experiments.

Transfection with Lipofectamine RNAiMAX (Invitrogen, 13778075) was performed according to the manufacturer's protocol. For electroporation, THP-1 monocytes and partially differentiated hMDM (after 3 days of M-CSF stimulation) were electroporated with siRNA at 250 nM using the Neon Transfection System with parameters of 1500 V, 10 ms, and 3 pulses. Cells were then seeded into RPMI-1640 medium (Sigma-Aldrich, R8758) supplemented with 20% FBS (HyClone, SV30160.03), 50 ng/ml M-CSF (for hMDM, BioLegend, 574808) or 100 ng/ml PMA (for THP-1, Sigma-Aldrich, P1585) and incubated at 37 °C for 72 h to complete macrophage differentiation.

## Western blot

Pierce BCA Protein Assay Kit (Thermo Scientific, 23225) was used for the determination of protein concentration in cell or extracellular vesicle lysates obtained after cell lysis in RIPA buffer (Thermo Scientific, 89901) supplemented with EDTA and protease/phosphatase inhibitors (Halt Protease and Phosphatase Inhibitor Cocktail; Thermo Scientific, 78442). Protein samples were mixed with 4× LDS buffer (Thermo Scientific, NP0007) and 10× Reducing Agent (Thermo Scientific, NP0004) and heated at 70 °C for 10 min. Protein ladder (Thermo Scientific, 26619) and samples (20 µg/well) were loaded on 4–20% 15-well Mini-PROTEAN TGX Gel (Bio-rad, 4561095) and separated during SDS−electrophoresis in the Mini-PROTEAN Tetra Cell (Bio-rad) with 1X Tris-Glycine-SDS (Bio-rad, 1610732) running buffer at 160 V for 30 min. Protein was transferred from a gel to a nitrocellulose membrane in Mini-PROTEAN Tetra Cell using prechilled 1X Tris-Glycine transfer buffer (Bio-rad, 1610734) at 75 V for 90 min. The membrane was blocked in 5% w/v non-fat dry milk (Mlekovita) in Tween 20 Tris Buffered Saline (TTBS) (Thermo Scientific, 28360) for 1 h at room temperature. Membranes were incubated overnight at 4 °C with primary antibodies: anti-clathrin heavy chain (1:1000; Abcam, ab21679), anti-TfR1 (1:1000; Cell Signaling Technology, 13113), anti-CD81 (D3N2D) (1:1000; Cell Signaling Technology, 56039), anti-CD9 (D8O1A) (1:1000; Cell Signaling Technology, 13174), anti-flotillin-1 (D2V7J) XP (1:1000; Cell Signaling Technology, 18634), anti-Alix (E6P9B) (1:1000; Cell Signaling Technology, 92880), anti-MSR1

(1:1000, Cell Signaling Technology, 17275), anti-GM130 (D6B1) XP (1:1000; Cell Signaling Technology, 12480), anti-Annexin V (1:1000; Cell Signaling Technology, 8555), anti-ICAM1 (E3Q9N) XP (1:1000; Cell Signaling Technology, 67836), anti-GAPDH (1:5000, Invitrogen, PA5-85074), anti-lamin B1 (1:1000, Abcam, ab16048), anti-β-actin (1:10000; Proteintech, 66009-1) or anti-α-tubulin (DM1A) (1:1000; Cell Signaling Technology, 3873), diluted in 2.5% w/v non-fat dry milk in TTBS. Membranes were then washed three times in TTBS and incubated with secondary antibodies diluted 1:10,000 in 2.5% w/v non-fat dry milk in TTBS: anti-rabbit IgG, HRP-linked antibody (Cell Signaling Technology, 7074) or anti-mouse IgG, HRP-linked antibody (Cell Signaling Technology, 7076) for 1 h at room temperature. Membranes were washed three times in TTBS, and protein was visualized by using ECL Western blotting substrate (Thermo Scientific, 32106) and iBright 1500 Imaging System (Thermo Scientific). The intensities of bands were quantified by using iBright Analysis Software (Thermo Scientific).

## Co-culture

To test HFt transfer macrophages were incubated with fluorochrome labeled HFt at indicated concentrations in serum-free DMEM medium (Gibco, 41965062) for 1 h (37 °C, 5% $CO_2$). Cancer cells were stained with the CellTrace dye (Invitrogen, C34557 or C34564) according to the manufacturer's instructions. Then cells were seeded at 1:1 ratio ($0.5 \times 10^6$ macrophages: $0.5 \times 10^6$ cancer cells) on a 48-well plate (Falcon, 353078) and incubated for 4 h or at 2:1 ratio ($0.4 \times 10^6$ macrophages; $0.2 \times 10^6$ cancer cells) on a 24-well plate (Falcon, 353047) and incubated for 24 h. To assess HFt transfer between physically separated donor and acceptor cells, the 24-well Transwell (Corning, 3413) inserts were used. $0.5 \times 10^6$ macrophages were seeded on the 0.4 μM inserts and $0.5 \times 10^6$ cancer cells were seeded in the lower chambers and incubated for 24 h. Cells were detached from the plate, washed twice with PBS (Biowest, L0615) and resuspended in FACS buffer consisting of PBS (Biowest, L0615), 2% BCS (Hyclone, SH30072.03), 2 mM EDTA (Invitrogen, AM9260G) in the presence of DRAQ7 dead cell stain (BioLegend, 424001) according to the manufacturer's instructions prior to flow cytometry acquisition using BD FACSAria II or BD FACSCanto II flow cytometer (BD Biosciences). Data was analyzed using FlowJo software (BD Biosciences).

To study in vitro cytotoxicity against cancer cells by MDC complexes, macrophages were first incubated with the HFt-drug complexes in serum-free DMEM medium (Gibco, 41965062) for 1 h at 37 °C and 5% $CO_2$. Loaded macrophages were then co-cultured with CellTrace-labeled cancer cells in a 24-well plate (Falcon, 353047) using a ratio of 2 macrophages to 1 cancer cell ($0.2 \times 10^6$ macrophages to $0.1 \times 10^6$ cancer cells per well), and the cultures were maintained for 48–96 h. Post-incubation, cells were gently detached using Accutase (Sigma-Aldrich, A6964), stained with DRAQ7 dead cell stain (BioLegend, 424001) according to the manufacturer's instructions, and resuspended in flow cytometry buffer for analysis using a BD FACSCanto II flow cytometer. CountBright Absolute Counting Beads (Invitrogen, C36950) were added prior to cell acquisition to determine the viability of cancer cells as a number of live cells relative to a control condition.

For the co-culture assay with blocking antibodies, macrophages loaded with HFt-AF488 were preincubated with FcR blocking solution (Miltenyi Biotec, 130-059-901) for 30 min. Following this, the cells were incubated with ICAM-1 monoclonal antibody (1A29, Invitrogen, MA5407) or an appropriate isotype control (P3.6.2.8.1, eBioscience, 16-4714-82) at a concentration of 20 μg/ml (dilution 1:50) for 1 h. The macrophages were then co-cultured with MDA-MB-231 cancer cells at a 2:1 ratio ($2 \times 10^5$ macrophages: $1 \times 10^5$ cancer cells) for 4 h, with the antibodies present throughout the co-culture period.

## Inhibitors

The effect of selected inhibitors on HFt transfer from macrophages to cancer cells was studied in RAW 264.7 and EMT6 co-culture.

Macrophages were loaded with HFt and cancer cells were stained with CellTrace dye (Invitrogen, C34557) as described above. Cells were preincubated in DMEM growth medium (Gibco, 41965062) at 37 °C, 5% $CO_2$ in the presence of inhibitor: latrunculin B (Sigma-Aldrich, L5288), cytochalasin D (Sigma-Aldrich, PHZ1063), dimethyl amiloride (DMA; Sigma-Aldrich, A4562), Wortmannin (Sigma-Aldrich, W3144), Piceatannol (Sigma-Aldrich, P0453) and a vehicle solution (DMSO, Invitrogen, D12345) at the same concentration as a negative control. Then cells were seeded at a 1:1 ratio ($0.5 \times 10^6$ macrophages: $0.5 \times 10^6$ cancer cells per well) on a 48-well plate (Falcon, 353078) and incubated for 4 h also in the presence of an inhibitor. Cells were detached from the plate, washed twice with PBS (Biowest, L0615), resuspended in flow cytometry buffer, and analyzed using BD FACSAria II flow cytometer (BD Biosciences).

## Conditioned medium

Cancer cells were cultured separately or in co-culture with macrophages for 24 h, then conditioned media (CM) were collected and filtered through a 0.45 μm syringe filter (Sigma-Aldrich, CLS431225). To achieve comparable experimental conditions, cell density and volumes of media were the same as in experiments with HFt transfer in direct co-cultures. To assess whether HFt is released to the control medium, macrophages were loaded for 1 h with 0.5 mg/ml HFt-AF488 -and incubated for 24 h in normal medium or in co-culture CM. Medium was then collected, filtered through a 0.45 μm syringe filter (Sigma-Aldrich, CLS431225), and added to the cancer cells seeded on a 24-well plate (Falcon, 353047). After 24 h, cells were detached and analyzed using BD FACS Aria II flow cytometer (BD Biosciences).

## Isolation of extracellular vesicles

Extracellular vesicles were isolated from the conditioned medium by differential centrifugation. To this end, THP-1 macrophages were incubated in the absence (control cells) or presence of HFt-AF488 (0.5 mg/ml) in serum-free RPMI-1640 medium (Sigma-Aldrich, R8758) for 1 h at 37 °C. Next, cells were washed in PBS (Biowest, L0615), centrifuged ($400 \times g$, 5 min), counted and used for co-culture experiments with MDA-MB-231 or seeded on 60 mm plates in serum-free RPMI-1640 medium (Sigma-Aldrich, R8758) and cultured for 24 h at 37 °C. Conditioned media were collected from 24 h cell cultures and centrifuged at $400 \times g$, 4 °C for 5 min to eliminate cell debris. Supernatant fractions were further centrifuged at $3000 \times g$ for 10 min and $10,000 \times g$, 4°C for 30 min. Extracellular vesicles were then pelleted by ultracentrifugation at $110,000 \times g$, 4 °C for 90 min, followed by washing with PBS and ultracentrifugation at $110,000 \times g$, 4 °C for 90 min. Extracellular vesicles were resuspended in PBS for uptake experiments or RIPA buffer for western blot analysis.

## Flow cytometry

Ligand uptake: macrophages were incubated with labeled ligands for the indicated time periods. Cells were washed in PBS (Biowest, L0615), detached with Accutase (Sigma-Aldrich, A6964), washed twice with PBS and finally resuspended in FACS buffer consisting of PBS (Biowest, L0615), 2% BCS (Hyclone, SH30072.03), 2 mM EDTA (Invitrogen, AM9260G) in the presence of DRAQ7 dead cell stain (BioLegend, 424001) for 10 min before analysis on BD FACSCanto II flow cytometer integrated with BD FACSDiva Software (BD Biosciences). Data analysis was performed using the FlowJo software.

Cell surface staining: cells after treatment were harvested with PBS (Biowest, L0615) supplemented with 2 mM EDTA (Invitrogen, AM9260G), washed with PBS (Biowest, L0615), counted, and transferred to a 96-well V-bottom plate ($3–4 \times 10^5$ cells/well, Falcon, 353263). Cells were then resuspended in 80 μl of FACS buffer consisting of PBS (Biowest, L0615), 2% BCS (Hyclone, SH30072.03), 2 mM EDTA (Invitrogen, AM9260G), mixed with 20 μl of the human FcR Blocking Reagent (Miltenyi Biotec, 130-059-901) and incubated for

10 min on ice to block non-specific antibody binding. The cells were then incubated with one of the following antibodies: PE anti-human CD204/MSR1 antibody (1:20, 371904, BioLegend), APC anti-human CD71/TfR1 antibody (clone OKT9) (1:20, 17-0719-42, eBioscience), PE mouse IgG2a, κ isotype control antibody (1:20, 400214, BioLegend), APC mouse IgG1 κ isotype control antibody (P3.6.2.8.1) (1:20, 17-4714-82, eBioscience) for 1 h on ice in the dark. Afterward, cells were washed twice with ice-cold PBS (Biowest, L0615), resuspended in 100 μl of FACS buffer consisting of PBS (Biowest, L0615), 2% BCS (Hyclone, SH30072.03), 2 mM EDTA (Invitrogen, AM9260G) and analyzed on BD FACSCanto II flow cytometer. Data analysis was performed using the FlowJo software.

To evaluate hMDM/MDC phenotype, single-cell suspension were incubated with specific antibodies (dilution 1:100): CD115-AF488 (clone 9-4D2-1E4, 347312, BioLegend), CD86-PE (clone GL1, 12-0862-82, Invitrogen), HLA-DR-APC (clone LN3, 17-9956-42, Invitrogen), SIRPα (CD172a)-APC (clone 15-414, 372106, BioLegend), CD14-APC (clone HCD14, 325608, BioLegend), CD11b-FITC (clone M1/70, 53-0112-82, Invitrogen), 25F9-e660 (clone 25F9, 50-0115-42, Invitrogen), CD204 (SRA)-BV421 (clone U23-56, 742438, Becton Dickinson), and CD280-PE (clone E1/183, 566817, Becton Dickinson), and FcR Blocking Reagent (Miltenyi Biotec, 130-059-901) at 4 °C for 20 min according to the manufacturer's manual, washed twice with PBS (Biowest, L0615), and resuspended in FACS buffer consisting of PBS (Biowest, L0615), 2% BCS (Hyclone, SH30072.03), 2 mM EDTA (Invitrogen, AM9260G) in the presence of Zombie Dead Cell Stain (BioLegend, 423102, 423106 or 423114 diluted 1:400) for 10 min at RT prior to flow cytometry analysis using a BD FACSCanto II flow cytometer. Data were analyzed using FlowJo software.

### Immunofluorescence and confocal microscopy

THP-1 monocytes were seeded onto glass coverslips (Marienfeld, 0111500) at $2.5 \times 10^5$ cells/well in a 24-well plate (Falcon, 351147) and differentiated to macrophages in complete RPMI-1640 medium (Sigma-Aldrich, R8758) with PMA (100 ng/ml, Sigma-Aldrich, P1585) for 3 days. Differentiated THP-1 macrophages were washed with PBS (Biowest, L0615) and incubated in a complete RPMI-1640 medium (Sigma-Aldrich, R8758) for another 24 h. Primary macrophages (hMDM) were detached from culture dishes using Accutase (Sigma-Aldrich, A6964), counted, and seeded onto glass coverslips (Marienfeld, 0111500) at $2.5 \times 10^5$ cells/well in a 24-well plate (Falcon, 351147) and left overnight to adhere. On the day of the experiment, THP-1 macrophages and hMDM were treated as indicated. Afterwards, cells were transferred on ice, washed twice with ice-cold PBS (Biowest, L0615) and incubated with 4% PFA (Thermo Scientific, 28908) for 15 min at room temperature. Fixed cells were then washed 3 times with PBS (Biowest, L0615) and incubated with 0.5 ml of solution I (PBS, 1% saponin (Sigma-Aldrich, S7900), 0.2% cold water fish skin gelatin (Sigma-Aldrich, G7765), 5 mg/ml BSA (Sigma-Aldrich, A3294)) for 1 h at room temperature. Permeabilized cells were then incubated overnight at 4 °C with the primary antibodies against an endocytic marker—Early Endosome Antigen 1 (EEA1, 1:500, MA514794, Invitrogen), or a lysosome marker—LAMP1 (1:200, 9091S, Cell Signaling Technology) diluted in 0.5 ml of solution II (PBS, 0.01% saponin, 0.2% fish skin gelatin). On the next day, cells were washed twice with solution II and incubated for 30 min at room temperature with the fluorochrome-conjugated secondary antibody (Goat anti-Rabbit IgG (H + L) Cross-Adsorbed Secondary Antibody, Alexa Fluor 647, A-21244, Invitrogen) diluted 1:500 in solution II and Hoechst 33342 at concentration of 2 μg/ml (Invitrogen, H3570). After immunostaining, cells were washed 3 times with PBS followed by a wash with deionized water. Coverslips were then transferred facing down on a drop of mounting medium (ProLong Gold Antifade Mountant, Thermo Scientific, P36934) on microscope slides and left overnight to dry at room temperature (in the dark).

Images of cells were obtained with the Leica TCS SP5II confocal microscope (Leica Microsystems) equipped with a ×63 objective and processed using ImageJ software (NIH).

To image HFt in donor and acceptor cells, macrophages and cancer cells were co-cultured on μ-Slide 8 well-chambered coverslips (Ibidi, 80826) at the ratio 2:1 ($1 \times 10^5$ macrophages and $0.5 \times 10^5$ cancer cells). Incubation with HFt and CellTrace dye (Invitrogen, C34557 or C34564) was performed as described above; plasma membranes were stained with wheat germ agglutinin (WGA) conjugated with Alexa Fluor 555 (5 μg/ml, Invitrogen, W32464), for 10 min at RT. Cells were imaged live or fixed with 4% PFA (10 min incubation, Thermo Scientific, 28908) using TCS SP5II (Leica Microsystems) or FV500 (Olympus) confocal microscopes under ×60 objective. For immunofluorescence imaging, fixed cells were washed with PBS and then permeabilized and blocked in PBS (Biowest, L0615) with 0.1% w/v saponin (Sigma-Aldrich, S7900), 0.2% w/v fish skin gelatin (Sigma-Aldrich, G7765) and 5 mg/ml BSA (Sigma-Aldrich, A3294) solution for 15 min. Samples were incubated with primary antibodies: anti-CD63 (1:100, 25682-1-AP, Proteintech) diluted in PBS with 0.1% w/v saponin and 0.2% w/v fish skin gelatin for 30 min at room temperature, washed twice with a PBS with saponin and gelatin solution and incubated for 30 min at room temperature with secondary antibodies: goat anti-Rabbit IgG (H + L) conjugated with Alexa Fluor 568 (1:500; A-11036, Thermo Scientific) diluted in the same buffer. Image analysis was performed using ImageJ software. Acquired z-stacks were visualized using smooth 2D manifold extraction plugin[77].

### Imaging flow cytometry

Method for the qualitative and quantitative analysis of immune synapses based on the imaging flow cytometry was adapted from the published protocol[78]. Preparation of cells for co-culture—incubation with HFt and CellTrace labeling dye (Invitrogen, C34557)—were performed as described above.

To quantify rearrangement of the actin cytoskeleton hMDM and MDA-MB-231 cells were co-cultured for 2 h in the 5 ml polystyrene tubes (Falcon, 352054) at the ratio 2:1 ($8 \times 10^5$ macrophages and $4 \times 10^5$ cancer cells) in 1 ml of medium (RPMI (Sigma-Aldrich, R8758) + 10% FBS (HyClone, SV30160.03) + 20 ng/ml M-CSF (BioLegend, 574808)). To fix cells, 1.5 ml PFA (2%, Thermo Scientific, 28908) was slowly added to the tubes during gentle vortexing. After 10 min incubation at RT, fixation was stopped by adding 1.5 of PBS (Biowest, L0615) + 1% BSA (AppliChem, A1391). Cells were centrifuged ($300 \times g$ for 8 min at RT). To visualize F-actin, cells were resuspended in permeabilization solution (PBS + 1% BSA + 0.1% saponin and stained with phalloidin labeled with iFluor 555 (Abcam, ab176756).

For cell surface staining with antibodies, hMDM and MDA-MB-231 cells were co-cultured for 24 h on 24-well plates (Falcon, 353047) at the ratio 2:1 ($6 \times 10^5$ macrophages and $3 \times 10^5$ cancer cells). Cells were detached by 5 min incubation with Accutase (Sigma-Aldrich, A6964) at RT and gently washed with PBS (Biowest, L0615) to avoid disruption of cell–cell connections. Following 20 min incubation with blocking buffer (FcR block (Miltenyi Biotec, 130-059-901) and 1% BSA (AppliChem, A1391) solution in PBS), cells were stained with anti-ICAM-1-APC (1:100, BioLegend, 353112) and/or anti-CD11b-FITC (1:100, Abcam, ab269333) for 30 min at 4 °C.

After incubation with staining solutions, cells were washed with PBS (Biowest, L0615) and centrifuged $250 \times g$, 5 min. Then, cells were gently resuspended in PBS at the concentration of $1–5 \times 10^7$ cells/ml and analyzed using the Image Stream X Mark II (Amnis) imaging flow cytometer (Flow Cytometry Facility, University of Zurich). Events were gated to capture images of macrophage-cancer cell doublets, in-focus and positive for marker signal. Recorded data was analyzed using IDEAS 6.2 software (Amnis). Fluorescence intensity analysis on exported images was performed using ImageJ.

## Animal experiments

**Optical Imaging.** An IVIS Lumina II imaging system (Caliper Sciences) was employed to detect live luciferase-labeled tumor cells and tumor growth in mice. Mice were injected intraperitoneally (i.p.) with 90 mg/kg D-luciferin (Caliper Life Sciences, 760504) dissolved in sterile water and anaesthetized using 2.5% isoflurane (Abbott Scandinavia AB, 506949) in 100% oxygen at 3.5 L/min (for induction and 1.5% for maintenance) in the anesthesia chamber. Images were taken every 3 min as a sequence of 10 images for every group, once a week. Automatic contour regions of interest were created, and the tumor sizes were quantified as photons per s per $cm^2$ per steradian ($ps^{-1}/cm^2/sr^1$). Progression and spread of tumors were evaluated by calculating the tumor radiance values from inoculated mice in each group.

**Efficacy study.** A total of 52 female athymic nude mice (strain code 490, bred in-house by Axis Bio Services Ltd.) aged 6–8 weeks were used for the study with MDC-250 in BxPC-3-luc model. As mice were bred in-house by Axis Bio Services Ltd. therefore they required no acclimatization period. Tumor cell implantation: animals were anaesthetized with a ketamine (Ketamidor, Richter)/xylazine (Rompun, Bayer) mix and BxPC-3-luc cells ($1 \times 10^6$ cells 1:1 in Matrigel, Corning, 356231) were orthotopically implanted into the pancreas of female athymic nude mice using a 25-gauge needle. Animals were imaged 14 days post-implant to confirm tumor growth. When tumor burden was detected by bioluminescence, animals were entered into the study. To ensure comparability at the onset of treatment, animals were randomized into seven groups, achieving statistically similar tumor sizes as verified by bioluminescence measurements ($n = 7$–$10$): vehicle control group (PBS control, intraperitoneal, Q2D for 4 doses), empty macrophage control group (Macrophages, intraperitoneal, $10 \times 10^6$ cells, Q2D for 4 doses), free HFt-drug conjugate control group (HFt-250, at the dose ensuring equivalent 250 drug dose given as MDC-250, intraperitoneal, Q2D for 4 doses), free drug control group (250, at the dose ensuring equivalent 250 drug dose given as MDC-250, intraperitoneal, Q2D for 4 doses), standard of care group gemcitabine (Gemzar, Eli Lilly) 50 mg/kg, intravenously, BIW) and treatment group (MDC-250, $5 \times 10^6$ cells, intraperitoneal, Q2D for 4 doses). Tumor growth was evaluated by the bioluminescence measurement 7 times, every week from day 0 to day 42. The bioluminescence measurements were not conducted throughout the study because, in some orthotopic tumor models, bioluminescence fails to accurately depict disease progression due to optical scattering and signal attenuation[79]. Mice were weighed and scored for tumor-related symptoms twice a week and were euthanized at humane endpoints or at the end of the study (day 100).

A total of 32 female athymic nude mice (strain code 490, bred in-house by Axis Bio Services Ltd.) aged 6–8 were used for the MDC-735 study in the BxPC-3-luc model. As the mice were bred in-house by Axis Bio Services Ltd. they required no acclimatization period. Mice were anesthetized with a ketamine (Ketamidor, Richter)/xylazine (Rompun, Bayer) mix, and BxPC-3-luc cells ($1 \times 10^6$ cells in a 1:1 mixture with Matrigel, Corning, 356231) were orthotopically implanted into the pancreas using a 25-gauge needle. MDC-735 cells were prepared by incubation in a 1 mg/ml HFt-735 solution in RPMI-1640 serum-free medium (Sigma-Aldrich, R8758) for 1 h. The cells were then frozen in Bambanker (Lyphotec, BB01) and stored in liquid nitrogen until use. Fourteen days post-implantation, tumor establishment was confirmed via imaging, and the mice were then randomly assigned to one of four treatment groups (the vehicle control group received PBS administered intraperitoneally (i.p.) every other day for four doses; the gemcitabine group received 50 mg/kg of gemcitabine (Gemzar, Eli Lilly) intravenously (i.v.) twice a week; the MDC-735 group received $5 \times 10^6$ cells in 200 µl administered i.p. every other day for four doses; and the combination group received 50 mg/kg of gemcitabine i.v. twice a week and $5 \times 10^6$ MDC-735 cells in 200 µl i.p. every other day for four doses).

Imaging was performed weekly on days 0, 7, 14, 21, and 28 using the In Vivo Imaging System (IVIS) after intraperitoneal injection of luciferin (150 mg/kg), 10–15 min before imaging.

A total of 24 female SPF mice BALB/ccmdb (strain is a direct inbred derivative of the BALB/cJ strain, Charles River) aged 6–8 weeks purchased from the Animal Facility of the Bialystok Medical University were used for the study with MDC-Dox in EMT6-luc model. Tumor cell implantation: animals were anaesthetized with a ketamine (Ketamidor, Richter)/xylazine (Rompun, Bayer) mix and EMT6-luc cells ($1 \times 10^6$ cells 1:1 in Matrigel, Corning, 356231) were subcutaneously implanted into the legs of mice using a 25-gauge needle. Mice with 7 days old EMT6 tumors were i.t. administered with $1 \times 10^6$ of MDC loaded with HFt-doxorubicin, HFt-doxorubicin given at the dose ensuring equivalent of doxorubicin administered in BMDM, and plain BMDM to check their role in tumor development. Mice received 4 injections: on days 7, 9, 12, and 14. On day 16 tumor burden was calculated based on bioluminescence.

A total of 40 female C3H/HeN mice (Charles River, strain code 025, aged 5-8 weeks) were used to assess the efficacy of MDC-735 in the SCC7 tumor model. The mice were acclimated for 7 days prior to the study. For tumor implantation, SCC7 cells ($5 \times 10^5$) were injected into the flank of the mice using a 25-gauge needle while under isoflurane anesthesia. After 7 days, when tumor volumes reached 10-30 mm³, the animals were assigned to one of four treatment groups: (1) Vehicle control (CS10 freezing media, Sigma-Aldrich, C2874) plus isotype control (Rat IgG2a, BioXcell, BP0089) with 40 µl of intratumoral (i.t.) injections on days 7 and 12, and 10 mg/kg of intraperitoneal (i.p.) administration twice weekly (BIW) for four doses; (2) Mouse MDC-735 ($4 \times 10^6$ cells in 40 µl i.t. on days 7 and 12); (3) aPD-1 antibody (Clone RMP1-14, BioXcell, BE0146) at 10 mg/kg i.p. BIW for four doses; and (4) Mouse MDC-735 ($4 \times 10^6$ cells i.t.) combined with aPD-1 antibody (10 mg/kg i.p. BIW for four doses). Tumor volume was measured three times per week using electronic calipers and calculated using a 3D equation.

A total of 40 male C57BL/6 mice (strain code: C57BL/6JRj, aged 6-8 weeks) were purchased from Janvier Laboratories and used to evaluate the efficacy of MDC-735 in the MB49 tumor model. For tumor implantation, MB49 cells ($5 \times 10^5$ cells in a 1:1 mixture with Matrigel, Corning, 356231) were injected into the flank of the mice using a 25-gauge needle while under isoflurane anesthesia. Seven days post-implantation, when tumors reached a volume of 30–50 mm³, animals were assigned to one of four treatment groups to ensure equal tumor volume distribution: (1) Vehicle control (CS10 freezing media, Sigma-Aldrich, C2874) plus isotype control (Rat IgG2a, BioXcell, BP0089) with 20 µl intratumoral (i.t.) injections and 10 mg/kg intraperitoneal (i.p.) injections twice weekly (BIW) for four doses; (2) Mouse MDC-735 ($2 \times 10^6$ cells in 20 µl i.t., with the second dose given 5 days after the first); (3) aPD-1 antibody (Clone RMP1-14, BioXcell, BE0146) at 10 mg/kg i.p. BIW for four doses; and (4) Mouse MDC-735 ($2 \times 10^6$ cells i.t.) combined with aPD-1 antibody (10 mg/kg i.p. BIW for four doses). Tumor volume was measured three times per week using electronic calipers, with volume calculated using a 3D equation.

A total of 28 female athymic nude mice (strain code: 490, Charles River, aged 5–7 weeks) were acclimatized for 7 days before the start of the study of MDC-735 in SK-OV-3. SK-OV-3-luc cells ($5 \times 10^5$ cells in 10 µl of Geltrex, Gibco, A1413202) were orthotopically implanted into both the left and right ovaries of each mouse using a 30-gauge needle. Tumor establishment was confirmed via bioluminescence imaging 14 days post-implantation, after which the mice were assigned to treatment groups. The groups were as follows: control group received vehicle orally every other day for five doses, the treatment group was administered MDC-735 ($1 \times 10^7$ cells) intravenously every other day for five doses, and the SoC group was administered paclitaxel (20 mg/kg, Santa Cruz, SC-201439) intraperitoneally every five days. The treatment duration was 28 days. Tumor growth was monitored weekly through bioluminescence imaging.

A total of 50 female SPF mice BALB/ccmdb (strain is a direct inbred derivative of the BALB/cJ strain, Charles River) aged 6–8 weeks purchased from the Animal Facility of the Bialystok Medical University and acclimatized for 7 days before the start of the study of MDC-735 in the EMT6-Fluc lung colonization model. Lung tumors were established via intravenous injection of $0.85 \times 10^5$ EMT6-Fluc-Puro cancer cells. Following tumor establishment, mice were assigned to one of three treatment groups: Group I received PBS, administered *i.v.* five times; Group II received MDC loaded with HFt-735 (0.25 mg/ml) at $5 \times 10^6$ cells, administered *i.v.* four times; and Group III received MDC loaded with HFt-735 (0.5 mg/ml) at $5 \times 10^6$ cells, administered *i.v.* four times. Bioluminescence imaging of EMT6-Fluc-Puro cancer nodules in the lungs was performed using the In-Vivo MS FX PRO imaging system (Bruker BioSpin) to monitor tumor progression.

In another study with this model, a total of 22 female SPF mice BALB/ccmdb (strain is a direct inbred derivative of the BALB/cJ strain, Charles River) aged 6–8 weeks purchased from the Animal Facility of the Bialystok Medical University. Lung tumors were established via *i.v.* injection of $0.7 \times 10^5$ EMT6-Fluc-Puro cancer cells. The mice were assigned to one of three treatment groups: Group I served as the control and received PBS administered *i.v.* twice (on days 8 and 12) following tumor implantation; Group II received 5-day-old murine MDC-735 (loaded in 0.5 mg/ml HFt-735) injected twice; and Group III was treated with a doxorubicin (1 mg/kg, MedChem Express, HY-15142), administered *i.v.* twice. Tumor progression was monitored using bioluminescence imaging as described.

**Investigation of the safety of the locally administered therapy.** Female SPF mice BALB/ccmdb (strain is a direct inbred derivative of the BALB/cJ strain, Charles River) aged 6-8 weeks purchased from the Animal Facility of the Bialystok Medical University. Healthy mice naïve mice received *i.v.* a single injection of $15 \times 10^6$ of autologous macrophages or a single injection of $15 \times 10^6$ of allogeneic macrophages obtained from C57BL6/cmdb (imported from The Jackson Laboratory) purchased from the Animal Facility of the Bialystok Medical University or PBS (control). 7- and 14-day post-injection mice were euthanatized. Blood and organ samples were taken for further analysis. Blood samples affected by hemolysis or deemed to be of insufficient quality were excluded from the analysis.

**Histology.** Three female C3H/HeN mice (strain code 025, Charles River, aged 5–8 weeks) were subjected to a 7-day acclimation period before any procedures. Non-tumor-bearing mice were treated with MDC-735 ($4 \times 10^6$ cells in 40 µl) by subcutaneous injection and were euthanized after 28 days. Skin surrounding the injection site, along with the lungs, liver, and spleen, were resected and processed into formalin-fixed paraffin-embedded (FFPE) samples for further analysis. Serial sections 3–4 µm-thick were stained with Hematoxilin and Eosin (HE, Sigma-Aldrich MHS16 and 318906) and evaluated by light microscopy. Images were digitized with the DS-Ri2 digitizing camera (Nikon Corporation, Japan) connected to the microscope Eclipse Ci-L (Nikon Corporation, Japan) using NIS-Elements Br-2 as software (v 5.10; Nikon).

A total of 9 female Balb/c nude mice (strain code: 194; bred in-house by Axis Bio Services Ltd., aged 5–7 weeks) were used for the intracerebral tumor model study. U87-MG-luc cells ($1 \times 10^5$ cells in 5 µl of Matrigel, Corning, 356231) were implanted intracerebrally using a 30-gauge needle into the brain of each mouse. Tumor establishment was confirmed via bioluminescence imaging at 1- and 2-weeks post-implantation. Mice were then assigned to one of three treatment groups, receiving intratumorally either PBS or MDC-735 at doses of $0.5 \times 10^6$ or $2.5 \times 10^6$ in 4 µl of suspension. Mice were monitored and maintained for 30 days post-implantation. On day 30, all mice were euthanized, and their brains were harvested. Brain samples were fixed in paraffin for sectioning and subsequent analysis. Brain tissue sections

were stained with anti-CD68 (26042, Cell Signaling Technology) and anti-Ki67 (34330, Cell Signaling Technology) antibodies at 1:500 according to the manufacturer's instructions and imaged using a Glissando imaging apparatus.

**Immunofluorescence imaging.** Athymic nude mice (strain code: 490; bred in-house by Axis Bio Services Ltd., females for SK-OV-3 model and males for A549-luc model, aged 5–8 weeks) were used for the macrophage targeting study. As the mice were bred in-house they required no acclimatization period. Mice were anesthetized with a ketamine (Ketamidor, Richter)/xylazine (Rompun, Bayer) mix, and SK-OV-3 cells ($5 \times 10^5$ cells in a 1:1 mixture with Geltrex, Gibco, A1413202) or A549-luc ($1 \times 10^5$ cells in a 1:1 mixture with Matrigel, Corning, 356231) were orthotopically implanted into the ovary or lung, respectively. hMDM were labeled with XenoLight DIR (Revvity, 125964) according to the manufacturer's protocol. Two weeks after tumor implantation, $5 \times 10^6$ hMDM was injected into the tail vein. 24 h after injection, the mice were imaged for fluorescence intensity, next euthanized, and organs were harvested, imaged for fluorescence intensity and embedded in OCT (Leica Biosystems, 14020108926), and frozen. Cryosections of tumor-bearing and control ovaries were fixed with 4% PFA (Thermo Scientific, 28908) for 10 min, permeabilized with PBS (Biowest, L0615) containing 0.1% Tween 20 for 10 min, and blocked in SuperBlock solution (Thermo Scientific, 37515) for 30 min. After blocking, the samples were incubated with primary Isolectin-IB4 Alexa Fluor 488 conjugate (Thermo Scientific, I21411). Nuclei were stained with DAPI (Thermo Scientific, 62248).

**Ethics for animal experiments.** Animal experiments were approved by the 1st and 2nd Local Ethical Committees in Warsaw (Poland) and the Animal Welfare and Ethical Review Committee (UK), and all procedures were carried out under the guidelines of the Animal (Scientific Procedures) Act 1986. For the experiment animals were housed in individually ventilated cages in the animal facility of Medical University of Warsaw or in the animal facility at Axis Bio, both in SPF conditions. To determine sample size the formula with sample size estimation with two means was used (https://eda.nc3rs.org.uk/experimental-design-group). Controls and experimental animals were housed together to ensure scientific robustness. The mice were housed in individually ventilated cages (IVCs), with up to five per cage, and were identified by tail marks. All animals had free access to a standard certified commercial diet and sanitized water, with the holding room maintained under standard conditions at 20–24 °C, 40–70% humidity, and a 12-h light/dark cycle. Animals are allocated to treatment groups non-randomly but using tumor volume to ensure an equal spread of tumor volume throughout all treatment groups. Tumor growth was monitored by BLI measurement or using electronic calipers. BLI measurements were done 1- or 2-times per week. If the tumor BLI readout is more than $1 \times 10^7$ AU, the mouse would be euthanized. The studies were finished before BLI levels reached human endpoint criteria. The maximal volume of tumor allowed is 1700 mm³. Animals have tumors measured three times per week (Monday, Wednesday, and Friday) using electronic calipers. Tumors were measured in 3 dimensions ($x, y$, and $z$) and used the formula for the volume of a sphere to calculate tumor volume. Mice were euthanized by cervical dislocation or by increasing concentration of $CO_2$.

**Statistical analysis**
The statistical analysis was conducted using Prism version 9.5.1 software (GraphPad Software, California, USA). The one-way ANOVA and Tukey honestly significant difference (HSD) post-hoc test, Dunnett's test, and t-test were applied as well as regression analysis, when appropriate. Survival analysis was conducted using the 'survival' package in R. The Kaplan–Meier curves were generated using the 'survminer' package. To compare survival across different groups, the

pairwise log-rank tests were applied with Bonferroni–Hochberg method used for *p*-value adjustment. The *p*-value ≤ 0.05 (*) was regarded as significant whereas values ≤ 0.01 (**), ≤ 0.001 (***) and ≤ 0.0001 (****) as highly significant. The data are expressed as means ± standard error of the mean (SEM) unless otherwise stated.

## Reporting summary

Further information on research design is available in the Nature Portfolio Reporting Summary linked to this article.

## Data availability

All data are included in the Supplementary Information or available from the authors, as are unique reagents used in this Article. The raw numbers for charts and graphs are available in the Source Data file whenever possible. Source data are provided with this paper.

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

## Acknowledgements

This research is funded by Cellis AG, European Research Council Starting Grant McHAP: 715048 (M.Krol), European Research Council Proof of Concept Grant TROJAN: 956900 (M.Krol), grant from the National Science Centre in Poland no. 2015/18/E/NZ6/00642 (M.Krol), Science development fund of the Warsaw University of Life Sciences and the PNRR M4C2-Investimento 1.4-CN00000041"finanziato dall'Unione europea—NextGenerationEU" (AB). The publication was cofinanced by Science development fund of the Warsaw University of Life Sciences - SGGW. The authors thank Prof. Jeffrey W. Pollard for his help with iPSC technology and the mentoring provided to M.Krol. The authors thank Dr. Magdalena Bederska-Blaszczyk and Dr. Elzbieta Rozanska from the Department of Botany, Institute of Biology, Warsaw University of Life Sciences for granting access to confocal microscopes. The authors thank Dr. Konrad Gabrusiewicz and Dr. Aleksandra Szulc from Cellis for the experimental support. The authors thank Dr. Agata Braniewska, Dr. Zofia Pilch, Dr. Zuzanna Sas, Dr. Katarzyna Tonecka from the Medical

University of Warsaw for the experimental support. The authors thank Axis Bio Services Ltd. for some in vivo experiments and Ryan Hutchinson for the histopathology examination of tissues after s.c. injection of MDC-735. The authors thank Dr. Aneta Jonczy and Dr. Katarzyna Mleczko-Sanecka from the International Institute of Molecular and Cell Biology for their experimental support with mouse hepatocytes. The authors thank Dr. Michal Godlewski from the Institute of Veterinary, Warsaw University of Life Sciences for his help in recording videos used as Supplementary Material. The authors thank Natalia Kubisa and Jakub Nowak from NanoTemper Technologies for their technical assistance with kinetic measurements performed using Monolith X and help with spectral shift data analysis and interpretation.

## Author contributions

M.Krol, T.P.R., and A.B. conceived and designed the project and were inventors of the technology. B.T., M.B., M.K., I.M., P.K., M.Krol, T.P.R. led experimental design and execution. M.G., O.O., D.K., D.S., K.B., M.S., K.B., J.N., W.L., Ł.K., J.B., E.G., A.S., I.P., M.S., J.G. performed the experiments. R.K. performed histopathological examinations. Ł.Krzeminski, G.P., R.P. performed kinetic analysis. J.K., K.W., M.Kisiala performed ferritin purifications. L.C., L.M.F. provided iPSCs cells and technology. L.B. and T.W. provided oncology expertise. B.T., M.B., M.K., P.K., M.Krol wrote the manuscript. B.T., A.B., P.K., T.P.R., M.Krol, provided supervision and mentoring. A.B., T.P.R., and M.Krol provided funding.

## Competing interests

A.B., T.P.R., and M.Krol hold equity in Cellis AG. B.T., M.B., M.K., I.M., A.B., P.K., T.P.R., and M.Krol are inventors of intellectual property related to this work (patent application numbers: PCT/EP2016/064484, PCT/EP2016/064483, PCT/PL2016/050057, 17/931,576, 23204741.5). B.T., M.B., M.K., I.M., M.G., O.O., D.K., D.S., K.B., M.S., J.N., W.L., J.B., E.G., M.S., L.B., P.K., T.P.R., and M.Krol were or are employees of Cellis. Cellis is a company pursuing the commercial development of this technology. There are no conflicts with this current research. The remaining authors declare no competing interests.

## Additional information

[1]Cellis AG, Zurich, Switzerland. [2]Center of Cellular Immunotherapies, Warsaw University of Life Sciences, Warsaw, Poland. [3]Department of Immunology, Mossakowski Medical Research Institute, Polish Academy of Sciences, Warsaw, Poland. [4]Institute of Veterinary Pathology, Free University of Berlin, Berlin, Germany. [5]Biosens Labs Sp. z o.o, Warsaw, Poland. [6]The International Institute of Molecular Mechanisms and Machines, Polish Academy of Sciences, Warsaw, Poland. [7]Department of Chemistry, Biological and Chemical Research Centre, University of Warsaw, Warsaw, Poland. [8]Department of Biochemical Sciences "Alessandro Rossi Fanelli", Sapienza University of Rome, Rome, Italy. [9]Center of Life Nano and Neuro Science, Institute of Italian Technology, Rome, Italy. [10]MRC Centre for Reproductive Health, Queen Medical Research Institute, University of Edinburgh, Edinburgh, UK. [11]Centre for Regenerative Medicine, University of Edinburgh, Edinburgh, UK. [12]Faculty of Medical and Health Sciences, Siedlce University of Natural Sciences and Humanities, Siedlce, Poland. [13]Department of Neurology, Clinical Neuroscience Center, University Hospital and University of Zurich, Zurich, Switzerland. [14]These authors contributed equally: Bartlomiej Taciak, Maciej Bialasek, Malgorzata Kubiak. ✉e-mail: paulina_kucharzewska_siembieda@sggw.edu.pl; t.rygiel@cellis.eu; m.krol@cellis.eu

