## [Transparent Peer Review file · Nature Communications]

Harnessing Macrophage-Drug Conjugates for Allogeneic Cell-Based Therapy of Solid Tumors via the TRAIN Mechanism

Corresponding Author: Professor Magdalena Krol

Version 0:

Reviewer comments:

Reviewer #1

(Remarks to the Author)

Nature

In this ms, Taciak et al explore a new method where macrophages are loaded with ferritin-drug complexes, termed Macrophage-Drug Conjugates (MDCs). Ferritin acts as an efficient drug carrier, and macrophages can internalize large amounts of it. The authors describe a novel process called "TRAnSfer of Iron-binding protein" (TRAIN), where drug-loaded macrophages transfer ferritin to cancer cells through direct contact and the formation of an immune synapse-like structure. Various anti-cancer drugs were encapsulated in ferritin, loaded into macrophages, and tested in vitro. In a mouse model of human pancreatic cancer, MDCs loaded with a kinase inhibitor significantly increased survival.

To demonstrate the translational potential of their approach, the authors chose an in vivo model of pancreatic cancer (PDAC), which is a lethal disease in great need for better therapeutic strategies. The tumor microenvironment (TME) is a highly dominant feature in PDAC and considered a major factor driving therapy resistance. Macrophages are a dominant compartment in the TME and drive immunosuppression. The findings of this study suggest a promising cell-based approach for treating solid tumors, leveraging macrophages and ferritin for targeted drug delivery.

Overall, this is a timely and innovative study with an interesting new approach. However, as described below, the data in my view not sufficiently support all statements of the authors and the in vivo part is not the most appropriate one to decipher the potential value of this approach in pancreatic cancer.

Main issues are:

- Regarding the proposed transfer mechanism (called TRAIN), it remains unclear to me why and how this is a preferred transfer to cancer (vs. non-malignant) cells as the authors did not extensively test this. Do non-malignant cells take up HfT as described for the cancer cells? Do they exchange HfT with macrophages? This could be especially relevant regarding toxicity for liver and colon cells (e.g. organoids) and should be addressed.

- Fig. 2: to better understand the selectivity of the TRAIN process, the authors may consider competition experiments of their labeled HfT with ferritin and BSA for uptake.

- The authors nicely address several potential mechanisms of HfT intake and identify ICAM-1 on macrophages as a lead, though the experimental data (Fig. 4i) appears not very strong (difference moderate, low sample number (which cells, how many biological replicates, etc). I think this should be evaluated in more and distinct macrophage – cancer/non-malignant cell co-culture experiments to support a strong statement on the role of ICAM-1 in this process. It may also help narrowing down potential ligands.

- I do not understand why they chose BxPC3, which is KRAS WT, only representing minority of PDAC cases; and also, most importantly, KRAS mutations contribute to a great extent to shaping TME, e.g. it regulates CXCR2 expression, which is critical for monocyte migration. For new targeting approaches, cell-line based xenograft models are not an appropriate proxy to evaluate targeting efficacy. Thus, a variety of primary cell lines (e.g. from KPC models but also from patients) should be evaluated in orthotopically transplanted tumor models. In addition, organoid-based co-culturing models may be of interest and could support the translational value of this approach.

- There is no characterization on the tumors after treatment. It is largely unclear how well the macrophages can infiltrate and get cell contact with tumor cells, how well are the drugs delivered and exert their effects, how does the tumor and TME

respond accordingly. How is the stromal content of the model? If it lacks stroma (which I believe so), the MDC may work well because there's not much physical barrier, but it's not reflecting the reality.

- Overall, the survival benefit is only modest. Why is this so? Resistance or lack of sufficient delivery or others?
- The authors state that this strategy leverages the properties of M1 macrophages for infiltrating tumor tissues, and their in vitro characterization also shows their macrophages carry M1 phenotypes. However, in PDAC M1 macrophages are typically rare, so I am not sure how exactly the infiltration works? Also, though PDAC is highly infiltrated with TAMs, a substantial portion of TAMs are actually tissue-residential macrophages, so in my view the assumption that monocyte/macrophage migration is strongly or actively occurring in PDAC and therefore the MDC should work needs further experimental evidence.

Reviewer #2

(Remarks to the Author)

The authors developed a potential cell-based method to solid tumor therapy, using the natural properties of macrophages and ferritin for targeted drug delivery. It is an interesting finding, which offers a promising advancement in the therapeutic landscape for solid tumors. There are several points that need to be improved in this manuscript.

Major comments:

1. In page 4, the author used fluorescently labeled HfT (HfT-AF488) to investigate the uptake of HfT by different leukocytes, however, how did the authors label AF488 to the HfT protein, and can the fluorescent molecules of the labeled HfT protein be released?
2. In page 4, the author claimed that the uptake of HfT by macrophage is "an energy-dependent endocytosis process rather than passive membrane passage", which is less rigorous, because cell function is affected at low temperatures.
3. How did the authors rule out the phagocytosis of macrophages on the tumor cells?
4. In Figure 1g-m, the author concluded the polarization state of macrophages does not impact the efficiency of HfT transfer from macrophage to cancer cells, however, whether overloaded HfT affects the function of macrophages, such as polarization and phagocytosis? And is there any difference in the transfer of HfT protein to cancer cells by different polarized macrophages?
5. In addition to tumor cells, whether macrophages can transfer drug-loaded HfT proteins to other normal cells and produce toxic side effects?
6. In page 14, the author checked the polarization surface markers and found that no significant changes were observed after co-culture with cancer cells. Does the mRNA of these markers also remain unchanged?

Minor comments:

1. The author should add the gene information of HfT.
2. In Figure 4K, the author should label tumor cells and macrophages.

Reviewer #3

(Remarks to the Author)

In their study, Bartłomiej Taciak et al. present a novel approach for solid tumor therapy utilizing macrophage-drug conjugates (MDCs). These MDCs consist of macrophages loaded with ferritin cages encapsulating anti-cancer drugs. The macrophages then migrate towards tumors, where they may potentially transfer the ferritin-drug complexes to cancer cells via a newly discovered mechanism called TRAIN. It is worth noting that the effectiveness of MDCs has been validated in human pancreatic cancer models, demonstrating a significant improvement in survival rates. This approach leverages the inherent properties of macrophages and ferritin for targeted drug delivery, offering a promising new avenue for solid tumor treatment.

In my view, this study was generally well done. However, I would like to make a few comments to improve the quality of this study. Below are four of my comments:

1. It might be beneficial to consider conducting an omics-based analysis, such as RNA-sequencing, to investigate potential changes in the phenotype of macrophages or other immune cells during the generation of MDCs.
2. In Figure 4j, ICAM-1 levels appear to be reduced by more than 80%. However, Figure 4i suggests a more modest decrease of 20-30% in MFIs. While the decrease in MFIs is statistically significant, the magnitude of this change raises questions about the necessity or key role of immune synapse-like structures in HfT transfer. This discrepancy might lead to the conclusion that 70-80% of transfer is independent of immune synapse-like connection. Additional analyses may be necessary to address this issue.
3. The manuscript lacks data from an in vivo cytotoxicity assay. Ideally, this assay should include (1) hematological and

biochemical analysis, and (2) histopathological examination of immune organs. (3) Additionally, a thorough analysis of macrophage-enriched organs such as the lung, liver, and spleen is warranted to assess potential adverse effects or toxicity induced by MDCs.

Version 1:

Reviewer comments:

Reviewer #1

(Remarks to the Author)

Taciak et al have resubmitted and consolidated their previous manuscripts into a single manuscript with considerable improvements that take into account the initial reviewer comments. I would like to thank the authors for their clear and thorough response to the issues raised. The in vivo part and used tumor models has been extended nicely and though the data rather confirm an early anti-tumor effect vs. sustained tumor control, I think this is a valid outcome and supports the authors approach. Thus, while there is certainly a lot to learn about the technology and approach in future studies, I have no further concerns to raise and support this manuscript progressing to publication.

Reviewer #2

(Remarks to the Author)

The authors innovatively found that ferritin-drug complexes on macrophages achieved drug delivery and effectively inhibited tumor progression. The study is novel and interesting, and provides more possibilities for anti-tumor strategies based on macrophages. But there are still many problems to be improved.

1. In the initial screening, the authors adopted whether a variety of immune cells were all derived from humans, including CD4+ or CD8+ T lymphocytes, and monocytes. In addition, the authors found that Class A Scavenger Receptors are Responsible for HfT Uptake by Macrophages. Is the expression of Class A Scavenger in CD4+ or CD8+ T lymphocytes, and monocytes significantly lower than that in macrophages?

2. The authors found Macrophages, a major phagocyte, efficiently internalize human h-ferritin (HFT) From medium via endocytic pathway, So do other phagocytes have the same effect, including neutrophils, eosinophils, and so on?

3. Whether HfT has an effect on macrophage itself, including proliferation, apoptosis and polarization?

4. Can macrophages transfer Hft to normal cells in vivo, is its transfer efficiency different from that of tumor cells, and what is the mechanism?

5. How many engineered macrophages reach the tumor site and the efficiency of ferritin-drug complexes that transfer to tumor cells.

6. The paper is very hard to follow. There is far too much text in the results section. A huge amount of data is presented and the authors should really distil down the key findings and present those in the results section in a much more compelling way. The paper therefore needs a major rewrite with a focus on clarity.

7. I'm confused about "How does Hft metastasize to tumor cells through vesicles? The author seems to have only observed the corresponding phenomenon, but the specific mechanism has not been clarified.

Reviewer #3

(Remarks to the Author)

Thank you for your effort and the revisions you have made. While the experiments were not conducted exactly as I initially requested, I believe the studies you performed sufficiently address the questions raised in my previous comments. I also noted some minor errors in your response, such as the mention of Fig. 11 for the histopathology of key organs, when it was actually Fig. 12. However, I was able to locate and confirm the relevant data successfully. I have no further comments or concerns at this time.

Reviewer #4

(Remarks to the Author)

The authors have appropriately addressed my concerns.

23 September 2024

Response to Reviewers' Comments

Manuscripts NCOMMS-24-24862-T and NCOMMS-23-54781-T

Dear Reviewers,

We would like to thank you for your thorough and insightful feedback on our manuscripts. We have carefully considered all the points raised and believe that addressing them has significantly strengthened the work. We have consolidated the two manuscripts into a cohesive study and performed additional experiments as requested, which are detailed below.

Please find our responses to each of your comments, point by point, with references to the revised sections of the manuscript.

NCOMMS-23-54781-T: Investigating the Mechanisms of Heavy-Chain Ferritin Uptake by Macrophages: Implications for Drug Delivery and Immunotherapy

Reviewer #1 (Ferritin)(Remarks to the Author)

This article illuminates Macrophage Scavenger Receptor 1 (MSR1) as the previously undisclosed receptor responsible for HFt uptake in human macrophages. It offers a comprehensive understanding of HFt uptake mechanisms in human macrophages. However, there are several issues that merit attention.

1. In reference to Figure 1 depicting endocytosis-dependent uptake of H-ferritin in human macrophages, it is imperative to incorporate endocytic inhibitors to further elucidate the uptake of H-ferritin by macrophages.

Data presented in Figure 1 shows that macrophages internalize HFt through the active, endocytic process, and not a passive one, as evidenced by the dependence of HFt uptake on the temperature. Additionally, the flow cytometry analysis of HFt-AF488 uptake in the presence of the excess of unlabeled HFt (**Fig. 1 g-h** and **Suppl. Fig. 2b-c**) clearly indicates that human macrophages internalize HFt through the receptor mediated-endocytosis. However, the lack of complete inhibition of HFt uptake in this analysis may suggest that macrophages may partially take up HFt by pinocytosis. To support this hypothesis, we performed additional analyses using pinocytosis inhibitors (EIPA, cytochalasin D), which showed a partial involvement of pinocytosis in the HFt uptake by macrophages (**Fig. 1i-l**,

Suppl. Fig. 2). More comprehensive characterization of various endocytic pathways involved in the HFt uptake in human macrophages are provided in **Fig. 2** and **Suppl. Fig. 3**.

2. Figure 2 illustrates that human macrophages internalize H-ferritin via clathrin-dependent endocytosis. Why is clathrin endocytosis highlighted over other forms such as caveolin-mediated endocytosis (CVME), clathrin-mediated endocytosis (CME), and lipid raft-mediated endocytosis?

We thank the reviewer for this valuable comment. In order to evaluate the role of other forms of endocytosis in HFt uptake in macrophages we performed a series of experiments with inhibitors and siRNA targeting molecules involved in caveolin-mediated endocytosis and lipid raft-mediated endocytosis. Obtained data strongly suggests that the clathrin-mediated endocytosis is a major route through which human macrophages internalize extracellular HFt. This data is presented in **Fig. 2** and **Suppl. Fig. 3**.

3. The efficiency of SiRNA knockdown is insufficient, necessitating additional evidence using CRISPR/Cas to establish cells with greater efficacy.

We agree with the Reviewer that the efficiency of some siRNA-mediated knockdowns (especially of clathrin (CLTC) in THP-1 macrophages; Fig. 2f of the old version of manuscript) was insufficient, therefore we repeated these experiments to improve the quality of this data. We achieved much better silencing results and therefore the conclusions can be clearly defined (**Fig. 2d**).

The use of CRISPR/Cas system to establish cells with better knockdown efficiency is a great suggestion, however this approach has some limitations, which did not allow us to use this method. Generation of cells with full knockout of proteins such as clathrin and TfR1 would probably be difficult due to the fact that these proteins are crucial for proper cell functioning.

4. The basis for cell death is insufficient; more data is required. Furthermore, clarification on the specific type of cell death observed would be beneficial.

This part of the study is focused on the uptake mechanism of HFt in macrophages and not HFt-driven cell death. The viability assay was used to evaluate the toxicity of various endocytosis inhibitors used in the study. Based on this analysis, we used non-toxic concentrations of the endocytosis inhibitors to study the HFt uptake mechanism in macrophages.

The HFt did not cause any death in the macrophages. Once HFt was complexed with the anticancer drug, loaded into macrophages, and transferred to cancer cells, it caused the

death of cancer cells. This phenomenon was described in the second part of the study (now, in the consolidated manuscript).

5. In further validation of these findings, cells were subjected to various concentrations of the dynamin GTPase inhibitor, dynasore, in serum-free medium. Is the GTPase inhibitor the sole inhibitor of cell death under investigation?

Dynasore was used in our studies to evaluate whether HfT is taken up by macrophages through the dynamin-dependent endocytic route. The focus of this experiment was solely on understanding the endocytic route involved in HfT uptake, rather than any processes related to cell death.

6. While TfR1 and SCARA5 are recognized as receptors for HfT, this study delves into macrophage phagocytosis of HfT, while previous research has identified TfR1 and MSR1 as ferritin receptors. Hence, is there a distinction in macrophage phagocytosis between HfT and ferritin? Are there any novel receptors aside from these two? Knocking out the TfR1 receptor is crucial to indicate the dependence of HfT phagocytosis on TfR1.

We thank the reviewer for this valuable comment. We do not have any experimental evidence that HfT and ferritin are taken up by macrophages in a similar way. However, based on our data and currently available studies, we conclude that human macrophages internalize ferritin mainly through TfR1 and MSR1, both of which bind the HfT subunit of this protein. The lack of SCARA5 in macrophages, as evidenced in the literature and in our analysis, indicates that the LfT subunit is not involved in ferritin uptake. Further analysis should be conducted to answer this question. Unfortunately, this type of analysis is beyond the scope of our research. Our Macrophage-Drug Conjugate technology uses only HfT ferritin for drug conjugation/encapsulation, and therefore, we focused on this protein to study the mechanism of action of the technology. We believe that once these two manuscripts are consolidated, it will provide a more comprehensive understanding.

7. Can TfR1-positive macrophages be isolated by flow cytometry to assess their HfT phagocytosis capability?

Based on our data, TfR1 is expressed in all human macrophages suggesting that TfR1 is involved in HfT uptake in all of these cells. Therefore, our data on the role of TfR1 in HfT uptake can be applied to the entire cell population.

8. Which segments of MSR1 are involved in identifying HfT? What mechanism is MSR1 involved in the endocytosis of HfT.

We thank the reviewer for this valuable comment. We showed that deletion of SRCR domain of the MSR1 receptor decreases the uptake of HFt in HEK293 cells transfected with the plasmid carrying mutated version of MSR1 (vs one with a wild type of the receptor). This result can be found in **Fig. 5**.

Reviewer #2 (TAM receptors)(Remarks to the Author):

In this manuscript, the authors primarily explored potential receptors for HFt in macrophages, including TfR1, SR-A and MSR1. These findings provide a more comprehensive understanding of drug delivery oncology therapies based on HFt uptake by macrophages for further exploration. However the significance of these findings are limited. Moreover, I have some concerns regarding the study.

1. Authors should use large-scale unbiased methods to screen potential HFt receptors, such as protein microarray.

We thank the reviewer for this valuable comment. A large-scale screening for potential HFt receptors, such as using protein microarrays, would indeed be very informative and might lead to the discovery of other receptors. However, we believe that this would be more important for the whole ferritin protein and not for the HFt construct and therefore was beyond the scope of our paper. This part of our study aimed to investigate the mode of action of the new Macrophage-Drug Conjugate therapy, and to find the way by which our HFt construct (that is complexed with anticancer drug) enters the macrophage cell. We believe that our study demonstrated that MSR1 is responsible for HFt internalization. This decision was based on existing literature suggesting interactions between Scavenger Receptor Class A members and ferritin, and the fact that MSR1 is a key receptor in macrophages.

2. This article only focuses on identifying the receptors responsible for HFt uptake at the cellular level. Therefore, the authors should further explore the effect of knocking down these receptors in macrophages on the uptake of HFt *in vivo*.

Our therapeutic platform is based on human macrophages that are loaded with the HFt-drug conjugate under the *in vitro* conditions and then administered to live organisms in order to treat solid tumors. We do not aim to target macrophages *in vivo* with solely administered ferritin-drug. We understand that before consolidation of the manuscripts the main goal of this all work was less clear. We believe that once these two manuscripts are consolidated, it will provide a more comprehensive understanding.

3. The authors should further examine whether these receptors affect the uptake of HFt by macrophages to exert their anti-tumor effects in animal models.

We thank the reviewer for this valuable comment. Our therapeutic platform is based on human macrophages that are loaded with the HFt-drug conjugate and then administered into the live animal to treat solid tumors. The effect of the HFt-drug loaded macrophages is demonstrated in the efficacy experiments (**Fig. 9** and **Fig. 10**, **Suppl. Fig. 16** and **Suppl. Fig. 21**). We showed that unloaded macrophages do not cause therapeutic effect (**Fig. 10a-d** and **Suppl. Fig. 21a**). We understand that before consolidation of the manuscripts the main goal of this all work was less clear. We believe that once these two manuscripts are consolidated, it will provide a more comprehensive understanding.

4. The rate of protein degradation is determined by the slope of the degradation curve, so the results presented here (Fig.3d, fig.5f and 5h) do not seem to support accelerated protein degradation.

Data presented in Fig. 3d, Fig 5f and 5h in fact showed the decrease in the amount of MSR1 and TfR1 protein in cells with increasing incubation time with HFT instead of the protein degradation level. We would like to thank the reviewer for this valuable comment and for pointing out the mistake. In a new version of the manuscript the data generated in these experiments is presented as a degradation curve and the shape of the slope supports accelerated protein degradation hypothesis (**Fig. 4f-h** and **Suppl. Fig. 4d-f**).

5. Authors should further explore the specific region or specific amino acid site of MSR1 binding to HFt.

We thank the reviewer for this valuable comment. We showed that deletion of SRCR domain of the MSR1 receptor decreases the uptake of HFt in HEK293 cells transfected with the plasmid carrying mutated version of MSR1 (vs one with a wild type of the receptor). This result can be found in **Fig. 5**.

6. Whether the uptake of HFt by macrophages affects their own polarization. MSR1 is a marker of M2 macrophages, so how to deal with the relationship between M2-type macrophages promoting tumor and inhibiting tumor as drug delivery?

The question of investigating potential changes in macrophage phenotype during MDC generation is indeed important and valid. In the revised manuscript we provided data that HFt loading of macrophages does not significantly impact their phenotype (**Suppl. Fig. 17** and **Suppl. Fig. 18**).

7. Is there a difference in receptor recognition between drug-loaded HFt and non-drug-loaded HFt?

We thank the reviewer for this question. In order to evaluate if there is a difference in receptor recognition between drug-loaded HFt and an empty one we performed a series of in vitro experiments with an excess of the unlabeled, competing ligands (AcLDL, poly(G) and AcLDL) and siRNA targeting MSR-1 in which we analyzed their influence on the uptake of drug-loaded HFt (some of the obtained results are shown below). We also carried out the kinetic analysis using AlphaScreen binding assay using drug-loaded HFt. Obtained data show that MSR1 inhibition in cells blocks drug-loaded HFt uptake and that there is no significant difference in MSR1 recognition between drug-loaded and non-drug-loaded HFt. The results of these additional experiments will be included in the final version of the manuscript (**Fig. 9**).

Reviewer #3 (Ferritin)(Remarks to the Author):

The study titled "Investigating the Mechanisms of Heavy-Chain Ferritin Uptake by Macrophages: Implications for Drug Delivery and Immunotherapy" delves into how macrophages, a type of immune cell, absorb a protein called heavy-chain ferritin (HFt). This research is crucial for developing ways to deliver drugs directly to tumors using macrophages. The authors show that macrophages primarily use a process called endocytosis to take in HFt, with a specific method known as clathrin-dependent endocytosis being the main route. They focus on the role of two receptors, TfR1 and MSR1, in this process, finding that MSR1, in particular, plays a significant role in helping macrophages take in HFt. The study also notes that macrophages that are more inclined to heal (M2 type) are better at taking in HFt due to higher levels of MSR1. This suggests a potential for using these cells in targeted drug delivery, especially for cancer treatment. While the research is detailed and offers promising insights into using macrophages for drug delivery, it would benefit from further experiments, particularly in living organisms, to better understand the practical applications and safety. The study is an important step in understanding how macrophages can be used to target drugs to specific body sites, like tumors. It opens up possibilities for new treatments, especially by targeting the MSR1 receptor, but more research is needed to fully understand and utilize this complex system. The study presents compelling evidence for the role of MSR1 in the uptake of HFt by macrophages, offering exciting prospects for Macrophage-Drug Conjugates (MDC) in targeted therapies. However, the leap from these foundational findings to the broader applications in drug delivery and immunotherapy, as suggested by the title, would be more convincing with the support of in vivo data. Demonstrating the practical therapeutic potential of MSR1-mediated drug delivery in a living system would significantly strengthen the study's impact. Furthermore, a careful revision to correct minor typos would ensure the manuscript's scientific rigor is well-reflected in its presentation.

Thank you for your detailed and insightful comment on our study. We appreciate your recognition of the significance of our findings regarding the uptake of heavy-chain ferritin (HFt) by macrophages and the potential implications for drug delivery and immunotherapy.

We are pleased to inform you that we have consolidated our initial findings with an additional manuscript, which now includes *in vivo* evidence demonstrating the efficacy of our Macrophage-Drug Conjugate (MDC) technology. These combined studies provide a comprehensive understanding of the mechanisms by which HFt is internalized by macrophages and how this process can be harnessed for targeted drug delivery.

The *in vivo* experiments have shown promising results, highlighting the practical therapeutic potential of macrophage-mediated drug delivery in living systems. This new data strengthens our conclusions and underscores the potential of using macrophages for targeted therapies, particularly in the context of cancer treatment.

We agree that demonstrating the therapeutic potential of our technology in living systems significantly enhances the impact of our findings. Furthermore, we will ensure that minor typos are corrected to maintain the manuscript's scientific rigor and clarity.

NCOMMS-24-24862-T: Macrophage-Drug Conjugate (MDC): A Promising Paradigm for Targeted Solid Tumor Therapy

Reviewer #1 (Remarks to the Author):

In their study, Bartłomiej Taciak et al. present a novel approach for solid tumor therapy utilizing macrophage-drug conjugates (MDCs). These MDCs consist of macrophages loaded with ferritin cages encapsulating anti-cancer drugs. The macrophages then migrate towards tumors, where they may potentially transfer the ferritin-drug complexes to cancer cells via a newly discovered mechanism called TRAIN. It is worth noting that the effectiveness of MDCs has been validated in human pancreatic cancer models, demonstrating a significant improvement in survival rates. This approach leverages the inherent properties of macrophages and ferritin for targeted drug delivery, offering a promising new avenue for solid tumor treatment.

In my view, this study was generally well done. However, I would like to make a few comments to improve the quality of this study. Below are four of my comments:

1. It might be beneficial to consider conducting an omics-based analysis, such as RNA-sequencing, to investigate potential changes in the phenotype of macrophages or other immune cells during the generation of MDCs.

The question of investigating potential changes in macrophage phenotype during MDC generation is indeed important and valid. In the revised manuscript we provided data that HFt loading of macrophages does not significantly impact their phenotype (**Suppl. Fig. 17** and **Suppl. Fig. 18**). Moreover, we provided results of flow cytometry analysis of macrophage markers in MDC-735 product after the freezing and thawing at different time-points (**Suppl. Fig. 19**). We present these data to show that this product is stable, can be

efficiently cryopreserved, stored, then thawed - this shows potential to be off-the-shelf therapy in the future.

As the current data does not suggest significant changes in phenotype that could impact the observed MDC function, we believe that a detailed omics-based characterization is beyond the scope of this particular study. However, we recognize its value and may consider it in future research.

2. In Figure 4j, ICAM-1 levels appear to be reduced by more than 80%. However, Figure 4i suggests a more modest decrease of 20-30% in MFIs. While the decrease in MFIs is statistically significant, the magnitude of this change raises questions about the necessity or key role of immune synapse-like structures in HFT transfer. This discrepancy might lead to the conclusion that 70-80% of transfer is independent of Immune synapse-like connection. Additional analyses may be necessary to address this issue.

Thank you for your valuable comment. This conclusion was supported by several experiments other than ICAM-1 knock-down: the dependence of HFt transfer on direct cell-cell contact, the involvement of actin polymerization and actin cytoskeleton reorganization, as shown by flow cytometry imaging, and the role of vesicle exchange, which appears to occur directly between the cells and not through secretion from macrophages to the medium. However, we agree with the reviewer and acknowledge the discrepancy between the data presented in Figures 4j and 4i. Indeed, while Figure 4j shows a significant reduction in ICAM-1 levels, Figure 4i indicates a more modest decrease in MFIs, raising questions about the necessity or key role of ICAM-1 in the immune synapse-like structures for HFt transfer.

To address this, we performed an additional experiment using an ICAM-1 blocking antibody, which has been included in **Figure 8o**. Blocking ICAM-1 on the macrophages during the macrophage-cancer cell co-culture, led to a significant reduction in ferritin transfer. However, the process was not entirely inhibited. Consequently, we have revised our manuscript to de-emphasize the exclusive role of ICAM-1 and highlight the potential involvement of other factors in this process.

3. The manuscript lacks data from an in vivo cytotoxicity assay. Ideally, this assay should include (1) hematological and biochemical analysis, and (2) histopathological examination of immune organs. (3) Additionally, a thorough analysis of macrophage-enriched organs such as the lung, liver, and spleen is warranted to assess potential adverse effects or toxicity induced by MDCs.

Thank you for your valuable comment. We acknowledge the importance of including comprehensive in vivo cytotoxicity assays to evaluate the safety of our Macrophage-Drug Conjugate (MDC) therapy.

To address this concern, we have included: **Fig. 11** showing histopathology of key organs (listed by the reviewer) of mice that received autologous or allogeneic macrophages i.v. and s.c., as well as in vitro safety of MDC-735 for normal healthy cells. Additionally, we included **Suppl. Fig. 20** showing changes in mouse body weight in extensive in vivo efficacy experiments and **Suppl. Fig. 21** showing results of blood analysis and histopathology of key organs of mice that received autologous or allogeneic macrophages i.v. Detailed histopathology images and findings which, all together with mouse body weight results in multiple efficacy studies demonstrate safety of the therapy.

Reviewer #2 (Remarks to the Author):

The authors developed a potential cell-based method to solid tumor therapy, using the natural properties of macrophages and ferritin for targeted drug delivery. It is an interesting finding, which offers a promising advancement in the therapeutic landscape for solid tumors. There are several points that need to be improved in this manuscript.

Major comments:

1. In page 4, the author used fluorescently labeled HFt (HFt-AF488) to investigate the uptake of HFt by different leukocytes, however, how did the authors label AF488 to the HFt protein, and can the fluorescent molecules of the labeled HFt protein be released?

To label HFt with Alexa Fluor 488 (AF488) or any other fluorescent dye, we used Alexa Fluor succinimidyl ester (NHS)-esters) for the conjugation process which is the most popular tool for conjugating this dye to a protein or antibody. The NHS-esters are reactive towards primary amines, which are typically found on lysine residues of proteins. This reaction forms a stable covalent bond, ensuring that the AF488 fluorophore is permanently attached to the HFt protein. Due to the covalent nature of this bond, the fluorescent molecules of the labeled HFt protein should not be released or detached within the cellular environment.

To further substantiate the stability and conjugation of the labeled HFt, we present data demonstrating the co-localization of an antibody specifically targeting HFt with the HFt-AF488 conjugate following HFt-AF488 transfer by RAW264.7 macrophages to cancer cells (**Suppl. Fig. 1f**). The observed co-localization confirms that the fluorescent signal corresponds to the HFt protein, indicating that both the antibody and the AF488 fluorophore are indeed bound to HFt. The single stained ferritin is a natural protein present in the cells.

Also, our study focused on the stability of HFt-drug conjugates. We demonstrated this stability through the co-localization of an anti-HFt antibody with an anti-drug antibody. This co-localization indicates that both the HFt protein and the conjugated drug remain stably associated both in macrophages and in cancer cells following transfer after 24 h co-culture (**Fig. 9l**).

2. In page 4, the author claimed that the uptake of HFt by macrophage is “an energy-dependent endocytosis process rather than passive membrane passage”, which is less rigorous, because cell function is affected at low temperatures.

The mechanism of HFt uptake by macrophages is now described in detail in the first part of the revised manuscript, which is a consolidated version of two original manuscripts. Initially, we submitted two separate manuscripts: one focused on the mechanism of ferritin uptake and the other on the new technology that uses macrophages loaded with ferritin-drug conjugates to treat solid tumors. As a result, the original manuscript did not emphasize the mechanism of ferritin uptake.

With the consolidation of these manuscripts, we believe that the revised version provides more comprehensive information, making it easier to follow and understand the technology as a whole.

However, to answer the reviewer's point, to clarify this part, we would rephrase our statement to point out that in this experiment we differentiate between all active cellular uptake mechanisms and passive diffusion. We conducted the uptake experiments at both low and physiological temperatures (4°C and 37°C) to investigate the internalization mechanism of HFt. It is well-established in the literature that reduced incubation temperature retards all active cellular uptake mechanisms, including endocytosis, while diffusion (passive uptake) is less affected, though it is also reduced to some extent (<https://doi.org/10.1021/acsbiomaterials.0c01639>). Our data showed significantly reduced uptake of HFt at 4°C compared to 37°C, consistent with the inhibition of active uptake processes at lower temperatures. This observation supports the conclusion that HFt uptake involves active cellular mechanisms rather than passive diffusion.

3. How did the authors rule out the phagocytosis of macrophages on the tumor cells?

We understand the reviewer's concern regarding the potential phagocytosis of tumor cells by macrophages, which could confound the interpretation of the HFt transfer. We elaborated more about this in the revised version of the manuscript. To ensure that we observe the transfer of HFt protein between cells and not the phagocytosis of cancer cells by macrophages, we provide additional experiment data as described below.

To distinguish macrophages from cancer cells in our co-culture experiments, we labeled each cell type with distinct fluorescent dyes. This approach allowed us to accurately identify and separate macrophages and cancer cells during FACS analysis. This approach allowed us to accurately identify and separate macrophages and cancer cells during FACS analysis. In the analysis, we observed that the majority of cancer cells were positive for the AF488 signal, while only a small fraction of macrophages was positive for CellTrace, which was used for cancer cell labeling. This result suggests that we are observing the transfer of HFt-AF488 rather than a significant number of macrophages phagocytosing cancer cells (**Suppl. Fig. 10a,b**).

Also, the confocal microscopy images shown in **Fig. 6b,f,j**, **Suppl. Fig. 8a,b**, **Suppl. Fig. 11** provide visual evidence that macrophages and cancer cells remained separated in the co-culture. The images clearly demonstrated that the HFt signal is localized within the cancer cells and not within macrophages.

Additionally, the flow cytometry imaging data presented in **Fig. 8a,g,h** further corroborates our findings. The imaging confirms that the HFt signal is associated with cancer cells,

indicating that the HFt protein transfer occurs to the cancer cells. Moreover, time-lapse microscopy imaging videos attached to the manuscript demonstrate that ferritin is transferred from macrophages to cancer cells in the form of small vesicles (**Suppl. Video 1,2,3**).

This can be also observed on the holotomographic microscopy analysis (**Suppl. Fig. 11a**). The process of HFt transport from one cell to another could be observed on a series of captured images. However, the main limitation of this microscope that is the low resolution of the fluorescence imaging module did not allow for the identification of the cellular structures involved in this mechanism.

4. In Figure 1g-m, the author concluded the polarization state of macrophages does not impact the efficiency of HFt transfer from macrophage to cancer cells, however, whether overladed HFt affect the function of macrophages, such as polarization and phagocytosis? And is there any difference in the transfer of HFt protein to cancer cells by different polarized macrophages?

In primary macrophages obtained from different buffy coat donors, the data shown in Figure below (**Fig. 6g**) indicated no significant differences in HFt transfer efficiency between M1 and M2 polarized macrophages. The variations in HFt transfer observed were greater among cells obtained from different donors than between differently polarized macrophages, suggesting that individual donor traits have a more substantial impact on HFt transfer efficiency than the polarization status.

Additionally, data provided in the answer to the question above indicate that HFt loading did not induce significant phagocytosis of cancer cells in co-culture, as evidenced by the low number of double-positive (macrophage dye and cancer cell dye) macrophages observed.

Moreover, another experiment involving a 24-hour co-culture of macrophages and cancer cells demonstrated that loading macrophages with the FT-drug conjugate neither induced significant phagocytosis nor altered their polarization. This is evidenced by the low number of double-positive macrophages (positive for both macrophage and cancer cell dyes) and the consistent expression of major M1 phenotype markers across the different macrophage groups (**Suppl. Fig. 10c**).

Regarding the question about how the HFt loading influences macrophage phenotype, in the revised manuscript we provided data that HFt loading of macrophages does not significantly impact their phenotype (**Suppl. Fig. 17** and **Suppl. Fig. 18**).

5. In addition to tumor cells, whether macrophages can transfer drug-loaded HFt proteins to other normal cells and produce toxic side effects?

Thank you for your question regarding the potential implications for non-malignant cells, particularly concerning safety and toxicity. To address these concerns, we conducted a series of experiments to assess whether MDC-735 could be toxic to non-malignant cells. The results are presented in **Fig. 11**. Specifically, the viability of several normal cell types—mouse hepatocytes (**Fig. 11c**), and human: dermal microvascular endothelial cells, bladder

fibroblasts, renal epithelial cells, brain microvascular cells, and lung fibroblasts (**Fig. 11d**) were tested in co-culture with MDC-735. The analysis revealed that malignant cells were significantly more sensitive to MDC-735 treatment than normal cells. This selective cytotoxicity is consistent with our in vivo results, showing that the MDC-735 therapy is safe for animals and does not adversely affect healthy tissues (**Fig. 11a,b**). These results suggest that the TRAIN mechanism may indeed favor cancer cells over non-malignant cells, potentially reducing the risk of toxicity in normal tissues. We believe these findings support the safety profile of our approach and provide reassurance regarding its selectivity.

6. In page 14, the author checked the polarization surface markers and found that no significant changes were observed after co-culture with cancer cells. Does the mRNA of these markers also remain unchanged?

We thank the reviewer for this comment. In response, we have performed additional RT-PCR experiment to analyze the expression of a series of polarization marker genes. While HfT loading of macrophages resulted in minor changes in the expression of some markers, the majority of genes showed no significant alterations, and most changes observed were not statistically significant. Additionally, we provided further data from flow cytometry surface marker analysis, which supports these findings. Together, this data confirms that HfT loading does not significantly impact macrophage phenotype at either the mRNA or surface marker levels (**Suppl. Fig. 17** and **Suppl. Fig. 18**).

7. Minor comments:

1. The author should add the gene information of HfT.
2. In Figure 4K, the author should label tumor cells and macrophages.

Thank you for your comments. We have made the following, suggested corrections:

1. We have added the gene information of HfT to the manuscript (Materials and Methods section) to provide clarity and context regarding the protein used in our studies.
2. We have labeled the tumor cells and macrophages in the mentioned figure to ensure clear identification and improve the interpretability of the figure.

Reviewer #3 (Remarks to the Author):

In this ms, Taciak et al explore a new method where macrophages are loaded with ferritin-drug complexes, termed Macrophage-Drug Conjugates (MDCs). Ferritin acts as an efficient drug carrier, and macrophages can internalize large amounts of it. The authors describe a novel process called "TRANsfer of Iron-binding protein" (TRAIN), where drug-loaded macrophages transfer ferritin to cancer cells through direct contact and the formation of an immune synapse-like structure. Various anti-cancer drugs were encapsulated in ferritin, loaded into macrophages, and tested in vitro. In a mouse model of human pancreatic cancer, MDCs loaded with a kinase inhibitor significantly increased survival.

To demonstrate the translational potential of their approach, the authors chose an in vivo model of pancreatic cancer (PDAC), which is a lethal disease in great need for better therapeutic strategies. The tumor microenvironment (TME) is a highly dominant feature in PDAC and considered a major factor driving therapy resistance. Macrophages are a dominant compartment in the TME and drive immunosuppression. The findings of this study suggest a promising cell-based approach for treating solid tumors, leveraging macrophages and ferritin for targeted drug delivery.

Overall, this is a timely and innovative study with an interesting new approach. However, as described below, the data in my view not sufficiently support all statements of the authors and the in vivo part is not the most appropriate one to decipher the potential value of this approach in pancreatic cancer.

Main issues are:

1. Regarding the proposed transfer mechanism (called TRAIN), it remains unclear to me why and how this is a preferred transfer to cancer (vs. non-malignant) cells as the authors did not extensively test this. Do non-malignant cells take up HfT as described for the cancer cells? Do they exchange HfT with macrophages? This could be especially relevant regarding toxicity for liver and colon cells (e.g. organoids) and should be addressed.

Thank you for your question regarding the potential implications for non-malignant cells, particularly concerning safety and toxicity. To address these concerns, we conducted a series of experiments to assess whether MDC-735 could be toxic to non-malignant cells. The results are presented in **Fig. 11**. Specifically, the viability of several normal cell types—mouse hepatocytes (**Fig. 11c**), and human: dermal microvascular endothelial cells, bladder fibroblasts, renal epithelial cells, brain microvascular cells, and lung fibroblasts (**Fig. 11d**) were tested in co-culture with MDC-735. The analysis revealed that malignant cells were significantly more sensitive to MDC-735 treatment than normal cells. This selective cytotoxicity is consistent with our in vivo results, showing that the MDC-735 therapy is safe for animals and does not adversely affect healthy tissues (**Fig. 11a,b**). These results suggest that the TRAIN mechanism may indeed favor cancer cells over non-malignant cells, potentially reducing the risk of toxicity in normal tissues. We believe these findings support the safety profile of our approach and provide reassurance regarding its selectivity.

To address the second part of your question whether macrophages exchange ferritin with other cells, we conducted a series of experiments to determine whether recipient cells, once they receive HfT from macrophages, can further transfer HfT to other cells. This is included in Suppl. **Fig. 9**. This was done through a two-step co-culture experiment.

Experiment Design:

- In the first step, RAW264.7 macrophages loaded with HfT were co-cultured with labeled recipient cells (1st level) for 24 hours. These recipient cells were then separated using a cell sorter.

- In the second step, these recipient cells were used as donor cells and co-cultured with EMT6 cancer cells labeled with a different fluorescent dye (2nd level recipient cells) for another 24 hours. The transfer of HFt was assessed using flow cytometry.

Key findings:

1. No Significant Secondary Transfer: When EMT6 cancer cells were used as the 1st level recipient cells, HFt transfer was confirmed in the first step. However, when these EMT6 cells were co-cultured with other EMT6 cells in the second step, no significant increase in HFt fluorescence was detected in the 2nd level recipient cells. This suggests that secondary HFt transfer between cancer cells does not occur.
2. Cancer Cells as HFt Carriers: It is known that cancer cells can also uptake ferritin, as reported by Geninatti Crich et al. (2015). We tested whether EMT6 cancer cells carrying HFt could transfer it to other EMT6 cancer cells or RAW264.7 macrophages. Although the EMT6 cells efficiently took up HFt, the fluorescence intensity in recipient cells was very low, suggesting that significant HFt transfer from EMT6 cells to other cells does not occur.

These results suggest that HFt transfer via the TRAIN mechanism has limited secondary transfer. This indicates reduced potential off-target effects in non-malignant cells, as the recipient cells do not pass ferritin-drug to other cells.

We hope this explanation clarifies the selectivity of the TRAIN mechanism and addresses your concerns regarding the potential impact on non-malignant cells.

2. Fig. 2: to better understand the selectivity of the TRAIN process, the authors may consider competition experiments of their labeled HFt with ferritin and BSA for uptake.

Thank you for your valuable comment. We appreciate the suggestion to perform competition experiments to better understand the selectivity of the TRAIN process. Results of this experiment are presented in **Suppl. Fig. 8e,f**. The competition experiment demonstrated that the transfer of BSA was significantly reduced when cells were co-loaded with HFt, compared to cells loaded with BSA alone (**Suppl. Fig. 8e**). In contrast, the presence of BSA did not affect HFt transfer (**Suppl. Fig. 8f**). These findings suggest that HFt transfer is not significantly impacted by the presence of BSA, while HFt inhibits BSA transfer. Moreover, we have generated data comparing the anti-tumor efficacy of macrophages loaded with ferritin-735 drug versus BSA-735 drug. As shown in the **Suppl. Fig. 13c**, the macrophages loaded with ferritin-735 demonstrate significantly higher anti-tumor efficacy, highlighting the importance of using this specific complex. These results will also be included in the revised manuscript to further support our findings.

Thank you again for your insightful suggestion, which has helped us strengthen our understanding and presentation of the selectivity of the TRAIN process.

3. The authors nicely address several potential mechanisms of HFt intake and identify ICAM-1 on macrophages as a lead, though the experimental data (Fig. 4i) appears not very strong (difference moderate, low sample number (which cells, how many biological replicates, etc). I think this should be evaluated in more and distinct macrophage – cancer/non-malignant cell co-culture experiments to support a strong statement on the role of ICAM-1 in this process. It may also help narrowing down potential ligands.

Thank you for your valuable comment. We agree with the reviewer and acknowledge the modest effect of ICAM-1 silencing on TRAIN efficacy, raising questions about the necessity or key role of ICAM-1 in immune synapse-like structures for HFt transfer.

To address this, we performed an additional experiment using an ICAM-1 blocking antibody, which has been included in **Figure 8o**. Blocking ICAM-1 on the macrophages during the macrophage-cancer cell co-culture, led to a significant reduction in ferritin transfer. However, the process was not entirely inhibited. Consequently, we have revised our manuscript to de-emphasize the exclusive role of ICAM-1 and highlight the potential involvement of other components in this process.

4. I do not understand why they chose BxPC3, which is KRAS WT, only representing minority of PDAC cases; and also, most importantly, KRAS mutations contribute to a great extent to shaping TME, e.g. it regulates CXCR2 expression, which is critical for monocyte migration. For new targeting approaches, cell-line based xenograft models are not an appropriate proxy to evaluate targeting efficacy. Thus, a variety of primary cell lines (e.g. from KPC models but also from patients) should be evaluated in orthotopically transplanted tumor models. In addition, organoid-based co-culturing models may be of interest and could support the translational value of this approach.

We appreciate the reviewer's concern regarding the choice of the BxPC3 cell line, which represents a minority of PDAC cases due to its KRAS wild-type status. We understand that KRAS mutations significantly influence the tumor microenvironment (TME), particularly in regulating CXCR2 expression, which is critical for monocyte migration.

Our decision to use the BxPC3 cell line in our efficacy study was not based on the TME composition of this tumor type or the specific patient population it represents, but rather on the molecular characteristics of the cell line. The BxPC3 cell line was selected for its "stem cell-like" properties and its expression of the EGFR/ERK1/2/AKT/mTOR signaling pathway (<https://www.cytion.com/Knowledge-Hub/Cell-Line-Insights/BXPC-3-Cell-Line-Illuminating-Pancreatic-Cancer-Research/>, <https://www.cytion.com/Knowledge-Hub/Cell-Line-Insights/BXPC-3-Cell-Line-Illuminating-Pancreatic-Cancer-Research/>, <https://linkinghub.elsevier.com/retrieve/pii/S2451945617303549>, <https://aacrjournals.org/mct/article/16/6/1041/332356/Combinatorial-Screening-of-Pancreatic>, <https://pubmed.ncbi.nlm.nih.gov/16373709/>), which is relevant to the drug delivered by the MDC-250 macrophage product. These characteristics made it a suitable model for investigating the efficacy of our targeted therapy.

To demonstrate the broader applicability of the MDC platform, we have included in the new version of the manuscript additional *in vivo* data using a different product, MDC-735. Unlike MDC-250, which targets specific pathways, MDC-735 contains a cytotoxic drug, making it a potentially universal tool for targeting various solid tumors. We have tested MDC-735 and showed its efficacy in several models, including lung metastasis of breast cancer, bladder cancer, head and neck cancer, ovarian cancer, and pancreatic cancer. All these efficacy results are now included in **Fig. 9** and **Fig. 10**.

We acknowledge the importance of evaluating our approach using a variety of primary cell lines, including those derived from KPC models and patients, in orthotopically transplanted tumor models. Additionally, we recognize the potential value of organoid-based co-culturing models to support the translational relevance of our approach, and we will consider these methodologies in future studies.

5. There is no characterization on the tumors after treatment. It is largely unclear how well the macrophages can infiltrate and get cell contact with tumor cells, how well are the drugs delivered and exert their effects, how does the tumor and TME respond accordingly. How is the stromal content of the model? If it lacks stroma (which I believe so), the MDC may work well because there's not much physical barrier, but it's not reflecting the reality.

Thank you for raising this valid point. We agree that *in vivo* characterization of MDC interaction with the tumor microenvironment (TME) and the resulting changes in the tumor post-treatment is important. While the model used in previous version of the manuscript may have less stromal content, which could reduce physical barriers and enhance the efficacy of MDC therapy, our findings suggest that even in this context, the treatment demonstrates significant efficacy in several other tumor models. In this revision, we have included efficacy studies from various other models (**Fig. 9** and **Fig. 10**), demonstrating that the observed treatment efficacy is not model-specific. Additionally, we have included data from tissue sections showing MDC infiltration into the tumor and the corresponding tumor changes following MDC injection (**Suppl. Fig. 20a**). More detailed characterization of TME changes following MDC treatment is planned for future studies.

Regarding tumor tissue characterization after treatment, we conducted a detailed analysis using U87MG tumors (**Suppl. Fig. 20b,c**). We locally treated these tumors with MDC-735 and then performed histopathological analysis at 3-, 7-, and 30- days post-administration. Microscopic examination of H&E-stained sections from MDC-735 treated animals showed clear evidence of treatment-related regression, with only small areas of tumor tissue remaining, if any. Notably, 14 out of 18 tumor-bearing animals treated with MDC-735 had no detectable tumor at the end of the study.

Further analysis included CD68 staining, which showed extensive CD68-positive macrophages throughout the tumor tissue, particularly dense in PBS control animals. Similarly, the tumor/stroma ratio was significantly higher in PBS controls compared to MDC-735 treated animals, indicating a more effective treatment response in the latter group.

Additionally, Ki67 staining performed 30 days after MDC-735 administration revealed a much higher proportion of proliferating tumor cells in the PBS control group compared to the MDC-735 treated group, further indicating that the treatment effectively killed or controlled the growth of U87MG tumor cells. Because the majority of mice in the MDC-735 treated group were cured, only one animal per group (those still having detectable tumor) was taken for immunohistochemical analysis.

6. Overall, the survival benefit is only modest. Why is this so? Resistance or lack of sufficient delivery or others?

Thank you for your comment. We would like to clarify that while the overall survival benefit observed in case of MDC-250 in BxPC3 model may appear modest in monotherapy settings, we were able to achieve a long-term survival in 50% of the animals when our Macrophage-Drug Conjugate (MDC) therapy was used in combination with another treatment. However, in the current version of the manuscript we present more efficacy data of the MDC technology (**Fig. 9** and **Fig. 10**). With the use of the other product (MDC-735) we demonstrated high efficacy across multiple tumor models, when given as monotherapy or in combination therapy (for example with check-point inhibitor). This outcome highlights the potential of our approach. The results suggest that while the standalone MDC therapy is effective, in particularly difficult-to-treat tumors, its full potential may be realized when combined with other treatments. We are encouraged by these findings and are continuing to explore combination strategies to further improve therapeutic outcomes. Therefore, we proceed for First in Human trial in Q2 2025.

7. The authors state that this strategy leverages the properties of M1 macrophages for infiltrating tumor tissues, and their in vitro characterization also shows their macrophages carry M1 phenotypes. However, in PDAC M1 macrophages are typically rare, so I am not sure how exactly the infiltration works? Also, though PDAC is highly infiltrated with TAMs, a substantial portion of TAMs are actually tissue-residential macrophages, so in my view the assumption that monocyte/macrophage migration is strongly or actively occurring in PDAC and therefore the MDC should work needs further experimental evidence.

Thank you for your thoughtful question. We acknowledge the complexity of the tumor microenvironment in pancreatic ductal adenocarcinoma (PDAC), where M1 macrophages are indeed less prevalent, and a significant portion of tumor-associated macrophages (TAMs) are tissue-resident.

Our strategy does not solely rely on the infiltration of M1 macrophages in PDAC. While the in vitro characterization shows that our macrophages exhibit rather M1 phenotypes, the efficacy of our Macrophage-Drug Conjugate (MDC) therapy has been demonstrated across multiple tumor models, not limited to PDAC (which is specific tumor type). This data is

presented in **Fig. 9** and **Fig. 10**. These models include various types of solid tumors, where we have consistently observed effective therapeutic outcomes.

We recognize that further experimental evidence may be needed to fully understand the dynamics of macrophage infiltration in PDAC specifically. However, the broader applicability and success of our MDC therapy in diverse tumor environments suggests that it has significant potential beyond the limitations posed by the unique microenvironment of PDAC.

Sincerely,

Prof. Magdalena Król, PhD

Warsaw University of
Life Sciences

Center of Cellular
Immunotherapies

Ciszewskiego 8 Str.
02-786 Warsaw
Poland
cik@sggw.edu.pl
cik.sggw.edu.pl

Warsaw, 19 Nov 2024

Thank you for your feedback on manuscript **NCOMMS-24-24862A**. We greatly appreciate your comments. Below we provide the detailed answers to the Reviewers.

Responses to Reviewers' Comments:

Reviewer #1 (Remarks to the Author):

Taciak et al have resubmitted and consolidated their previous manuscripts into a single manuscript with considerable improvements that take into account the initial reviewer comments. I would like to thank the authors for their clear and thorough response to the issues raised. The in vivo part and used tumor models has been extended nicely and though the data rather confirm an early anti-tumor effect vs. sustained tumor control, I think this is a valid outcome and supports the authors approach. Thus, while there is certainly a lot to learn about the technology and approach in future studies, I have no further concerns to raise and support this manuscript progressing to publication.

Thank you for your kind and supportive comments. We're glad that the improvements and extended in vivo data addressed your concerns. Your feedback has been invaluable, and we appreciate your support in moving this manuscript towards publication.

Reviewer #2 (Remarks to the Author):

The authors innovatively found that ferritin-drug complexes on macrophages achieved drug delivery and effectively inhibited tumor progression. The study is novel and interesting, and provides more possibilities for anti-tumor strategies based on macrophages. But there are still many problems to be improved.

1. In the initial screening, the authors adopted whether a variety of immune cells were all derived from humans, including CD4+ or CD8+ T lymphocytes, and monocytes. In addition, the authors found that Class A Scavenger Receptors are Responsible for HfT Uptake by Macrophages. Is the expression of Class A Scavenger in CD4+ or CD8+ T lymphocytes, and monocytes significantly lower than that in macrophages?

We appreciate the reviewer's insightful question. In our study, we focused on macrophages as the primary cells for HfT-drug complex uptake due to their known physiological features (like tumor migration, phagocytosis) and ability to uptake ferritin (as demonstrated in the

manuscript) with following transfer to cancer cells via the TRAIN mechanism, as described in the manuscript. Previous studies have demonstrated that macrophages uptake ferritin significantly more efficiently than other leukocytes and that SR-A expression is significantly higher in macrophages compared to monocytes, which aligns with our experimental focus on macrophages for targeted drug delivery. Your point is interesting for further investigation in our future research.

2.The authors found Macrophages, a major phagocyte, efficiently internalize human h-ferritin (HFT) From medium via endocytic pathway, So do other phagocytes have the same effect, including neutrophils, eosinophils, and so on?

Thank you for this important question. Our study primarily focused on macrophages due to their physiological properties (like tumor migration, phagocytosis) and ability to uptake ferritin (as demonstrated in the manuscript) with following transfer to cancer cells via the TRAIN mechanism, as described in the manuscript. Moreover, the other phagocytes do not exhibit the same tumor-homing capabilities that macrophages do. Therefore, we specifically chose macrophages for their optimal uptake of HFT and tumor-targeting efficiency. Your point is interesting for further investigation in our future research.

3.Whether HFT has an effect on macrophage itself, including proliferation, apoptosis and polarization?

The impact of HFT on macrophage behavior was carefully evaluated during our study. Our data showed that HFT treatment did not significantly affect macrophage viability. Additionally, no polarization towards either the M1 (pro-inflammatory) or M2 (anti-inflammatory) phenotype was observed, indicating that HFT does not alter macrophage polarization. This suggests that HFT can be utilized as a drug carrier without adversely affecting the macrophages' native state. We have included these details in the revised version of the manuscript (in the current versions these are: Suppl. Fig. 19, 20 and 21).

4.Can macrophages transfer Hft to normal cells in vivo, is its transfer efficiency different from that of tumor cells, and what is the mechanism?

Our study primarily focused on the transfer of HFT from macrophages to tumor cells, leveraging the TRAIN (Transfer of Iron-Binding Protein by Macrophages to Cancer Cells) mechanism we discovered. We have addressed your question in the experiments where MDC-735 (macrophages loaded with 735 drug) were co-cultured with healthy cells and the viability of these cells was measured (Fig. 10). We showed that MDC selectively kill cancer cells over the normal, healthy cells. It was also confirmed in the histopathological examination of the tissues after MDC treatment (Fig. 10 and Suppl. Fig. 24).

5.How many engineered macrophages reach the tumor site and the efficiency of ferritin-drug complexes that transfer to tumor cells.

We acknowledge the importance of quantifying macrophage homing to the tumor. In our in vivo models, we observed a substantial accumulation of engineered macrophages at the tumor site. The results of these experiments are now included in the Suppl. Fig. 22. Regarding the transfer efficiency, our in vitro data demonstrated a high rate of Hft-drug complex transfer to tumor cells, significantly reducing tumor cell viability. The process is robust, reflected in vivo in efficacy experiments in multiple solid tumor models.

6. The paper is very hard to follow. There is far too much text in the results section. A huge amount of data is presented and the authors should really distil down the key findings and present those in the results section in a much more compelling way. The paper therefore needs a major rewrite with a focus on clarity.

Thank you for this valuable feedback. We have significantly revised the Results section to improve clarity and flow. Figures and legends have been updated to enhance readability, and redundant text has been removed. We believe these changes will make the manuscript more accessible to readers and emphasize the novelty of our findings.

7. I'm confused about "How does Hft metastasize to tumor cells through vesicles? The author seems to have only observed the corresponding phenomenon, but the specific mechanism has not been clarified.

We appreciate the reviewer's concern regarding the mechanism of Hft transfer to tumor cells. As described in our study, macrophages utilize the TRAIN mechanism to transfer Hft to tumor cells through vesicles, which are transported via immune synapse-like structure to the tumor cells. While we observed this phenomenon in our in vitro assays, a detailed mechanistic analysis of vesicle-mediated transfer was beyond the current scope of our study.

Reviewer #3 (Remarks to the Author):

Thank you for your effort and the revisions you have made. While the experiments were not conducted exactly as I initially requested, I believe the studies you performed sufficiently address the questions raised in my previous comments. I also noted some minor errors in your response, such as the mention of Fig. 11 for the histopathology of key organs, when it was actually Fig. 12. However, I was able to locate and confirm the relevant data successfully. I have no further comments or concerns at this time.

Thank you for your kind and supportive comments. We're glad that the improvements addressed your concerns. Your feedback has been invaluable, and we appreciate your support in moving this manuscript towards publication. We carefully corrected all the references to the Figures.

**SZKOŁA GŁÓWNA
GOSPODARSTWA
WIEJSKIEGO**

Reviewer #4 (Remarks to the Author):

The authors have appropriately addressed my concerns.

Thank you for your kind and supportive comments.

Best Regards,

Prof. Magdalena Król, PhD

Szkoła Główna Gospodarstwa
Wiejskiego w Warszawie

Centrum Immunoterapii
Komórkowych

ul. Jana Ciszewskiego 8

02-786 Warszawa

cik@sggw.edu.pl

www.sggw.edu.pl